# Direct Measurements of Ozone Response to Emissions Perturbations in California

Shenglun Wu[1], Hyung Joo Lee[2], Andrea Rohrbacher[3], Shang Liu[4], Toshihiro Kuwayama[4], John H. Seinfeld[5], and Michael J. Kleeman[1]

[1]Department of Civil and Environmental Engineering, University of California Davis, 1 Shields Ave, Davis, CA 95616, USA
[2]Division of Environmental Science and Engineering, Pohang University of Science and Technology (POSTECH), Pohang, Gyeongbuk 37673,South Korea
[3]Department of Chemistry, University of California Irvine, Irvine, CA 92697, USA
[4]Research Division, California Air Resources Board, 1001 I Street, Sacramento, CA 95814, USA
[5]Department of Chemical Engineering, California Institute of Technology, 1200 E. California Blvd, Pasadena, CA 91125, USA

*Correspondence to*: Michael J. Kleeman (mjkleeman@ucdavis.edu)

**Abstract.**

A new technique was used to directly measure $O_3$ response to changes in precursor $NO_x$ and VOC concentrations in the atmosphere using three identical Teflon "smog chambers" equipped with UV lights. One chamber served as the baseline measurement for $O_3$ formation, one chamber added $NO_x$, and one chamber added surrogate VOCs (ethylene, m-xylene, n-hexane). Comparing the $O_3$ formation between chambers over a three-hour UV cycle provides a direct measurement of $O_3$ sensitivity to precursor concentrations. Measurements made with this system at Sacramento, California, between April 2020 – December 2020 revealed that the atmospheric chemical regime followed a seasonal cycle. $O_3$ formation was VOC-limited ($NO_x$ – rich) during the early spring, transitioned to $NO_x$-limited during the summer due to increased concentrations of ambient VOCs with high $O_3$ formation potential, and then returned to VOC-limited ($NO_x$-rich) during the fall season as the concentrations of ambient VOCs decreased and $NO_x$ increased. This seasonal pattern of $O_3$ sensitivity is consistent with the cycle of biogenic emissions in California. The direct chamber $O_3$ sensitivity measurements matched semi-direct measurements of $HCHO/NO_2$ ratios from the TROPOspheric Monitoring Instrument (TROPOMI) onboard the Sentinel-5 Precursor (Sentinel-5P) satellite. Furthermore, the satellite observations showed that the same seasonal cycle in $O_3$ sensitivity occurred over most of the entire state of California, with only the urban cores of the very large cities remaining VOC-limited across all seasons. The $O_3$-nonattainment days (MDA8 $O_3$ > 70 ppb) have $O_3$ sensitivity in the $NO_x$-limited regime, suggesting that a $NO_x$ emissions control strategy would be most effective at reducing these peak $O_3$ concentrations. In contrast, a large portion of the days with MDA8 $O_3$ concentrations below 55 ppb were in the VOC-limited regime, suggesting that an emissions control strategy focusing on $NO_x$ reduction would increase $O_3$ concentrations. This challenging situation suggests that emissions control programs that focus on $NO_x$ reductions will immediately lower peak $O_3$ concentrations, but slightly increase intermediate $O_3$ concentrations until $NO_x$ levels fall far enough to re-enter the $NO_x$-limited regime. The spatial pattern of

increasing and decreasing $O_3$ concentrations in response to a $NO_x$ emissions control strategy should be carefully mapped in order to fully understand the public health implications.

## 1 Introduction

Ground-level ozone ($O_3$) is an oxidant that inflames airways and damages tissue in the respiratory tract leading to increased coughing, wheezing, shortness of breath, and other asthmatic symptoms (US EPA, 2020b). Maximum daily average 8 hour (MDA8) $O_3$ concentrations designed to protect public health are codified in the National Ambient Air Quality Standards (NAAQS) (US EPA, 2021) and the California Ambient Air Quality Standards (CAAQS) (California Air Resources Board, 2007). Seven of the ten cities across the United States with the highest $O_3$ concentrations are located in California (American Lung Association, 2020), making $O_3$ pollution a continued public health threat for millions of California residents more than four decades after $O_3$ abatement efforts began.

$O_3$ levels are often described by the maximum daily average 8-hr concentration. The annual fourth-highest MDA8 $O_3$ concentration averaged over three years has special regulatory significance. This "design value" determines whether the region containing the monitor complies with the $O_3$ NAAQS. $O_3$ design values in California decreased steadily between the years 1980 and 2019 (Figure 1) due to the success of emissions control programs that reduced concentrations of precursors broadly divided into two groups: oxides of nitrogen ($NO_x$) and volatile organic compounds (VOCs) (Parrish et al., 2016; Simon et al., 2015). Continued progress after the year 2010 has been slower, and $O_3$ design values even increased in some air basins between the years 2015 – 2018 (Figure 1). Multiple factors have been proposed to explain the lack of further reductions in $O_3$ concentrations in recent years. These potential factors include: (i) growing importance of precursor VOC emissions not previously accounted for in the planning process as major sources such as transportation have been controlled (McDonald et al., 2018; Shah et al., 2020), (ii) an imbalance in the historical degree of $NO_x$ and VOC reductions (Cox et al., 2013; Parrish et al., 2016; Pollack et al., 2013a; Steiner et al., 2006), or (iii) more frequent heat waves (Jacob and Winner, 2009; Jing et al., 2017; Pusede et al., 2015; Rasmussen et al., 2013; Weaver et al., 2009) and wildfires (Jaffe et al., 2013; Lindaas et al., 2017; Lu et al., 2016; Singh et al., 2012) as a consequence of climate change. All these theories are supported to varying degrees by indirect measurements or model predictions, but there is an absence of strong direct evidence that identifies dominant factors contributing to the increased $O_3$ concentrations. The uncertainty that lingers over the recent $O_3$ trends suggests that fresh approaches are needed to directly verify the optimum emissions control path.

$O_3$ formation has been studied for decades in California, using both measurements and model simulations (Kroll et al., 2020). These past studies provide important background information about the effects of precursor $NO_x$ and VOC species and help build the foundation for new studies. Statistical analyses of long-term surface measurements have determined that lower $NO_x$ concentrations are associated with higher $O_3$ concentrations on weekends (Pollack et al., 2012; Pusede and Cohen, 2012) and

65 higher temperatures are associated with increased VOC emissions and chemical reaction rates, leading to higher $O_3$ concentrations during warm stagnation events (Lafranchi et al., 2011; Nussbaumer and Cohen, 2020). These long-term studies suggest that VOCs are the limiting precursor for $O_3$ formation in the center of large cities, while $NO_x$ is the limiting precursor in downwind areas (Lafranchi et al., 2011; Pusede and Cohen, 2012). Neither long-term analysis method clearly explains the recent trend of increasing $O_3$ concentrations in Los Angeles.

$O_3$ sensitivity has also been analyzed over shorter timescales using ratios of photochemical "indicator" species including $H_2O_2/HNO_3$ and $HCHO/NO_2$ (Sillman, 1995; Tonnesen and Dennis, 2000). Satellite retrievals of $HCHO/NO_2$ from the Global Ozone Monitoring Experiment (GOME), the SCanning Imaging Absorption spectroMeter for Atmospheric CartograpHY (SCIAMACHY), the Ozone Monitoring Instrument (OMI), and the TROPOspheric Monitoring Instrument (TROPOMI) have

75 extended these $O_3$ sensitivity calculations over broad geographical regions (Chossière et al., 2021; Duncan et al., 2010; Jin et al., 2017; Martin et al., 2004; Schroeder et al., 2017a). The short-term measurements generally support the findings from the long-term studies but once again fail to identify the dominant factor(s) driving the recent increase in $O_3$ design values. Reactive chemical transport models (CTMs) have been used extensively to predict the effectiveness of candidate emissions control programs (Brown, 2018; California Air Resources Board, 2018; Meng et al., 1997; Sillman, 1999) and so one might expect

that they would provide the most detailed explanation for recent $O_3$ trends. Models are necessarily incomplete approximations to highly complex real-world systems, and so they are often incapable of predicting subtle features in pollutant trends. No model calculation has been able to reproduce the observed increase in $O_3$ design values (Parrish et al., 2017). It is unclear whether this failure stems from a lack of accurate emissions trends, an incomplete description of atmospheric chemistry, or an incomplete representation of the effects of shifting climate on $O_3$ formation mechanisms.

Recent advances in measurement techniques provide new tools to study $O_3$ sensitivity directly. Mobile smog chambers bridge the gap between laboratory studies and the real atmosphere. Past studies have designed mobile smog chambers to measure the aging of secondary pollutants (i.e., $O_3$, SOA) from certain emission sources (Howard et al., 2008, 2010b; Li et al., 2019; Platt et al., 2013; Presto et al., 2011). It is difficult to evaluate sensitivity of secondary pollutants formed from multiple sources

using a single smog chamber. Recently, a mobile dual smog chamber system has been used to directly measure the SOA formation in ambient air (Jorga et al., 2020; Kaltsonoudis et al., 2019). The design used in the current study consists of three chambers that can simultaneously measure the non-linear response of $O_3$ formation to $NO_x$ and VOC perturbations. The automated valve and sampling incorporated into this design also allows long-term remote field measurements to evaluate the seasonal trends in $O_3$ sensitivity. At the same time, the satellite TROPOspheric Monitoring Instrument (TROPOMI) launched

by the European Space Agency (ESA) in October 2017 provides measurements of HCHO and $NO_2$ tropospheric vertical column densities (TVCDs) with 3.5 km × 5.5 km spatial resolution that can start to resolve $O_3$ perturbations around major sources such as wildfires (Ialongo et al., 2020; Veefkind et al., 2012; Vigouroux et al., 2020b). The purpose of this study is to combine these two new measurement techniques into a detailed analysis of $O_3$ sensitivity to precursor $NO_x$ and VOC emissions

spanning an entire spring-summer-fall cycle in California. Daily measurements from smog-chamber perturbation experiments are analysed for short-term trends (day-of-week) and long-term trends (seasonal variation) to reveal the effects of traffic, natural vegetation, and wildfires. The direct $O_3$ sensitivity measurements are then combined with TROPOMI HCHO/$NO_2$ ratios to extend our understanding of the $O_3$ sensitivity across the entire state of California.

## 2. Methods

### 2.1 Ground-based measurement

Three identical transportable smog chambers were used to directly measure basecase $O_3$ concentration and $O_3$ sensitivity to precursor $NO_x$ and VOC. Each chamber was constructed from fluorinated ethylene propylene (FEP) with a volume of 1 m$^3$ housed in an enclosure measuring 2.13 m H x 1.22 m L x 1.22 m W. UV lamp panels were placed on the floor and the roof of the chamber support frame. Each panel can hold up to six UV lamps (Sylvania, F40BL 40W T12) that emit at wavelengths between 280–400 nm. The lamp panels were configured to produce 50 W/m$^2$ to replicate the mid-day photochemistry in California during the summer. The enclosure walls were constructed from polished aluminium with total reflectivity of ~95%. Figure S1 represents the cross-sectional view of the transportable smog chamber system.

One cubic meter of ambient air was injected into the FEP chambers at the start of an experiment using a Teflon diaphragm pump (Model DOA-V751-FB, Gas Manufacturing, Benton Harbor, MI, USA) operating at a flow rate of 10 L·min$^{-1}$ for each chamber. Solenoid valves were configured to inject perturbation gases $NO_x$ (8 ppb $NO_2$) and VOC surrogates (4.4 ppb ethylene, 2.8 ppb n-hexane, and 0.8 ppb m-xylene) respectively into chambers #1 and #3 for comparison to basecase chamber #2. Perturbation gases were added halfway through the chamber filling operation so that they would be thoroughly mixed with the ambient air during the remainder of the chamber filling process. The composition of the VOC perturbation was based on the VOC mixture used to determine ozone formation potential (Carter et al., 1995). The magnitude of the perturbations was selected to be as small as possible while still generating an observable change in monitored ozone concentrations. A single set of monitors sequentially measured concentrations within each chamber to increase the precision of the inter-chamber comparisons. The current experiment includes measurements of $NO_x$ (Model nCLD-855-Yh, ECO Physics, Duernten, Switzerland), $NO_y$ (sum of all oxidized atmospheric odd-nitrogen species) (Model 42i, Thermo Fisher, Franklin, MA, USA), $O_3$ (Model 205, 2B technology, Boulder, CO, USA), and temperature – relative humidity sensor (Model RH-USB, Omega Engineering, Norwalk, CT, USA). The total sample flow rate for all monitors was approximately 3 L/min. Seven measurements with a duration of 10 min were made from each chamber resulting in a total sample volume of 210 L air, or approximately 21% of the chamber volume (leaving 79% of the total air in the chamber). The shape of the chambers was not greatly distorted at any point during the experiment. Chambers were drained at the conclusion of an experiment using a rotary vane vacuum pump (Model 0523-101Q-G588DX, Gast Manufacturing, Benton Harbor, MI, USA). All chamber operations were controlled

automatically using a program written in LabView that interfaced with a customized set of data acquisition devices and solenoid valves (DAQ-SV).

The consistency of the $O_3$ formation rates across chambers was tested in a controlled laboratory environment prior to deployment in the field. All three 1 $m^3$ FEP chambers were filled with laboratory air and were perturbed by an equal mixture of both $NO_x$ and VOC prior to 180 min of UV exposure. Several blank tests were performed by adding zero air (AI 0.0Z-K,

Praxair) into all three chambers as part of the consistency tests to further develop confidence in the chamber measurements. Figure S2 illustrates the agreement between chamber #1 and #3 $O_3$ measurements vs. chamber #2 $O_3$ measurements during a typical QA/QC check. Uncertainty between chamber measurements is ~ 1% across a wide range of concentrations. The loss rate of $O_3$ to chamber walls was determined in the dark for all three 1 $m^3$ FEP chambers filled with identical $NO_x$-VOC

mixtures. Average loss rates of 5% $hr^{-1}$ were calculated over the 3-hour experiments. Loss rates were identical for all chambers in the system and so this issue will not influence the comparisons between chambers in the current study.

To confirm that chamber measurements represent the behavior observed in the atmosphere, weekly-averaged $O_3$ concentrations in the basecase chamber were compared to weekly-averaged ambient $O_3$ concentrations measured at the nearby monitoring

station (marked in Figure S5) from April to December 2020 in Figure S3. The $O_3$ concentrations in the basecase chamber at the start of each experiment were similar to the ambient $O_3$ concentrations, indicating that the gas-phase chemical composition related to $O_3$ formation was not changed while injecting ambient air into the chamber. The $O_3$ formation in the chamber generally reflects the $O_3$ chemical production from the in-situ ambient air around 10 am ~ 12 pm in the morning, while the ambient $O_3$ is influenced by chemical production, mixing, and deposition (Cazorla et al., 2012). As expected, the initial rate

of $O_3$ formation in the chamber is therefore higher than the initial rate of change in the ambient $O_3$ concentrations. The current experiment is focused on measuring the response of this chemical production rate to changes in precursor $NO_x$ and VOC concentrations because this most closely approximates the local effects of potential emissions control programs.

VOC measurements are useful to help interpret $O_3$ formation trends and to identify the chemical regime on the $NO_x$-VOC

isopleth for $O_3$ (Seinfeld and Spyros N. Pandis., 2016). Ground-level daily VOC measurements from Photochemical Assessment Monitoring Stations (PAMS) are only available for a limited number of summer months and so alternative indicator species were investigated. Baker (2008) found that non-methane hydrocarbon (NMHC) concentrations were correlated with CO concentrations in 28 U.S. cities during the years 1999 – 2005. This may reflect situations where dominant sources that emit CO also emit large amounts of NMHC, or it may reflect situations where relatively constant sources of CO

and NMHC are correlated because they are diluted by the same amount of atmospheric mixing. The success of emissions control programs targeting anthropogenic VOCs has increased the relative importance of residual biogenic VOCs in many

urban atmospheres across the US (US EPA, 2020a). Biogenic sources do not emit CO but biogenic VOCs can react in the atmosphere to produce CO (Hudman et al., 2008). CO also acts as an indicator of atmospheric mixing that equally affects all primary sources. In an effort to improve the ability of CO to represent biogenic VOCs in the current study, an additional metric

was calculated by multiplying the measured CO concentrations by the temperature and relative humidity-induced enhancement factor for isoprene emissions (Guenther et al., 1991). Figure S4 shows the correlation between measured VOC reactivity (VOCR) and CO*Biogenic at Sacramento during summer months between 2010 – 2019. VOCR was calculated from PAMS measurements of VOC concentrations multiplied by their reaction rate constant with OH (Chen et al., 2010; Kleinman, 2005; Steiner et al., 2008). VOCR and CO*Biogenic are reasonably well correlated (r = 0.6, p < 0.001), while VOCR and CO were

less correlated (r = 0.39). This analysis supports the preference for CO*Biogenic as an approximate surrogate for VOCR in the current study, with the understanding that real time measurements of VOCR would be highly preferred in future studies.

**2.2 Satellite data**

Tropospheric HCHO and $NO_2$ retrievals (Level 2; Unit: $mol/m^2$) over California were obtained from the TROPOMI for February – October 2020. The TROPOMI is onboard the Sentinel-5 Precursor (Sentinel 5-P) satellite, which was launched by

the ESA in October 2017. The polar-orbiting satellite enables quantitative information on trace gases to be retrieved approximately at 13:30 local sun time (ascending node) each day on a global scale. The retrieval algorithms for TROPOMI $NO_2$ data use the measurements of the earth's radiance in the visible absorption wavelengths (405 – 465 nm) made by the hyperspectral imaging spectrometer. The algorithms first derive the total slant column density of $NO_2$ using a Differential Optical Absorption Spectroscopy (DOAS) method. The total slant column $NO_2$ is then separated into stratospheric and

tropospheric slant column densities of $NO_2$ while utilizing information from a data assimilation system. Finally, the tropospheric vertical column density of $NO_2$ is obtained by applying conversion factors, called air mass factors (AMFs), to the tropospheric slant column density of $NO_2$. The retrievals of TROPOMI HCHO data apply a similar DOAS method to the ultraviolet (UV) wavelengths (328.5 – 359 nm) of the solar spectrum. Further details about the TROPOMI data are provided by Veefkind et al. (2012), Van Geffen et al.(2020), and De Smedt et al.(2018).


The spatial resolution of TROPOMI $NO_2$ and HCHO TVCDs are 3.5 km × 5.5 km, which is finer than that of the predecessor OMI (13 km × 24 km). Quality assurance (QA) values were obtained alongside the HCHO and $NO_2$ data, and only measurements with QA values ≥ 0.50 were retained to ensure good data quality and sufficient data points when computing monthly averages (Van Geffen et al., 2021). Correction factors were not applied to TROPOMI data in the current study.

Verhoelst et al. (2021) and Vigouroux et al. (2020a) analyzed the accuracy of the TROPOMI data using ground-based measurement sites across the globe. Measurements were not made in California, but several of the evaluation sites had attributes similar to locations in California. Bias in daily TROPOMI $NO_2$ retrievals varied between -15% to -56% in moderately polluted areas with $NO_2$ column measurements between $3\times10^{15}$ - $14\times10^{15}$ molec $cm^{-2}$ (typical for moderate-sized cities in California). The bias in TROPOMI HCHO measurements ranged between +26%±5% at low HCHO levels to -

30.8%±1.4% at high HCHO levels. HCHO levels measured in Sacramento (~0.6 x $10^{15}$ molec $cm^{-2}$) had a bias of approximately zero. These results suggest that TROPOMI measurements over California almost certainly contain some amount of bias that could only be removed through a comparison to measurements from a ground-based network. Application of global-average bias correction factors would not change the trends in HCHO and $NO_2$ in time and space even if they would change the absolute magnitude of those values. The current analysis will therefore focus on trends in the TROPOMI

measurements.

### 2.3 Experimental description

$O_3$ sensitivities to precursor $NO_x$/VOC concentrations were measured in central Sacramento, CA (N 38.57, W 121.49) from April – December 2020 (222 experiement days out of a total of 251 days). Sources in the vicinity of the site include commercial office buildings, restaurants, two major highways, freight and passenger rail lines, a shipping port, and suburban residences

(see map in Figure S5). Grab samples of ambient air were collected between 10:00 AM to 12:00 PM to characterize the daytime $O_3$ formation rates in the presence of variable atmospheric mixing and regional emissions. Sensitivities were based on perturbation concentrations of approximately 8 ppb of $NO_x$ injected into chamber #1 and 8 ppb of VOC surrogates injected into chamber #3. Initial gas concentrations were measured from the full chambers in the dark over a 30 min period (10 min for each chamber). The UV lamp panels were then illuminated for 180 min and the chamber concentrations were measured in a

continuous cycle of 10 min intervals over a total of seven cycles. Each active monitoring period lasted 210 min (=30 min of dark measurements + 180 min of light measurements). Measurements in different chambers are made at different times, making it difficult to compare chamber results at the conclusion of the experiment. It was noted that $O_3$ concentrations within each chamber averaged in each 10 min sampling interval increased linearly over the 180 min period when the UV lights were on. A linear regression model was therefore applied to extrapolate $O_3$ concentrations in each chamber to the end of the

measurement period to facilitate direct comparisons between the basecase chamber #2 and perturbed chambers #1 and #3. $O_3$ concentration after 3-hour UV exposure. The difference of O3 concentration after 3-hour UV exposure was calculated between chamber #1 to chamber #2 ($\Delta O_3^{+NO_x}$), and chamber #3 to chamber #2 $\Delta O_3^{+VOC}$ to quantify the $O_3$ sensitivity. An example of typical day of $O_3$ results analysis is shown in SI.

### 2.4 Chamber model description

A chamber model developed by Howard et al (2008, 2010a, 2010b) was employed as a part of this analysis to quantify the sensitivity of the $O_3$ response to $NO_x$ perturbations under different experimental configurations. The chemical reaction system used by the chamber model is based on the SAPRC11 chemical mechanism (Carter and Heo, 2013) with wall loss rates based on the measured value of 5% $hr^{-1}$. The time integration procedures used to solve the set of differential equations that predict concentrations as a function of time are taken from the full UCD/CIT chemical transport model (Venecek et al., 2018; Ying et

al., 2007). Day-specific values of NO, $NO_2$, and $O_3$ initial concentrations used in the chamber simulations are based on

measurements near the study location. VOC initial concentrations used in the chamber simulations are based on UCD/CIT simulations over the study location. The seasonal profile of the simulated VOC concentrations matches the CO*biogenic trends illustrated in Figure 2, but the amplitude of the simulated seasonal trend was damped. VOC initial concentrations used in the chamber simulations were therefore scaled to match the amplitude of the CO*biogenic factor. Section 4.1 presents a

sensitivity study on the chamber measurement result using the chamber model described here.

## 3. Results

### 3.1 Chamber measurement and satellite results in Sacramento

### 3.1.1 Monthly variation of ambient gas concentrations

Figure 2 compares the ground-based measurements and the TROPOMI column measurements of $NO_x$ and VOC surrogate

concentrations at the Sacramento sampling site. Good agreement is observed between the time trends of the chamber and TROPOMI satellite remote sensing measurements. Both techniques identify strong seasonal patterns for the concentrations of the $O_3$ precursors.

Figure 2a shows the monthly averaged TROPOMI satellite $NO_2$ measurements and the boxplot of daily chamber $NO_2$

measurements at Sacramento between February – December 2020. $NO_2$ concentrations remained relatively stable between April and July and then sharply increase in August – September possibly due to increased wildfires in the late summer months. Enrichment of $NO_2$ and other pollutants in wildfire plumes has been noted in previous research (Jaffe and Wigder, 2012). The open boxes in Figure 2a represent days within the months of August – November that were not influenced by wildfire smoke (Rohrbacher and Kuwayama, n.d.), leading to reduced $NO_2$ concentrations. The upward trend in $NO_2$ concentrations in October

– December, 2020 is likely associated with decreased boundary layer heights and increased fuel consumption for heating during the colder fall – winter season. This seasonal association can also be viewed in the decreasing TROPOMI $NO_2$ levels measured during the warmer spring season (February – April, 2020). TROPOMI column meausrements will not directly depend on boundary layer height, but increased boundary layer heights are usually associated with higher average boundary layer wind speeds, leading to downwind advection and dispersion of pollutants. The effects of reduced transportation emissions

in March – April, 2020 caused by COVID-19 shelter-in-place orders are notably minor in the ambient $NO_2$ measurements. Although light-duty vehicle traffic decreased by as much as 50% during this time period, heavy-duty truck traffic was more constant (Liu et al., 2020; Parker et al., 2020). The ground-based measurement site is 0.8 and 1.8 km from two major freeways, but $NO_x$ concentrations at this site do not appear to be strongly influenced by the COVID-19 reduction in light-duty traffic activity. Increasing $NO_x$ emissions from residences and relatively quick recovery of the heavy-duty traffic compared to the

light-duty traffic may also minimize COVID-19 effects on $NO_x$ concentrations (Liu et al., 2021). The seasonal pattern of $NO_x$

concentrations driven by wildfires, reduced boundary layer height, and increased residential fuel consumption appears to dominate at the urban Sacramento location.

Figure 2b shows the monthly averaged TROPOMI satellite HCHO levels and the daily ground-based CO*Biogenic concentration at the Sacramento sampling site. The agreement between the seasonal trend in the CO*Biogenic and TROPOMI HCHO builds confidence in the use of CO*Biogenic as a ground-based indicator of VOC concentrations at this location. Both indicators suggest that VOC concentrations increased from April – August 2020, and sharply declined in October 2020. Wildfires can emit large amounts of VOCs that can be transported to urban areas (Zhang et al., 2018). It is possible that wildfires contributed to the highest VOC concentrations observed between August and September 2020. Removing the days influenced by wildfires (open box) still leaves a strong seasonal trend with increasing VOC concentrations between April – August 2020, which is consistent with increasing VOC emissions from biogenic sources. Biogenic VOC (BVOC) emissions increase during warmer spring months and continue to increase as temperatures rise into summer (Guenther et al., 2006, 1991). The CO*Biogenic factor inherently incorporates this effect, but the strong agreement between the TROPOMI HCHO levels and the CO*biogenic metric in Figure 2b suggests that the seasonal pattern of the biogenic emissions is a real feature of the dataset and not an artifact of how the CO*biogenic metric was constructed. Similarly, the declining VOC concentration observed in October, 2020 and beyond, matched the expected decrease of biogenic emissions during the colder fall and winter seasons when vegetation becomes dormant. The seasonal pattern illustrated in Figure 2b suggests that BVOC is an important precursor of HCHO in Sacramento.

PAMS measurements of ground-level isoprene concentrations in Sacramento are shown as blue diamonds in Figure 2b. Isoprene is highly reactive in the atmosphere and so PAMS measured concentrations are lower than 4 ppb. The limited time period of available measurements makes it difficult to discern seasonal trends, but the slightly lower measured isoprene concentrations in July, slightly higher isoprene concentrations in August followed by decreasing (non-wildfire) isoprene concentrations in September generally match the VOC trends generated using both TROPOMI HCHO and CO*Biogenic. Once again, the agreement between the three independent techniques builds confidence in the overall assessment of VOC seasonal trends.

Volatile chemical products (VCP) are another important category of VOCs emissions (McDonald et al., 2018). The expanded usage of spray disinfectant and sanitization products during the COVID-19 pandemic might have been a significant source of VOCs in the urban area, but the expected usage pattern of these products does not include a sharp decline in the fall period. The seasonal pattern of VOC concentrations increasing during spring – summer and decreasing during fall – winter is more consistent with a combination of biogenic sources and wildfires, as discussed above.

### 3.1.2 Seasonal trends in O₃ sensitivity

Figure 3a shows the monthly trends in measured $\Delta O_3^{+NO_x}$ and TROPOMI HCHO/NO₂ from February 2020 to December 2020 at the Sacramento site. The $\Delta O_3^{+NO_x}$ value represents the change in O₃ concentrations in response to a +8 ppb NOₓ perturbation. O₃ formation is NOₓ-limited when the $\Delta O_3^{+NO_x}$ value is positive, and VOC-limited when the $\Delta O_3^{+NO_x}$ value is negative. Changes in the absolute magnitudes of the $\Delta O_3^{+NO_x}$ values reflect the degree of O₃ sensitivity to the NOₓ perturbation. $\Delta O_3^{+NO_x}$ and TROPOMI HCHO/NO₂ both increase from April to August 2020, and then sharply decline in October 2020. By comparing the transition points of $\Delta O_3^{+NO_x} = 0$ and TROPOMI HCHO/NO₂ = 4.6 (discussed in Section 3.2), it is evident that O₃ formation evolved from VOC-limited conditions in spring towards NOₓ-limited conditions from June to August, followed by a return to VOC-limited conditions after October 2020. It is notable that the seasonal trend for $\Delta O_3^{+NO_x}$ matches the trend of increased BVOC emissions during the summer and increased NOₓ emissions during the winter. The travel restrictions associated with COVID-19 that occurred in March – May 2020, appeared to have little impact on the overall seasonal trends in $\Delta O_3^{+NO_x}$ behavior.

The median ground-based $\Delta O_3^{+NO_x} < 0$ indicates VOC-limited conditions in September 2020, but the TROPOMI satellite HCHO/NO₂ > 4.6 indicates NOₓ-limited conditions for this same month. Removing the wildfire days from the analysis period (open box in Figure 3a) did not reconcile the two measurements. The divergence of the ground-based measurements and satellite measurements in this month may reflect the presence of elevated plumes of wildfire smoke above the monitoring site that were detected by the satellite measurements (Jin et al., 2017). Cleaner air at the ground-based monitors, therefore, yielded $\Delta O_3^{+NO_x}$ values in a different chemical regime than the satellite measurements that are based on the tropospheric vertical column densities. This comparison suggests that ground-based measurements may be required to supplement satellite-based measurements to fully characterize the surface O₃ formation regime under special circumstances that generate concentrated pollution layers above the ground-level.

Removing the days influenced by wildfires from the chamber measurement (open box) and TROPOMI satellite measurement (open diamond) in Figure 3a reduces both $\Delta O_3^{+NO_x}$ and TROPOMI HCHO/NO₂. Figure S7 compares TROPOMI HCHO and NO₂ on wildfire days and non-wildfire days. Median TROPOMI HCHO measurements increased by 44% and TROPOMI NO₂ measurements increased by 14% on wildfire days. The comparison between wildfire vs. non-wildfire days implies that wildfires emit more VOC than NOₓ, which is in agreement with previous studies (Jaffe and Wigder, 2012). It is also notable that the decrease of $\Delta O_3^{+NO_x}$ is larger than the decrease in TROPOMI HCHO/NO₂. This observation might once again reflect the fact that the wildfire identification algorithm (Rohrbacher and Kuwayama, n.d.) was based on ground-level measurements that do not flag all of the days with elevated plumes above the monitoring site that could differentially affect the satellite measurements.

Figure 3b shows the monthly variation of ground-based $\Delta O_3^{+VOC}$ and TROPOMI satellite HCHO/NO$_2$ February – December 2020, at the Sacramento sampling site. $\Delta O_3^{+VOC}$ (Figure 3b) has an inverse time trend compared to $\Delta O_3^{+NOx}$ and TROPOMI HCHO/NO$_2$ (Fig 2a). The $\Delta O_3^{+VOC}$ trend is well-correlated to the TROPOMI HCHO/NO$_2$ trend plotted on a reversed axis between April – August 2020, but the two trends diverge in September – October 2020 when wildfires were prevalent.

Removing the wildfire days from August to October (open box) increased the ground-based $\Delta O_3^{+VOC}$, once again suggesting that wildfires contributed more VOCs than NO$_x$ to the atmosphere (Altshuler et al., 2020). The divergence between the ground-based $\Delta O_3^{+VOC}$ measurements and TROPOMI satellite HCHO/NO$_2$ measurements during the wildfire season once again reflects the presence of elevated plumes that were measured by the satellite but not by the ground-based monitors (Schroeder et al., 2017a).

### 3.1.3 Weekend effect

Figure 4 separately plots concentrations of O$_3$ precursors and O$_3$ sensitivity on weekdays (shaded bars) and weekends (open bars) during the current study period. Direct wildfire days have been removed from the analysis (Rohrbacher and Kuwayama, n.d.) to focus on the day-of-week patterns. Hypothesis tests were carried out to determine if weekday and weekend responses were similar in each month. The results indicate that weekend reductions in NO$_2$ concentrations were significant at a 90% confidence level (or higher) before July. The similarity between weekday and weekend NO$_2$ concentrations after July may be associated with increased NO$_x$ emissions from wildfires in the late summer and space heating in the fall – winter since neither of these sources follows a weekday/weekend pattern. Although days directly affected by the wildfire smoke were removed from the analysis, residual emissions from smoldering fires and multi-day recirculation of air mass that have been affected by wildfire smoke may have contributed to elevated regional NO$_x$ concentrations through the formation of reactive nitrogen reservoir species such as peroxyacetyl nitrate (PAN) that can be transported over long distances (Lindaas et al., 2017). The CO*Biogenic VOC surrogate did not display statistically significant differences between weekdays vs. weekends except in June and July. Extremely hot days (> 35°C) occurred on weekdays in June and weekends in July, driving the CO*Biogenic factor higher.

Reduced NO$_x$ emissions on weekends are reflected in the O$_3$ sensitivity to precursors shown in Figure 4c and d. The median $\Delta O_3^{+NOx}$ sensitivity is higher on weekends for most months indicating that the atmosphere was more NO$_x$-limited. Large variability in the data makes the weekend vs. weekday $\Delta O_3^{+NOx}$ response statistically significant at the 90% (or higher) level only in April, September, and October. The large weekend reductions in median NO$_2$ concentrations detected in May and June did not lead to significantly higher weekend $\Delta O_3^{+NOx}$, possibly because of higher weekday median VOC concentrations in these months. Median O$_3$ sensitivity was NO$_x$-limited ($\Delta O_3^{+NOx} > 0$) on both weekdays and weekends from June to August when BVOC emissions are expected to be highest. In spring and early fall (April, May and September), the median weekday

$O_3$ sensitivity is VOC-limited but the median weekend $O_3$ sensitivity is $NO_x$-limited. In late fall and winter (October ~ November), the median $O_3$ sensitivity is VOC-limited on both weekends and weekdays. Weekend $NO_x$ reductions have an inverse effect on $\Delta O_3^{+VOC}$ shown in Figure 4d compared to $\Delta O_3^{+NO_x}$. The median $\Delta O_3^{+VOC}$ is lower on weekends than weekdays
because the $O_3$ formation is more $NO_x$-limited on weekends.

### 3.1.4 $O_3$ isopleth measurements

Figure 5 summarizes the $NO_x$, CO*Biogenic, $O_3$, $\Delta O_3^{+NO_x}$, and $\Delta O_3^{+VOC}$ measurements in Sacramento from April to December in 2020 in the format of an $O_3$ isopleth diagram. Each data point in Figure 5 corresponds to measurements on a single day. The color of each symbol represents the $O_3$ concentration in the basecase chamber after 3-hours of UV irradiation. The $NO_x$ and
360 CO*Biogenic scale factors are relative to the $NO_x$ and CO*biogenic levels measured on the day with the mean $O_3$ concentration. The arrow attached to each data symbol points in the direction of maximum $\Delta O_3$ in response to $NO_x$ and VOC addition. The magnitude of the arrow corresponds to the strength of the $\Delta O_3$ response. All arrows generally point right, meaning that VOC addition increased $O_3$ concentrations. Arrows pointing to the bottom right indicate that $NO_x$ addition decreased the $O_3$ concentration, while arrows pointing to the upper right indicate that $NO_x$ addition increased the $O_3$
concentrations. The most effective emissions control program acts in the direction opposite to each arrow.

The mixture of daily data points (yellow to red points) shows the $O_3$ isopleth pattern where higher $O_3$ concentration (darker color) exists at higher $NO_x$ and VOC concentrations. The combination of the colors and the arrows illustrated in the isopleth diagram help to define the measured "ridgeline" in the $O_3$ isopleth diagram that denotes the transition between VOC-limited
chemistry and $NO_x$-limited chemistry at Sacramento. Arrows in the upper left of the diagram point downwards (VOC-limited) towards the ridgeline, while arrows in the lower right of the diagram point upwards ($NO_x$-limited) towards the ridgeline. The atmospheric system experiences a range of conditions throughout the nine-month study period that moved the measurements around the $O_3$ isopleth diagram. The average seasonal cycle is illustrated in Figure 5 using monthly-average points shown as blue circles with white month numbers. The monthly-average $O_3$ chemical regime traces an oval path through the isopleth
diagram as $NO_x$ concentrations decrease and CO*Biogenic (proxy of VOC) concentrations increase moving from spring to summer months. $NO_x$ concentrations increase rapidly in fall while CO*Biogenic concentrations simultaneously decrease at the Sacramento sampling location, transitioning the $O_3$ chemistry to VOC-limited conditions. The pattern is expected to reverse for the months of January – March (not shown) to produce a repeatable annual cycle. The direct measurement of the seasonal pattern of the $O_3$ chemical regime clearly illustrates the effects of $NO_x$ and VOC emissions controls at different times
of the year.

### 3.1.5 Extreme value analysis for O₃ sensitivity

The days with the highest measured O₃ concentrations are of particular interest in the current study since emissions control programs are traditionally tailored to reduce the O₃ design value, which is determined by MDA8 O₃ concentration. Figure 6 illustrates box-and-whisker plots of measured $\Delta O_3^{+NO_x}$, and $\Delta O_3^{+VOC}$ at Sacramento binned according to the MDA8 O₃ concentration measured at the monitoring station near the chamber measurement site. The right two bins, corresponding to the O₃-nonattainment days (MDA8 O₃ > 70 ppb), have O₃ sensitivity in the NOₓ-limited regime where NOₓ addition increases O₃ concentrations and VOC addition has minor effects on O₃ concentrations. These measurements suggest that a NOₓ emissions control strategy would be most effective at reducing these peak O₃ concentrations. In contrast, a large portion of the days with MDA8 O₃ concentrations below 55 ppb were in the VOC-limited regime, suggesting that an emissions control strategy focusing on NOₓ reduction would increase O₃ concentrations. VOC controls on these intermediate days would be difficult, however, if biogenic VOCs account for the majority of the O₃ formation. This challenging situation suggests that emissions control programs that focus on NOₓ reductions will immediately lower peak O₃ concentrations, but slightly increase intermediate O₃ concentrations until NOₓ levels fall far enough to re-enter the NOₓ-limited regime.

Additional statistical analysis was carried out to characterize the extreme values in the O₃ sensitivity plots (Coles, 2001; Gilleland and Katz, 2016). Extreme value analysis characterizes high concentrations using "return levels" corresponding to a specified time period (T). In the context of the current analysis, the return level is the ΔO₃ perturbation response that is expected to be exceeded once during the specified time period. The probability of exceeding the return level is therefore 1/T. Figure 7 shows the 90-day return level for $\Delta O_3^{+NO_x}$ and $\Delta O_3^{+VOC}$ sensitivity based on statistical analysis of the measured perturbation response in each month. The 90-day time period was chosen to correspond to the time period inherent in the O₃ design value values that are based on the annual 4th highest O₃ concentration averaged in three years (12 "exceedances" / 1095 days equals approximately one "exceedance" / 90 days). The 90-day return value of O₃ sensitivity can therefore be viewed as the design value for O₃ sensitivity. Figure 7 shows that the 90-day return levels for O₃ sensitivity and the median O₃ sensitivity follow similar seasonal trends, but the extreme values are shifted higher such that they are NOₓ-limited from April to December, except November which is slightly VOC-limited. The positive 90-day return levels of $\Delta O_3^{+NO_x}$ once again suggest the NOₓ control is an efficient strategy to reduce peak O₃ concentrations in Sacramento.

### 3.2 Chamber and TROPOMI data correlation

The consistency between the NOₓ and VOC measurements made using ground-based chambers and satellite observations enables a joint analysis to directly calculate the TROPOMI HCHO/NO₂ ratio at the transition between NOₓ and VOC limited O₃ formation regimes. Three circular buffers (2.5, 5, and 7.5 km radii) centered on the monitoring location were used to generate the TROPOMI HCHO/NO₂ ratio that was then compared to the measured $\Delta O_3^{+NO_x}$ ratio at the monitoring site. The HCHO/NO₂ ratio generated using the 5 km radius buffer shows the best correlation with ground-based chamber results shown

in Figure 8a (results from other buffers are shown in Figure S8). Linear regression analysis between 1-week-averaged $\Delta O_3^{+NO_x}$ and HCHO/NO$_2$ with and without wildfires shows that removing the wildfires always improves the correlation coefficient (R), likely because the elevated wildfire plumes have different effects on surface vs. integrated column measurements. The regression carried out using a 5 km buffer radius with wildfires removed yielded a correlation coefficient R = 0.62 (p < 0.001). The transition point between NO$_x$-limited and VOC-limited conditions (corresponding to $\Delta O_3^{+NO_x}$ = 0) occurs when HCHO/NO$_2$ = 4.6 (95% confidence interval: 4.39 ~ 5.90). When the TROPOMI satellite HCHO/NO$_2$ ratio fell below 4.6 then the ground-based measurement of $\Delta O_3^{+NO_x}$ was usually negative, and when the satellite HCHO/NO$_2$ ratio rose above 4.6 then the ground-based measurement of $\Delta O_3^{+NO_x}$ was usually positive. Ordinary lease square (OLS) regression was used to estimate the transition point HCHO/NO$_2$ = 4.6 between chemical regimes. This approach does not account for uncertainty in chamber $\Delta O_3^{+NO_x}$. Repeating the analysis using reduced major axis (RMA) regression that accounts for errors in both x and y yields an estimated transition point HCHO/NO$_2$ = 4.4 between chemical regimes (Figure S9).

The HCHO/NO$_2$ transition point directly measured in the current study is consistent with previous estimates constructed from the combination of satellite measurements and routine ground-based O$_3$ monitoring data (Jin et al., 2020). Other previous efforts to estimate HCHO/NO$_2$ value at the transition point between NO$_x$-limited and VOC-limited regimes typically couple satellite HCHO/NO$_2$ measurements with O$_3$ sensitivity or O$_3$ sensitivity indicators (i.e., LNO$_x$/LRO$_x$) predicted using reactive chemical transport models. These hybrid studies predict HCHO/NO$_2$ transition points lower than the value of 4.6 derived in the current study. Martin (2004) used HCHO/NO$_2$ from GOME to calculate the regime transition value HCHO/NO$_2$=1.0 for polluted areas across the globe. Duncan (Duncan et al., 2010) used OMI to estimate the regime transition value HCHO/NO$_2$=1~2 across the continental U.S.. Schroeder (2017b) found the transition range could between HCHO/NO$_2$=1.3~5.0 during DISCOVER-AQ in Houston. These estimated HCHO/NO$_2$ transition values vary due to the different satellite resolution, retrieval algorithms, and inherent air pollution patterns over the different study areas. The finer resolution satellite data used in the current study combined with direct ground-based measurements of O$_3$ sensitivity should provide accurate information for the HCHO/NO$_2$ transition point between chemical regimes over California.

### 3.3 TROPOMI O$_3$ sensitivity in California

Figure 9 displays the monthly-average spatial distribution of TROPOMI HCHO/NO$_2$ ratios across California for the time period April – October, 2020. Overall, TROPOMI HCHO/NO$_2$ was the lowest (mean (Standard deviation) = 3.5 (1.2)) in April and the highest (mean (standard deviation) = 9.7 (3.2)) in July. The seasonal pattern of increasing NO$_x$ limitation during the summer months at Sacramento (Figure 3a, b) is mirrored across most of California (Figure 9), especially in the mountainous areas with dense vegetation. The majority of California is in the VOC-limited regime in April and May due to the low BVOC emissions. Only very remote regions with low NO$_x$ concentrations are still in the NO$_x$-limited regime during these spring months. Most areas outside of major urban centers transition toward NO$_x$-limited conditions between June and September as

ambient temperature and BVOC emissions increase.  These areas then transition back to the VOC-limited regime in the fall months beginning in October as temperatures decrease and vegetation becomes dormant.

Large urban centers including Los Angeles, San Diego, and the San Francisco Bay Area exhibit low HCHO/NO$_2$ ratios (VOC-limited conditions) throughout the study period. These urban areas contain less vegetation and larger numbers of NO$_x$ sources

than outlying suburban and rural areas. Therefore, reducing NO$_x$ emissions in these urban centers may increase monthly-average O$_3$ concentrations throughout the year. The HCHO/NO$_2$ ratio in California's Central Valley is lower than the HCHO/NO$_2$ ratio in the surrounding mountainous area during all months of the study period. This spatial pattern reflects the high BVOC emissions from coniferous forests in the mountainous regions compared to the cropland in the Central Valley (Misztal et al., 2014).


Past studies have found that wildfire smoke plumes mixing with high urban NO$_x$ emissions can lead to enhanced urban O$_3$ concentrations (Jaffe and Wigder, 2012). The effects of wildfires during August – September 2020, can be observed in Figure 9 as zones of reduced HCHO/NO$_2$ immediately around the active burn areas followed by a larger "halo" zone of increased HCHO/NO$_2$ as the VOCs emitted from wildfires have time to react to form HCHO. This "halo" pattern is most obvious in

October 2020, when the seasonal cycle of biogenic emissions declined sufficiently to shift the O$_3$ sensitivity back to the VOC-limited regime for the majority of the state except for the region surrounding a wildfire in the Sierra Nevada mountain range east of Fresno (near Yosemite National Park).  VOCs emitted from the wildfire in October 2020, reacted to produce HCHO in the "halo" region, keeping the HCHO/NO$_2$ ratio in the NO$_x$-limited regime. The extensive wildfires that occurred in 2020 appear to have extended the natural peak of the HCHO/NO$_2$ ratio from July into August, September, and even October 2020.

It is unknown whether this satellite observation accurately represents conditions at ground level.  The results at the Sacramento monitoring site in September 2020 (Figure 3a and b) suggest that elevated smoke plumes can dominate the satellite observations, but they may not accurately represent conditions at ground level.

The seasonal variation of O$_3$ sensitivity can be observed over the entire state of California using the TROPOMI HCHO/NO$_2$

(Table S1). Figure 10a shows how the O$_3$ sensitivity seasonal pattern differs among different air basins. The air basins with the highest populations have suppressed seasonal variation of O$_3$ sensitivity because of the higher anthropogenic NO$_x$ emissions. The SoCAB had the lowest HCHO/NO$_2$ ratio among all the air basins in California during the study period.  This is noteworthy since the SoCAB has the highest population and the highest O$_3$ concentrations. The San Francisco Bay Area and San Diego County, two other heavily populated areas in California, also have relatively low HCHO/NO$_2$ ratios compared to

other air basins. Using HCHO/NO$_2$ = 4.6 as the transition point, even these highly urbanized air basins appear to transition from VOC-limited to NO$_x$-limited O$_3$ formation chemistry in summer 2020. It is noteworthy, however, that the urban cores of these regions remain VOC-limited across all months due to very high NO$_x$ emissions (see persistently green regions in Figure 9).  Figure 10b illustrates the TROPOMI HCHO/NO$_2$ monthly variation for different cities in SoCAB between February to

October, 2020. The cities inside/around the LA urban core have HCHO/NO$_2$ < 4.6 throughout the entire year with a weak

seasonal variation. This might be caused by reduced BVOC emissions in the urban center. The remote areas (darker colors in Figure 10b) have greater seasonal variation and higher peak HCHO/NO$_2$. The sharp increase of HCHO/NO$_2$ in summer leads to a shift in O$_3$ sensitivity from the NO$_x$-saturated regime to the NO$_x$-limited regime in the cities further away from the urban core. Due to the different seasonal variation of HCHO/NO$_2$ at different sites, the NO$_x$-saturated region around the urban core will shrink in the summer and expand in the winter. Figure S10 shows this seasonal pattern of O$_3$ sensitivity regime distribution

in Los Angeles as an example. Thus, the optimal emissions control strategy for the entire air basin may differ from the optimal emissions control strategy for urban cores areas.

## 4. Discussion

### 4.1 Sensitivity analysis

The chamber measurements made in the current study capture the sensitivity of the O$_3$ chemical production term in response

to the concentration of NO$_x$ and VOC. The experiment does not directly account for atmospheric processes such as mixing and deposition, but the chemical production term is the dominant processes that determine how local emissions affect local O$_3$ concentrations. The good agreement between ground-based chamber meaurements and satellite O$_3$ sensitivity measurements in the current study builds confidence in the reported seasonal trend of O$_3$ sensitivity. Limitations and uncertainties associated with the temperature, UV intensity, and NOx/VOC perturbations used in the ground-based chamber measurements are

discussed in this section to build further confidence in the results. The potential of these issues to influence the results is analyzed through a combination of measurements and model calculations. The configuration of the chamber model is described in Section 2.4. Figure 11 shows that the chamber model can accurately predict the measured seasonal trends in O$_3$ sensitivity, providing a solid foundation for sensitivity tests.

### 4.1.1 Temperature

The temperature in the reaction chambers was higher than the ambient temperature due to the heating effects of the UV lights. Figure S11 shows that the difference between the chamber gas temperature and the ambient temperature increased by 5-10$^o$C over the course of each experiment, with the exact temperature profile depending on the measurement month. Despite this temperature increase, all three chambers experience the same temperature profile, and so the comparison of O$_3$ formation between the chambers is not strongly biased by this issue. Figure 12a shows the calculated $\Delta O_3^{+NO_x}$ during each month of the

experiment under the chamber and ambient temperature profiles. The difference between the chamber and ambient temperature has little effect on the O$_3$ sensitivity in each month. Temperature effects do not significantly modify the seasonal variation of the measured O$_3$ sensitivity in the current study. Similar behavior was shown in $\Delta O_3^{+VOC}$ in Figure S12a.

### 4.1.2 UV intensity

The UV intensity in the chambers was intentionally maintained at a constant level through all seasons so that the changes in O$_3$ sensitivity could be directly attributed to the changes in the ambient concentrations. A representative average UV intensity was selected for this purpose. As was the case with temperature, all chambers experience the same UV conditions and so this factor is not expected to overly bias the comparison between chambers that acts as the core of the current study. The actual seasonal cycle of UV radiation would generate higher photolysis rates in the summer and lower photolysis rates in the winter that would further amplify the seasonal signal already detected by the measurements with constant UV intensity. SAPRC11 chamber model simulations were used to quantify the effect of seasonal variations in UV intensity. Simulations were carried out using the measured constant UV radiation in the chambers and using the clear sky UV intensity calculated with the routines in the UCD/CIT CTM based on the lat/lon of the measurement site and the day of year. The calculations summarized in Figure 12b show that the difference associated with the use of constant UV radiation does not change the seasonal pattern of O$_3$ sensitivity to NO$_x$ perturbations. The seasonal changes to UV intensity slightly amplify the magnitude of the seasonal trend in O$_3$ sensitivity (increase the absolute value of $\Delta O_3^{+NO_x}$), but the overall seasonal pattern is unchanged. Similar behavior was shown in $\Delta O_3^{+VOC}$ in Figure S12b.

### 4.1.3 Perturbation size

The constant 8 ppb NO$_2$ perturbations used in the current study are greater than or equal to ambient NO$_x$ concentrations during the summer season at Sacramento. O$_3$ formation chemistry is non-linear, meaning that the size of the perturbation may complicate the interpretation of the sensitivity results. O$_3$ sensitivity measurements were conducted using NO$_x$ perturbations ranging from 1-10 ppb at the UC Davis campus from December 2021 to January 2022 to investigate the non-linear behavior of the O$_3$ formation chemistry. The results summarized in Figure S13 show the O$_3$ response expressed as $\Delta O_3$ (final O$_3$ concentration in base case chamber minus final O$_3$ concentration in NO$_x$ perturbed chamber). The $\Delta O_3$ is negative in all NO$_x$ perturbed tests due to the low VOC emission in winter in Davis, CA (similar to Sacramento). Increasing the magnitude of the NO$_x$ perturbation increased the absolute magnitude of the $\Delta O_3$ value but did not shift the chemistry into a different regime.

The size of the NO$_x$ and VOC perturbation used in the chamber experiments is most important when ambient conditions are close to the ridgeline on the O$_3$ isopleth diagram (spring and fall in the current experiment). An 8 ppb NO$_2$ perturbation may jump over the ridgeline in this case, suggesting that the chemistry is NO$_x$-rich rather than NO$_x$-limited. SAPRC11 chamber model simulations were used to quantify the effect of the 8 ppb NO$_2$ perturbation vs. a smaller 2 ppb NO$_2$ perturbation. As shown in Figure 12c, the 8 ppb NO$_2$ perturbations used in the current study do not affect the shape of the seasonal trend in O$_3$ sensitivity measurement, but the 8 ppb NO$_2$ perturbation (open box in Figure 12c) does affect the transition months when the atmospheric system changes to NO$_x$-limited behavior. This issue may influence the estimated value of HCHO/NO$_2$ that characterizes the transition to NO$_x$-limited behiavior in Section 3.2, but it should be noted that the value of 4.6 derived in the

current study is in good agreement with the value of 4.5 reported by Jin (2020). The non-linearities associated with VOC chemistry are less severe and so the size of the VOC perturbation does not complicate the interpretation of results (see $\Delta O_3^{+VOC}$ in Figure S12c).

### 4.1.4 Combined effects of temperature, UV intensity, and perturbation size

The $O_3$ sensitivity calculated with the combined effects of temperature, UV intensity, and lower perturbation size was compared to the basecase calculated $O_3$ sensitivity in Figure 12d and Figure S12d. The $NO_2$ perturbation size has the largest effect on the chamber $O_3$ sensitvity results, with relatively minor changes introduced by temperature and UV intensity. None of these issues changes the basic pattern of increasing $NO_x$-limitations during summer months transitioning to VOC limitations during winter months. It should be noted that operation of the mobile smog chamber system in cities with higher ambient $NO_x$ concentrations is expected to give $O_3$ sensitivity results that are even less dependent on the $NO_2$ perturbation size.

### 4.2 $O_3$ control strategies in California

Current California's $O_3$ control strategies mainly focus on $NO_x$ emissions from motor vehicles (William and Burke, 2016) and especially heavy-duty trucks (Burke, 2020). Additional control strategies would require cleaner engines and zero / near-zero emission technologies (Brown, 2018; South Coast AQMD, 2021). VOC sources that dominate $O_3$ formation are still not clear due to the large numbers of activities that release VOCs and the complex reactions that VOCs undergo in the atmosphere. Controls on VOC emissions have been more effective than controls on $NO_x$ emissions over the past decades, mainly because of reduced emissions from large stationary sources (Barcikowski et al., 2017). The estimated VOC emissions decreased by a factor of 3 while $NO_x$ emission decreased by a factor of 1.5 between 1980 to 2010 according to the California inventory (Cox et al., 2013; Rasmussen et al., 2013). Long-term ambient measurements in the SoCAB confirm that ambient VOC concentrations decreased at an average rate of 7.5% $yr^{-1}$ , while ambient $NO_x$ concentrations decreased at an average rate of 2.6% $yr^{-1}$ between the years 1980 to 2010 (Pollack et al., 2013b; Warneke et al., 2012). These ambient measuements suggest emissions reductsion by a factor of 10 for VOCs and a factor of 2.2 for NOx. Recent studies have shown that VOCs from consumer products are underestimated in the emission inventory (McDonald et al., 2018). However, the clear seasonal pattern in the measured $O_3$ sensitivity and the corresponding pattern for concentrations of VOC proxies (HCHO and CO*Biogenic) suggests that BVOCs are also important.

The 2016 California State Implementation Plan calls for a 34% reduction in $NO_x$ emissions and a 30% reduction in VOC emissions (California Air Resources Board, 2018), which will increase the VOC/$NO_x$ ratio. This will reduce peak $O_3$ concentrations in most areas across California that become $NO_x$-limited in the middle of the summer. In contrast, the $NO_x$ emissions control program could cause a short-term increase in peak $O_3$ concentrations in the urban cores that are currently VOC-limited and it could increase intermediate $O_3$ concentrations in late spring or early fall as regions transition back to VOC-

limited conditions. These regions do not currently violate the $O_3$ NAAQS, but they could experience future violations depending on the timing of the transition to lower NOx concentrations. Despite these penalties, controls on $NO_x$ emissions may be the only alternative for long-term $O_3$ reductions in regions where VOC emissions are dominated by biogenic sources. As the $NO_x$ keeps decreasing, the $O_3$ photochemical regime will eventually transition back to $NO_x$-limited conditions and all further $NO_x$ reductions will yield decreasing $O_3$ concentrations. Previous studies have observed such a transition between

VOC-limited to $NO_x$-limited conditions in polluted urban areas with high $NO_x$ concentrations. Jin et al. (2020) observed a suppression of the $NO_x$-limited area between 2013 – 2016 vs. 1996 – 2000 in Los Angeles by analysing satellite $HCHO/NO_2$ ratios. Baidar et al. (2015) observed a weakening of the higher $O_3$ concentrations on weekends in the SoCAB between 1996 to 2014, reflecting a transition towards more $NO_x$-limited conditions. These studies suggest that continued reductions in $NO_x$ emissions will eventually yield a transition to fully $NO_x$-limited conditions in Los Angeles, albeit this transition may not be

fully complete for decades.

Wildfires are an unpredictable factor that enhances $O_3$ formation in California. $O_3$ formation during wildfire events shifts towards more $NO_x$-limited conditions, making reductions in $NO_x$ emissions attractive. The frequency and scale of wildfires in the western U.S. have increased over time due to the effects of drought and climate change (U.S. Global Change Research

Program, 2018). Abatement strategies may focus on wildfire prevention as an effective way to reduce incidental $O_3$ concentrations.

**5. Conclusion**

Direct measurements of $O_3$ sensitivity to precursor $NO_x$ and VOC concentrations using a mobile smog chamber system in Sacramento, CA from April to December, 2020 show that $O_3$ sensitivity follows a seasonal cycle. $O_3$ formation is VOC-

limited in the spring, $NO_x$-limited in the summer, and returns to VOC-limited in fall – winter. This seasonal pattern reflects higher emissions of reactive VOCs during the summer season and increased $NO_x$ concentrations during the other seasons. The most obvious potential source of increased VOC emissions during the summer season is biogenics. Comparing the ground-based chamber measurements to satellite measurements from TROPOMI suggests that the transition between $NO_x$-limited and VOC-limited chemical regimes for $O_3$ formation occurs at a TROPOMI $HCHO/NO_2$ ratio of 4.6. Monthly-averaged

TROPOMI measurements show that $O_3$ sensitivity across most of California follows a seasonal cycle similar to Sacramento, but locations with higher population density are more VOC-limited. The urban cores of most large cities remain VOC-limited in all seasons even when the surrounding areas become $NO_x$-limited in the middle of summer. The variability of the chemical regime for $O_3$ formation across space and time makes it difficult to design an emissions control strategy that will equitably reduce $O_3$ concentrations for all California residents currently living in air basins that violate the 8-hour $O_3$ NAAQS.

Reductions in $NO_x$ emissions will be the most efficient control strategy to reduce present-day peak $O_3$ concentrations, but this strategy will lead to increasing $O_3$ concentrations in urban cores during the middle of summer and increasing $O_3$ concentrations

in surrounding regions during late spring and early fall. These penalties will persist until $NO_x$ emissions are reduced sufficiently to push the entire region into $NO_x$-limited conditions sometime in the coming decades. VOC emissions reductions never cause increasing $O_3$ concentrations and $O_3$ formation is VOC-limited during some seasons. It may be advisable to augment $NO_x$ emissions control programs with some amount of controls on volatile consumer products (VCPs) and mitigation of wildfires in an attempt to reduce any near-term increases in $O_3$ concentrations. Continued deep $NO_x$ emissions reductions should eventually transition all locations across California into the $NO_x$-limited regime, and will effectively push the state toward 8-hour $O_3$ NAAQS attainment

**Acknowledgment:**

The authors thank Michael Miguel, Anthony Esparza, and Aimee Davis of the California Air Resources Board (CARB) for their logistical support surrounding the siting of the chamber experiments. Part of this study was supported by CARB Agreement No. 19RD012 and Coordinated Research Council (CRC) Agreement No. A-121. The views expressed in this article are those of the authors and do not represent the views or policies of CARB or the CRC.

**Data Availability:**

Monthly chamber measurement data will be archived on a public data site when the manuscript is accepted for final publication.

**Author Contribution:**

SW made field measurements and wrote the initial draft of each version of the manuscript. HJL processed TROPOMI data. AR analyzed wildfire vs. no-wildfire periods. SL provided project management. TK constructed the initial version of the chambers. JHS hosted initial measurements and helped revise manuscript. MJK designed the experiment, directed data analysis, coded the chamber model, and revised the manuscript.

**Competing Interests:**

The authors declare that they have no competing interests.

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

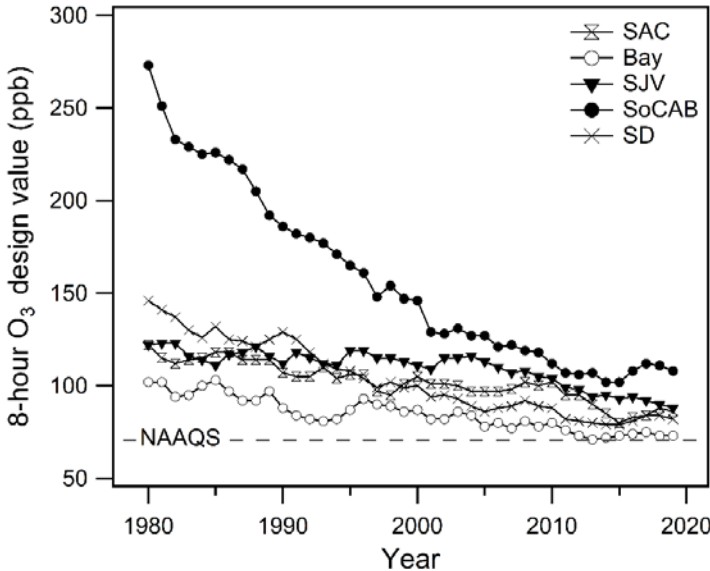

**Figure 1. 8-hour O$_3$ design value in 5 air basins in California from 1980 to 2019. Dash line is the 2015 8-hr O$_3$ NAAQS (= 70 ppb). 5 air basins include Sacramento Valley (SAC), San Francisco Bay area (Bay), San Joaquin Valley (SJV), South Coast Air Basin (SoCAB), San Diego County (SD). Data collected from California Air Resources Board (https://www.arb.ca.gov/adam).**

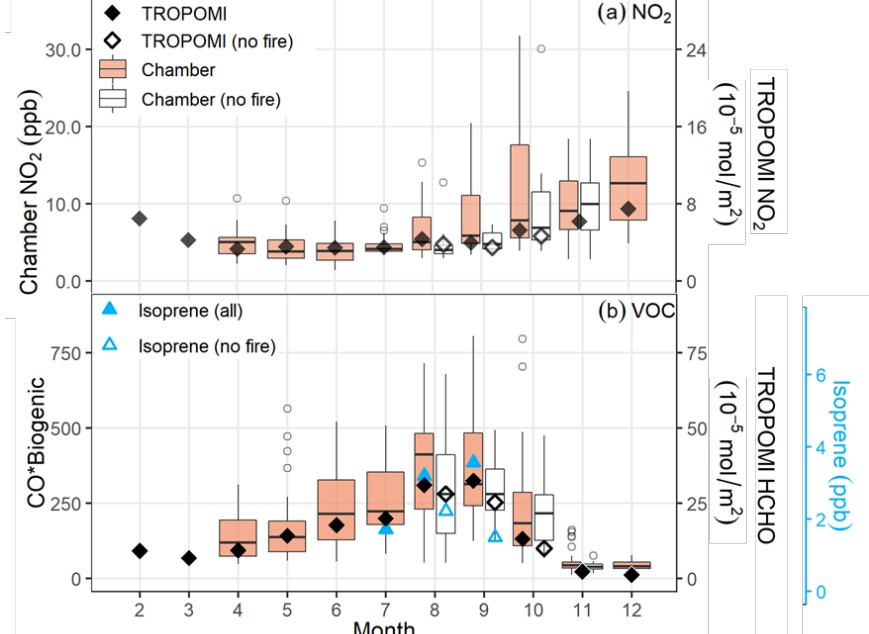

**Figure 2. Monthly concentrations of NO₂ (panels a) and CO*Biogenic/HCHO/Isoprene (panels b) from February to December 2020. Ground-based chamber measurements use the left axis with results shown as box and whisker plots. TROPOMI measurements use the right axis and are shown as diamonds. Isoprene from ground monitoring station shown as blue triangles. The open box and points show the results after removing wildfire days.**

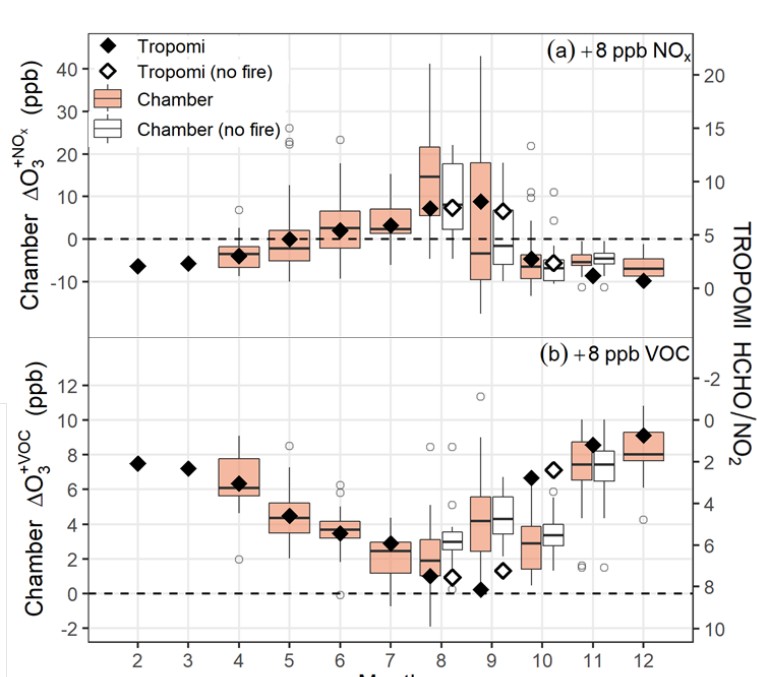

**Figure 3. Monthly variation of TROPOMI HCHO/NO₂ (diamond) and ΔO₃ (box) due to NOₓ addition ($\Delta O_3^{+NO_x}$) and VOC addition ($\Delta O_3^{+VOC}$) from April to December including wildfire days (top) and without wildfire days (bottom).**


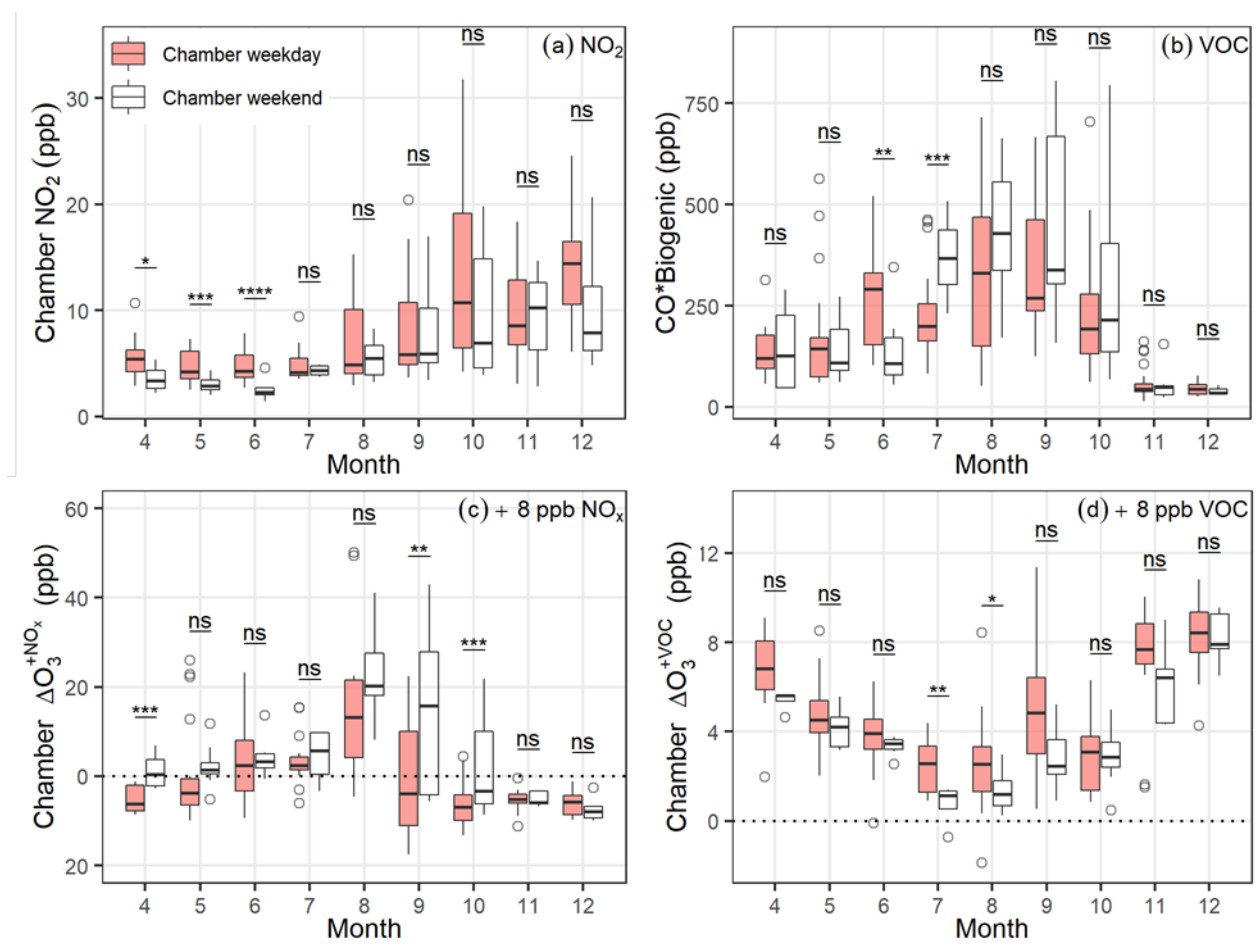

**Figure 4 .** Weekday (solid box) and weekend (open box) monthly-average concentrations of NO₂ and CO*biogenic (panels a, b), and $\Delta O_3^{+NO_x}$ and $\Delta O_3^{+VOC}$ (panels c, d) from April to December, 2020 after removing wildfire days. The stars above each box and whisker plot represent the significance of the weekday vs weekend difference. (∗: p value < 0.1, ∗∗: p value < 0.05, ∗∗∗: p value < 0.01, ∗∗∗∗: p value < 0.001, ns (not significant): p value >= 0.1)


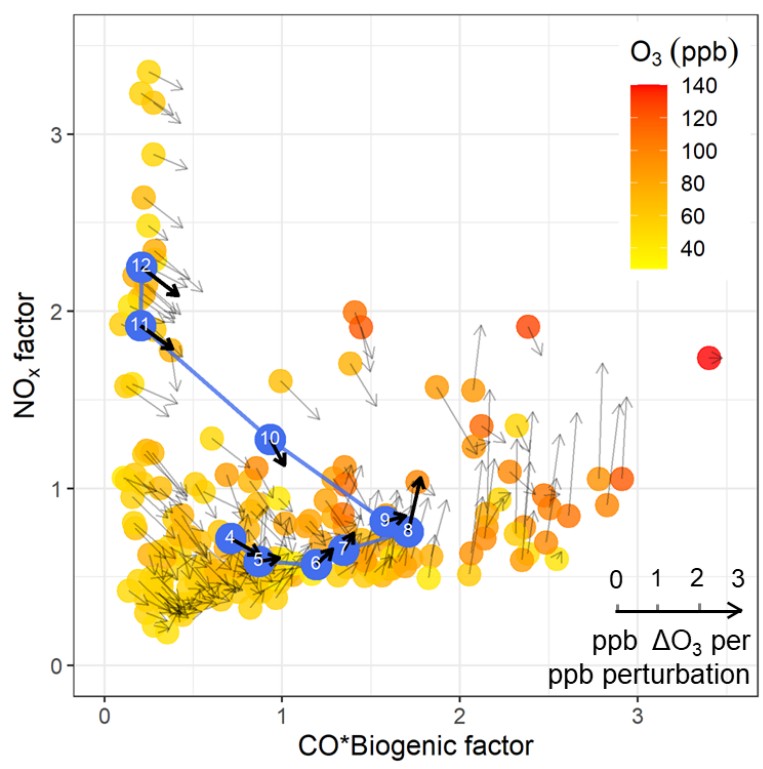

**Figure 5. Measured O₃ isopleth diagram. The NO$_x$ and CO*Biogenic factor is calculated by the daily value divided by averaged value. The O₃ concentration is the daily O₃ concentration in the basecase chamber after 3 hours UV exposure. Arrows represent the O₃ sensitivity. The blue dots are the monthly averaged values, the blue line shows the seasonal cycle in the O₃ isopleth diagram. Days influenced by wildfires are removed from the plot.**


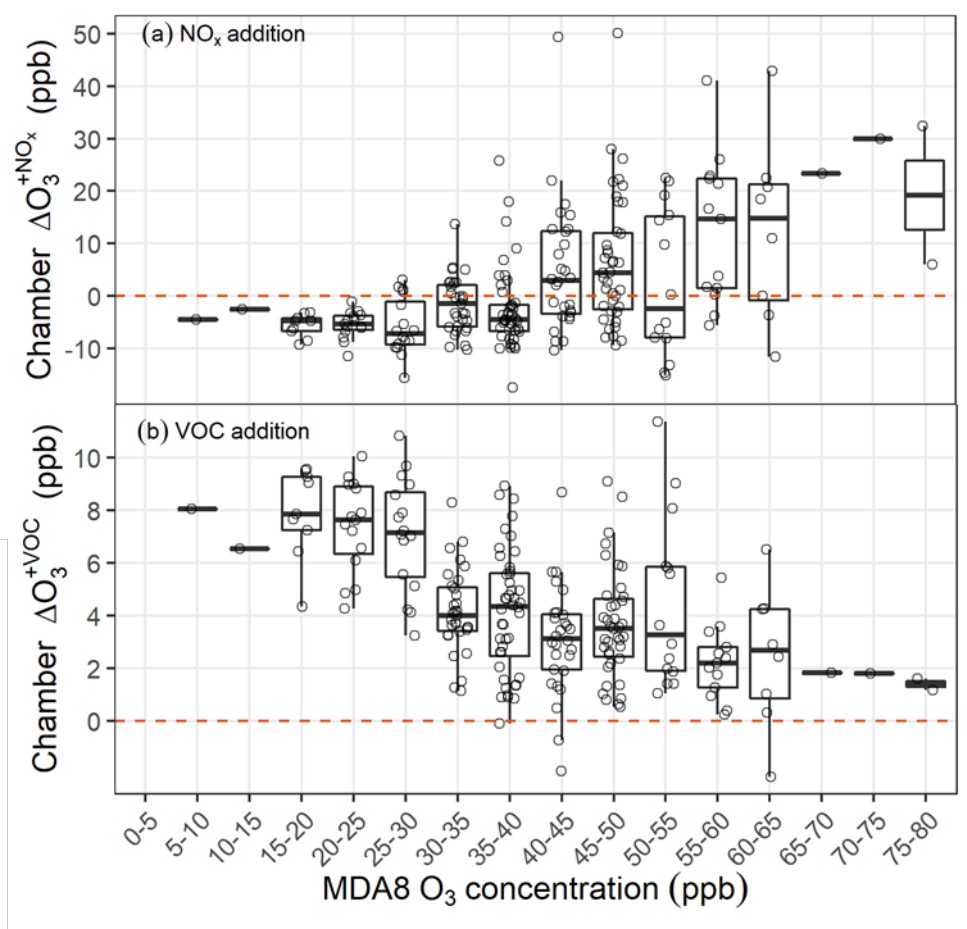

**Figure 6. Boxplot of O₃ sensitivity to NOₓ and VOC as a function of MDA8 O₃ concentration. Points indicate the data point in each range of MDA8 O₃ concentration.**

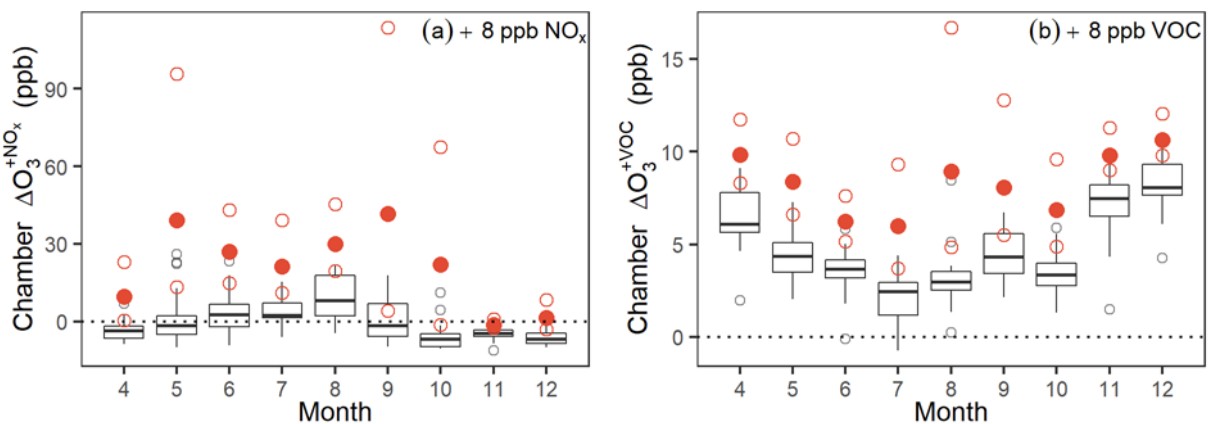

Figure 7. 3-year return level (red dot) and 95% confidence interval (red open dot) from extreme value analysis of O₃ sensitivity to NOₓ and VOC.

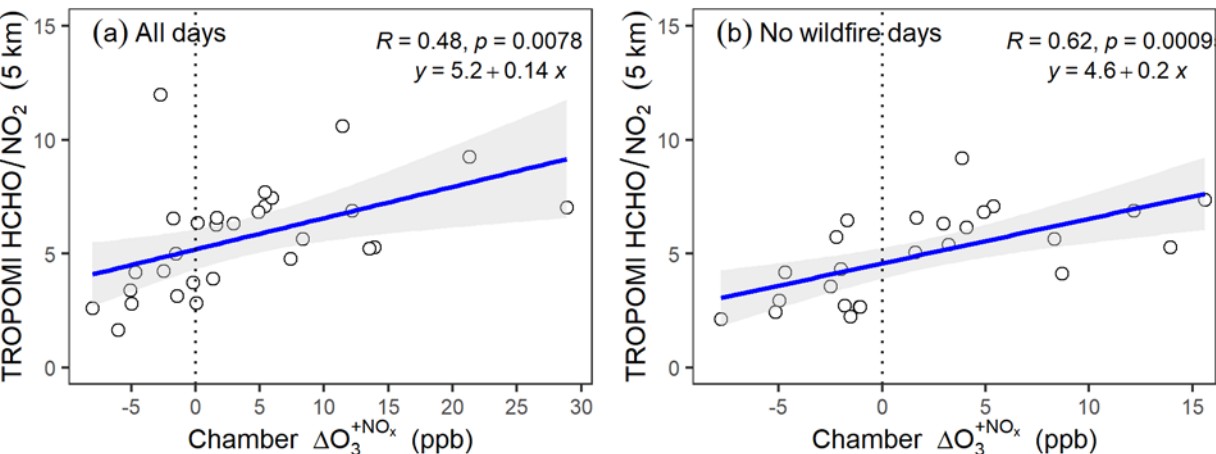

Figure 8. Ordinary least squares (OLS) regression between weekly averaged TROPOMI HCHO/NO₂ at 5 km circular buffers and the weekly averaged $\Delta O_3^{+NO_x}$ from ground-based measurement. The shaded area shows the 95% confidence interval of the mean response of the predicted value.

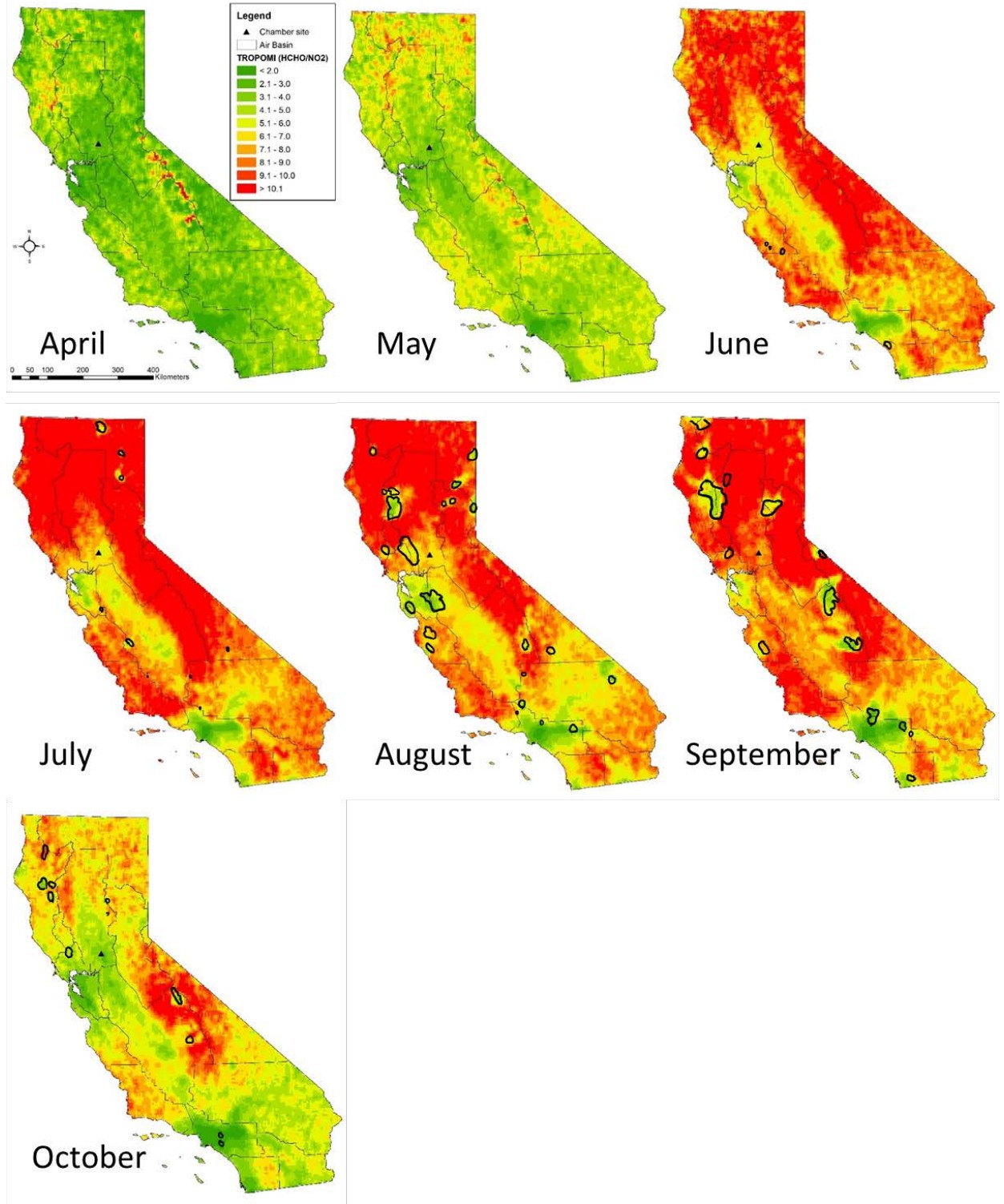

**Figure 9. Spatial distribution of TROPOMI satellite (HCHO/NO₂) ratios in California for April – October 2020. TROPOMI NO₂ and HCHO data are re-gridded to 5 km resolution when calculating monthly-average ratios. The black bold line circles the burned area in each month detected by MODIS from Fire Information for Resource Management System (FIRMS). The NOx-limited conditions correspond to HCHO/NO₂ ratios above 4.6.**

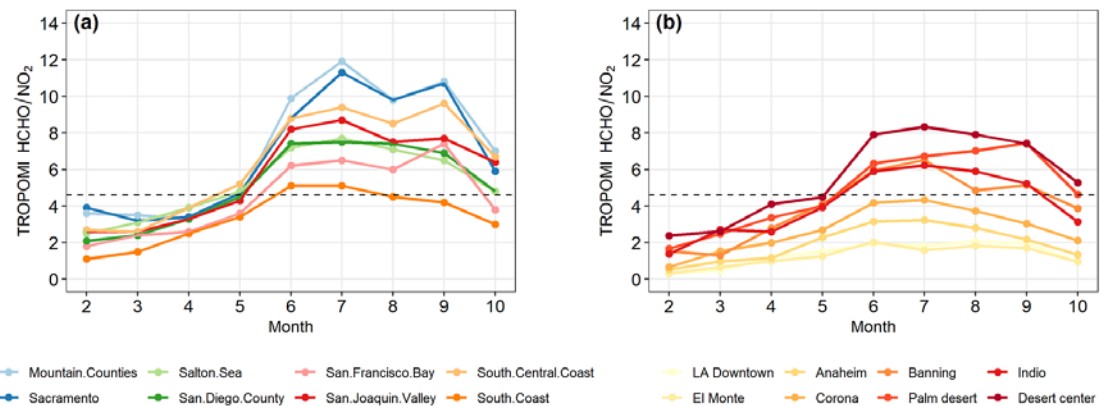

**Figure 10. Monthly variation of TROPOMI HCHO/NO₂ in different air basins (left panel) and in different cities in Southern California (right panel). The darker colors in the right panel indicate increasing distance from the urban center of Los Angeles.**

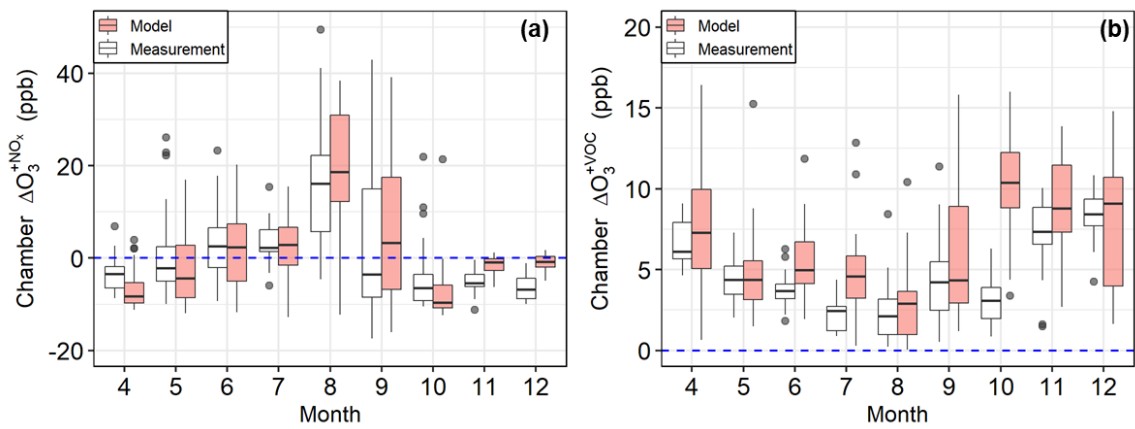

**Figure 11. Monthly variation of the $\Delta O_3^{+NO_x}$ (a) and $\Delta O_3^{+NO_x}$ (b) predicted by the chamber model (solid box) and directly measured in the chamber (open box) from April to December, 2020 at the Sacramento measurement site.**

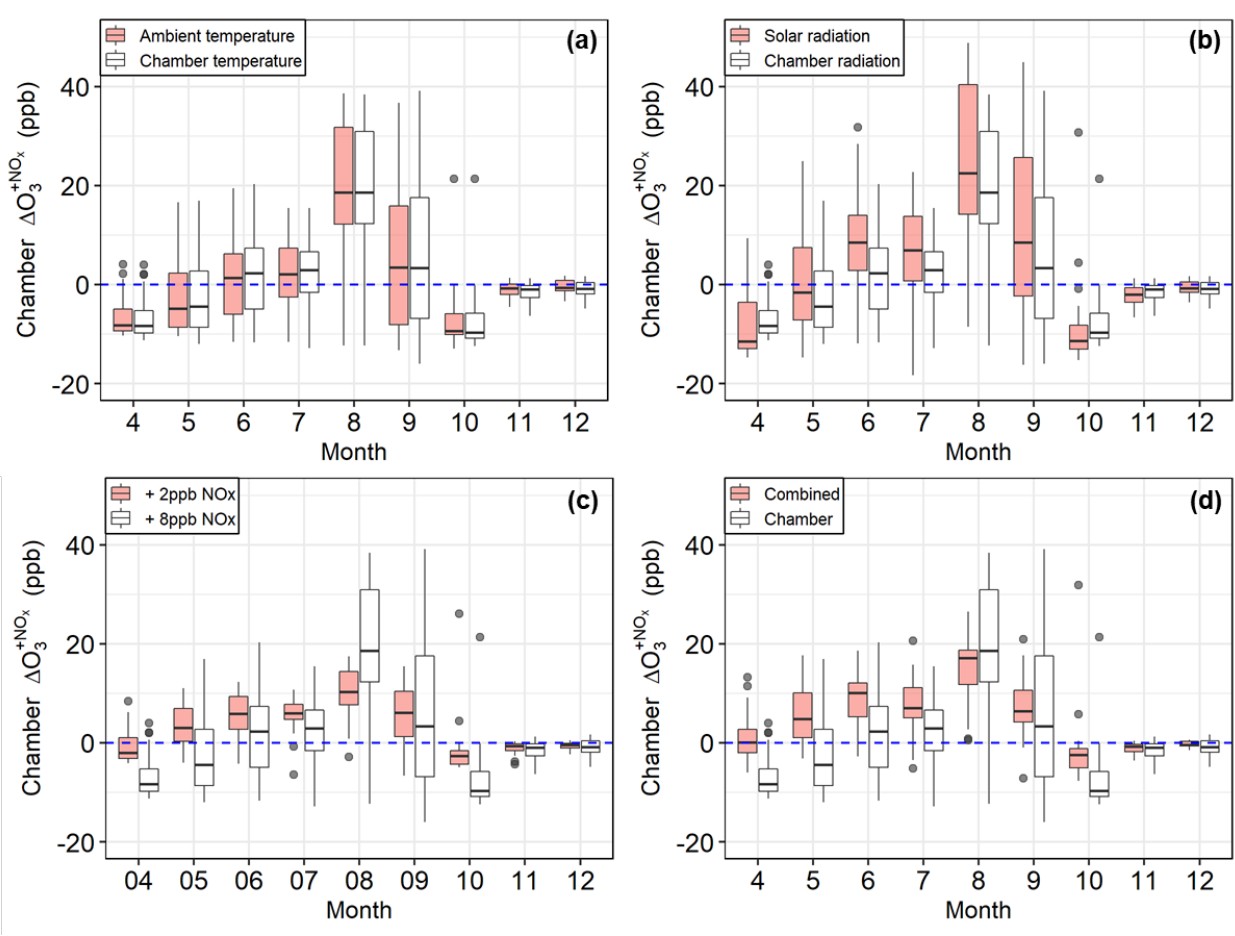

**Figure 12. Effect of temperature, radiation, and perturbation amount on the monthly variation of the predicted chamber $\Delta O_3^{+NO_x}$** from April to December, 2020 at the Sacramento measurement site. Open boxes show the predicted response under chamber measurement conditions (chamber temperature, radiation, and 8ppb $NO_2$ perturbation). Solid boxes show the predicted response when conditions are updated: (a) using the ambient temperature profile; (b) using clear-sky solar radiation; (c) using a 2 ppb NOx perturbation; (d) using the combination of ambient temperature, solar radiation, and 2ppb $NO_2$ perturbation.