# Peer review of "Direct Measurements of Ozone Response to Emissions Perturbations in California"

_Atmospheric Chemistry and Physics, 2021_

## Referee Comment (RC1)

Review of "Direct Measurements of Ozone Response to Emissions Perturbations in California" by Shenglun Wu et al.

MS Number: acp-2021-708

**Summary:**

This paper aims to characterize the overall ozone response to NOx and VOC emission reductions in California by synthesizing satellite data with the results of experiments that perturbed the ambient photochemistry through in-the-field smog chamber runs. This is an interesting study that I think possibly can be developed into a useful addition to our understanding of ozone formation in California. However, there are two major shortcomings in the description and interpretation of the smog chamber experiments that prevent a full understanding of the experimental results and significantly impact the conclusions of the paper. These shortcomings are significant enough that I recommend that the paper be rejected until the authors are able to thoroughly address the issues discussed below.

It should be noted that my expertise is in laboratory and field measurements, and that I have little expertise in the interpretation of satellite data. The following comments focus on the smog chamber work; in my view this paper should not be accepted until it is reviewed by someone that does have the necessary experience to thoroughly review the satellite data interpretation.

Given my major concerns regarding the smog chamber approach, I am unable to thoroughly review the Results, Discussion and Conclusions of the paper. Such a review must await resubmission of a manuscript that improves the experimental discussion.

**Major Issues:**

- The first line of the abstract states: "A new technique was used to directly measure O3 response to changes in precursor NOx and VOC concentrations in the atmosphere." However, neither this paper nor any of the references demonstrate the validity of this statement. In this regard, I see three major problems:
  - a. First, the smog chamber system is inadequately described, and many questions remain in my mind regarding its performance. An adequate description could possibly be added to this manuscript, probably in the supplement, but the authors should consider publishing a stand-alone paper in a journal such as *Atmospheric Measurement Techniques* (AMT), before attempting to develop this paper describing the results. Issues that should be included in that description are:
    - Temperature control of the chambers It has been widely reported (e.g., Coates et al., 2016) that temperature affects ozone formation. Are the chambers held at ambient temperature in spite of irradiation by the UV lamp panels? (I presume that the chambers are enclosed in a light-tight structure to exclude ambient sunlight this should be fully described).
    - Light intensity control It is generally recognized that ozone formation is a function of solar radiation intensity. In the ambient atmosphere, this intensity varies with solar zenith angle on diurnal and seasonal cycles, as well as ambient clouds and meteorological conditions (clouds, aerosol loading, stratospheric ozone column, etc.) Is there a mechanism to allow the chamber light intensity to mimic the ambient light

intensity? Most important I suspect is the seasonal cycle of solar radiation intensity. Figure 3 of the paper shows the seasonal cycle of the smog-chamber results; does the experiment mimic the seasonal variation of the solar radiation intensity? If not, how can this "technique … directly measure O3 response to changes in precursor NOx and VOC concentrations in the atmosphere"?

- Blank tests To develop confidence in the reported results, the authors must show that results of "blank runs" (i.e., filling all three chambers with zero air, adding the standard perturbation amounts of NOx and VOC to the two perturbed chambers, and irradiating for the standard three hours) result in zero ozone formation in all three chambers.
- Ambient condition tests Again, to develop confidence that the reported results actually "directly measure O3 response to changes in precursor NOx and VOC concentrations in the atmosphere" it would seem critical to remove any light-tight shroud around the chambers so that they are exposed to ambient solar radiation, to not
- operate the UV lamps, and then to compare the ozone evolution in the chamber with the evolution of ambient ozone. Only if the chamber ozone actually tracks the ambient ozone, can it be accepted that a direct measurement of the ozone response is actually obtained.
- Linearity tests With regard to comment c below, the response of the smog-chamber system to different magnitude perturbation concentrations must be investigated. The figure at right shows example diurnal cycles of the NOx concentrations in four months, one from each season, measured at the

monitoring site adjacent to the smog-chamber location (Fig. S5). In summer and spring (the seasons of most policy relevance) the added  $NO_2$  perturbation (8 ppb) in the smog chamber more than doubles the  $NO_X$  concentration. Thus, the physical significance of the derived ozone formation sensitivity is questionable.

b. Second, I do not believe that the system can actually directly measure ozone sensitivity in the sense that it accurately reflects how actual ambient ozone concentrations would respond to precursor emission changes. In the real atmosphere, during the photochemical active period of a day, ambient air parcels are transported through an air basin. During that transport, dilution and mixing processes occur, fresh emissions are injected into the air parcel and ozone is lost to surface deposition simultaneously with in situ photochemical ozone production. It seems to me that the smog chamber can only reproduce one (albeit very important) aspect of this extremely complex ambient ozone production process. A late morning, integrated air parcel is captured in the chamber, and then the in situ photochemical ozone production is mimicked in isolation from all other processes. This issue should be thoroughly discussed, and the authors should acknowledge that their approach can potentially determine the sensitivity of the in situ

photochemical ozone production to precursor NOx and VOC, but likely that does not directly correspond to the sensitivity of the actual ozone concentrations in the ambient Sacramento boundary layer.

- c. Third, when the sensitivity of ozone is discussed, it is generally understood that the sensitivity is referring to the response of ozone to decreases in precursor NOx and VOC. However, the smog chamber experiment operates by investigating increases in those precursors. If ozone production chemistry responded linearly to precursor changes, this distinction would be unimportant; however it is widely acknowledged that ozone chemistry is highly non-linear. Thus, the smog chamber approach must give biased results. For example, if the ambient atmosphere were on the "ridgeline" of the corresponding ozone isopleth diagram, then the smog chamber data would indicate VOC sensitivity, since the ozone production would decrease with added NOx (i.e.,  $\Delta O_3^{+NOx}$ would be negative). But if the experiment could be run with a NOx decrease, rather than an increase, the ozone produced would again decrease, indicating NOx sensitivity. The extent of the bias resulting from the non-linearity of the ozone response depends upon the relative magnitude of the precursor perturbations. A very small, potentially infinitesimal, perturbation would reduce, potentially eliminate, the bias; however to obtain a precisely measurable response, I suspect that the precursor perturbations were rather large relative to the ambient concentrations. Since the NOx concentrations were actually measured in these experimental runs, a thorough discussion of this potential source of bias should be given in the context of the magnitude of the NOx perturbations relative to the initial ambient NOx concentrations in the chamber when the experimental run is initiated.
- 2. This paper emphasizes the policy relevance of the results. The last two sentences of the abstract state:

"This challenging situation suggests that emissions control programs that focus on NOx reductions will immediately lower peak O3 concentrations, but slightly increase intermediate O3 concentrations until NOx levels fall far enough to re-enter the NOx-limited regime. The spatial pattern of increasing and decreasing O3 concentrations in response to a NOx emissions control strategy should be carefully mapped in order to fully understand the public health implications."

However, the smog chamber work is analyzed from the perspective of the final ozone concentration in the chamber at the end of the experimental run. The policy relevance would be much more clearly evident in this work if the analysis perspective focused on the ambient MDA8 ozone concentration on the day of each run. In particular, Figure 6 would be more informative if the x-axis variable were the MDA8 ambient ozone concentration recorded at the monitoring site adjacent to the smog chamber field location (see Figure S5). A great deal more support must be given before these policy-relevant statements can be accepted. In this regard, the findings must be directly related to the conditions that produce ambient MDA8 ozone concentrations that exceed the NAAQS, as discussed in the 2nd paragraph of the Introduction Section of the paper.

**Minor Issues:**

1. Lines 54-61: These sentences discuss references that propose causes of the increase of  $O_3$  design values in some air basins between the years 2015 - 2018. However, many of the cited references were published before that increase occurred, so they obviously do not directly address that increase. This discussion must be improved with the inclusion of appropriate

references. To my knowledge tenable proposed causes include the influences of wildfire emissions and particularly pronounced heat waves; however the causes the authors discuss in these lines really are not tenable. For example, "growing importance of precursor VOC emissions not previously accounted for in the planning process" could possibly account for a slowing of the ozone decrease, but (unless those emissions increased markedly over that short 2015 - 2018 period) could not account for an increase. Similarly, climate has not changed markedly over that short 2015 - 2018 period, so this cause also is not tenable. If the authors wish to discuss this rather minor feature of Figure 1 (i.e., there are other wiggles in the trend of similar magnitude), then they should do so in a rigorous manner. Perhaps a cause could be sought that accounts for the increase in some (e.g. SoCAB and San Diego as discussed in the manuscript), but not in other Southern California air basins (e.g., South Central Coast Air Basin, which is adjacent to SoCAB).

- 2. Lines 143-153: This paragraph is not persuasive. The statement "The initial O3 concentration in the basecase chamber was similar to the ambient O3 concentration, indicating that the gas-phase chemical composition related to O3 formation is not modified during chamber injection" requires more discussion. Figure S3 clearly indicates that the initial O3 concentration in the basecase chamber was always significantly (10-30 ppb) below the ambient concentrations at the initial time. This issue and its impacts on the entire analysis must be thoroughly discussed. I suggest that this discussion include an expanded time scale for some specific examples so comparison between the basecase chamber and ambient air is much more clearly illustrated. Further, the tests included in Fig. S3 were conducted in Los Angeles (an urban area with very different ozone levels and presumably photochemical environment) than Sacramento, where the primary field work was conducted (e.g., see Figure 1 of the paper).
- 3. Lines 143-153: This same paragraph discusses the comparison of the O3 increase in the basecase chamber and in the ambient air; that discussion is greatly oversimplified. In the ambient atmosphere, the early morning O3 increase is largely driven by mixing down of ozone rich air from aloft as the boundary layer rapidly grows during that period. In the SoCAB, the land-sea breeze circulation affects the diurnal ozone cycle during the day. The statement "The O3 formation in the chamber, therefore, captures a realistic "worst-case scenario" for surface-level O3 formation under conditions where atmospheric mixing cannot dilute the NOx and VOC concentrations that build up in the nocturnal ground-level stagnation layer." is simply not justified the conditions inside the chambers are very different from ambient conditions. These differences must be thoroughly discussed not simply "hand waved" away. It should be realized that the predominant growth of the convective boundary layer generally approaches its maximum extent by noon, which is the time that the experimental run begins (e.g., see Figures 4-7 of Bianco et al., 2011).
- 4. Many of the figures show linear regression fits. However, It appears that there may be shortcomings and errors in some of them. These issues should be checked and corrected if necessary; specifically:
  - a. Figure S2. Confidence limits (preferably 2 sigma or 95%) for the slopes should be included to indicate that the slopes are indeed consistent with unity.
  - b. Figure S4. Given the large scatter in the data points and the small correlation coefficients, the exceedingly small p values, and the relatively small shaded areas appear to me to not

be realistic (and the meaning of the shaded areas should be defined.) Please check all such fits in all figures to be sure the fitting is properly calculated.

- 5. The discussion of the VOC and CO relationships (lines 155-173) requires improvement.
  - a. The statement "Biogenic sources do not emit CO and so any correlation between biogenic VOCs and CO purely reflects the utility of CO as an indicator of atmospheric mixing that equally affects all sources" is incorrect and misleading. An important source of CO is partial oxidation of biogenic VOCs, so their correlation is more complex than indicated here. Further, atmospheric mixing does not equally affect all sources, since the result of mixing is dependent on the background concentrations in the diluting air.
  - b. The quantity CO\*Biogenic is not clearly defined. Where were the sites of those VOC measurements? More details of the "temperature and relative humidity-induced enhancement factor for isoprene emissions" must be given. The cited reference is now 30 years old; in the intervening 3 decades a great deal has been learned about biogenic VOC emissions. Is this "enhancement factor" consistent with current understanding?
  - c. Given the quoted R value in Fig. S4, it should be mentioned that use of CO\*Biogenic as an approximate surrogate for VOCR only captures ~36% of the variance of VOCR at the site where the CO and VOCR measurements were made.
  - d. It should be explicitly stated whether the CO and VOC measurements were made at the same monitoring site, and the location of this site relative to the location of the chamber measurements should be discussed.
- 6. Section 2.3. The brief experimental description is not adequate. Questions that occur to me include: How can air be sampled from the chambers without disturbing the environment? Do the sides of the chamber gradually collapse? If the sides collapse, what fraction of the air is exhausted through the sampling process over the 210-minute experimental run? Why is a linear extrapolation required? Section 2.1 reports that ozone loss rates were 5%/hour in the chambers; was correction made for this loss rate? Figure S2 indicates that the perturbed chambers gave 1 to 2% greater ozone production than the base chamber; was correction made for this difference? How many experimental runs were made over the 11 month period of Figure 2, and included in the box and whisker plots? In addition to an expanded experimental discussion that answers these questions, I suggest including sample chamber measurement data from a typical experimental run as a section in the Supplement. That section should clearly describe all steps included in the process of deriving the  $\Delta O_3$  values of Figure 3 from the 3-hr time series of concentration measurements. It would also be useful to indicate the number of experimental runs included in each box and whisker plot in Figure 3 (and in subsequent figures).
- 7. In this regard, Figure S1 seems to indicate that four lamps were mounted on the floor of the middle chamber, and eight were mounted on the floor of each end chamber. Does this difference in the figure reflect the reality of the chambers? If so, please explain why this arrangement was used, and give more discussion regarding why this arrangement does not bias the results.
- 8. The discussion of Figure 2 is not adequate. Why are there no TROPOMI measurements in November and December? Reading the figure caption seems to indicate that CO measurements were made in the ground-based chambers; however, Section 2.1 seems to indicate that only NOx, NOy, O3, temperature and relative humidity were measured in the chambers. Please explain clearly how the CO\*biogenic values were determined. Evidently

the isoprene concentrations in Figure 2 were measured at an EPA PAMS site; Figure S5 indicates two monitoring sites. It should be indicated which (if either) of those sites reported the isoprene measurements discussed here. The meaning of the lines in the box and whisker plots should be explicitly indicated, here and in later figures.

9. The final paragraph of Section 3.1.1 discusses VCPs, but requires improvement. VCP emissions are not related to either CO emissions or isoprene emissions (except if isoprene is one of the VCPs). Thus, there is no reason to expect seasonal pattern similarity between VCPs and CO\*biogenic values, or between VCPs and isoprene. Nevertheless, there is nothing here to indicate that VCPs are not important (or even dominant) in driving ozone production in Sacramento (although I agree that this is very unlikely). This paragraph should be modified or eliminated.

**Reference:**

- Bianco, L., et al. (2011), Diurnal Evolution and Annual Variability of Boundary-Layer Height and Its Correlation to Other Meteorological Variables in California's Central Valley. Boundary-Layer Meteorology, 140, 491-511, DOI 10.1007/s10546-011-9622-4.
- Coates, J., K.A. Mar1, N. Ojha, and Tim M. Butler (2016), The influence of temperature on ozone production under varying NOx conditions a modelling study, Atmos. Chem. Phys., 16, 11601–11615, doi:10.5194/acp-16-11601-2016.

---

## Referee Comment (RC3)

Review of "Direct Measurements of Ozone Response to Emissions Perturbations in California" by Shenglun Wu et al.

This manuscript investigates the ozone ($O_3$) formation sensitivity to precursor VOC and NOx concentrations and seasonal variations by deploying three smog chambers equipped with lamps that produce constant UV radiation. Satellite retrievals of $HCHO/NO_2$ ratios were used to complement the results of the chamber measurements and support the seasonal changes of BVOCs abundance and $O_3$ formation sensitivity. One of the main motivations of this study is to identify the dominant factors contributing to the recent increased $O_3$ concentrations in some of the air basins in California, after decades of decreasing trends due to the success of implementing emissions control actions. This is an interesting study; however, the manuscript will require more detailed description of the methodology and analysis of the results. Considering the scientific significance and policy relevance of the topic, more work is also needed to substantiate the conclusions regarding the response of $O_3$ to precursor emission control strategies. The following are some specific comments.

1. The authors stated that "A new technique was used to directly measure $O_3$ response to changes in precursor NOx and VOC concentrations...…"; it will be helpful for the readers to have more detailed description of the measurements and the improvements compared to other recent smog chambers studies.

2. The authors used artificial light to provide constant UV radiation in the chamber experiments, which is different from the real atmospheric conditions. Additionally, the settings of other parameters for the smog chambers, such as temperature, relative humidity, etc., are important in modifying the $O_3$ formation but they were not provided in the measurement section. More importantly, it is not reasonable to explore the seasonal changes of the $O_3$ sensitivity using chambers with constant UV radiation. Except for anthropogenic emissions changes, variations of solar radiation play a major role in the seasonal pattern of $O_3$ formation sensitivity. $O_3$ formation regime becomes more $NO_X$-sensitive in warm seasons, which is mainly caused by intensified solar radiation. Increasing solar radiation enhances BVOCs emissions that are light- and temperature-dependent, facilitates photochemical reactions, and promotes development of the planetary boundary layer to decrease near-surface $NO_2$ concentrations.

3. $O_3$ formation sensitivity is investigated only by adding 8 ppb $NO_X$ in chamber #1 and 8 ppb surrogate VOCs in chamber #3. Lacking a series of linear experiments with different concentrations of precursor gases, the current conclusions are drawn from the effects of 8 ppb precursor perturbations on $O_3$ levels of the air masses sampled at one site, which is not sufficient to assess the $O_3$ formation sensitivity *in situ*, let alone the regional $O_3$ sensitivity.

4. $O_3$ production sensitivity is determined by the ratio of $NO_X$ to VOCs. Adding constant 8 ppb $NO_X$ or surrogate VOCs to experimental air masses sampled in different seasons with various precursor concentrations could lead to varying perturbations for the ratio, possibly contributing to the measured seasonal variations in $O_3$ production sensitivity.

5. While smog chamber experiments have been used to simulate the photochemical reactions occurring in the atmosphere, the experiments can not accurately represent the complex real atmospheric conditions. This should be taken into consideration in discussing ozone sensitivity to the precursor gases and in drawing conclusions about emissions control policies.

---

## Author Comment (AC3)

We thank the reviewer for a set of very comprehensive comments. We have used a combination of measurements and model calculations to evaluate how these issues could impact the overall results of the paper. Our detailed responses for each comment are listed below, along with the changes made to the manuscript to make these findings clear to readers. Our responses to the comments are presented in blue. The comments are shown in black.

1. Clearly this research is very relevant to policymakers for the development of emission control strategies to improve air quality. This study is not the first to develop and apply mobile smog chambers to air quality measurements at specific sites. For example, Mobile Smog Chamber, https://www.psi.ch/en/lac/mobile-smog-chamber; Kaltsonoudis et al., A portable dual-smog-chamber system for atmospheric aerosol field studies, Atmos. Meas. Tech., 12, 2733–2743, 2019. I believe that there was even a commercial chamber, about the size of a soccer ball, that was marketed to directly measure NOx and VOC limitation by an Australian company (I apologize for not being able to find a reference). My point is that the authors should include a couple of paragraphs summarizing previous mobile chambers and discuss how their new system is an improvement.

The original manuscript only has 2 sentences talking about the previous study about transportable smog chamber. As requested, we have searched for additional studies that use transportable smog chambers. A more thorough discussion has been added to the Introduction section in the revised manuscript.

"Mobile smog chambers bridge the gap between laboratory studies and the real atmosphere. Past studies have designed mobile smog chambers to measure the aging of secondary pollutants (i.e.,  $O_3$ , SOA) from certain emission source (Howard et al., 2008, 2010; Li et al., 2019; Platt et al., 2013; Presto et al., 2011). It is difficult to evaluate sensitivity of secondary pollutants formed from multiple sources using a single smog chamber. Recently, a mobile dual smog chamber system has been used to directly measure the SOA formation in ambient air (Jorga et al., 2020; Kaltsonoudis et al., 2019). Our smog chamber system consists of three chambers designed to simultaneously analyze the non-linear response of  $O_3$  formation to  $NO_x$  and VOC perturbations. The automated valve and sampling system also allows longterm remote field measurements to evaluate the seasonal trends in  $O_3$  sensitivity."

2. The TROPOMI measurements are especially interesting in that they indicate how important biogenic emissions may be in California. I commend the authors for including both Figure 9 and Figure 10. Examination of Figure 9 seems to suggest that the HCHO/NO2 ratio in the most populated regions, the San Francisco Bay Area and Los Angeles South Coast Air Basin (SoCAB), do not have a strong seasonal dependence while Figure 10 makes it clear that they have some. It might be good if the authors expanded their discussion of the relative and rather strong seasonal differences in the HCHO/NO2 ratio between different sites in California.

Figure 9 is the TROPOMI HCHO/NO2 map for California. Figure 10 is the monthly averaged TROPOMI HCHO/NO2 averaged for each air basin. We expand Figure 10 below to show the monthly variation of HCHO/NO2 for different cities in SOCAB. The paragraph discussing this revised Figure has been updated in Section 3.3 of the revised manuscript.

"The seasonal variation of  $O_3$  sensitivity can be observed over the entire state of California using the TROPOMI HCHO/NO2. Figure 10a shows how the O3 sensitivity seasonal pattern differs among different air basins. The air basins with the highest populations have suppressed seasonal variation of O3 sensitivity because of the higher anthropogenic NOx emissions. The difference in the seasonal variation of O3 sensitivity can also be observed within air basins. Figure 10b illustrates the TROPOMI HCHO/NO2 monthly variation for different cities in SoCAB between February to October, 2020. The cities inside/around the LA urban core have HCHO/NO2 < 4.6 throughout the entire year with a weak seasonal variation. This might be caused by reduced BVOC emissions in the urban center. The remote areas (darker colors in Figure 10b) have greater seasonal variation and higher peak HCHO/NO2. The sharp increase of HCHO/NO2 in summer leads to a shift in O3 sensitivity from the NOx-saturated regime to the NOx-limited regime in the cities further away from the urban core. Due to the different seasonal variation of HCHO/NO2 at different sites, the NOx-saturated region around the urban core will shrink in the summer and expand in the winter. Figure SX shows this seasonal pattern of O3 sensitivity regime distribution in Los Angeles as an example."

Figure 10. Monthly variation of TROPOMI HCHO/NO2 in different air basins (left panel) and in different cities in South Coast Air Basin (SoCAB) (right panel). The darker colors in the right panel indicate increasing distance from the urban center of Los Angeles.

---

## Author Comment (AC4)

We thank the reviewer for a set of very comprehensive comments. We have used a combination of measurements and model calculations to evaluate how these issues could impact the overall results of the paper. In summary, none of the issues changes the major findings of the manuscript. Our detailed responses for each comment are listed below, along with the changes made to the manuscript to make these findings clear to readers. Our responses to the comments are presented in blue. The comments are shown in black.

1. The authors stated that "A new technique was used to directly measure O3 response to changes in precursor NOx and VOC concentrations..…."; it will be helpful for the readers to have more detailed description of the measurements and the improvements compared to other recent smog chambers studies.

This comment is similar to the 1$^{st}$ comment from RC2. We have clarified the novel features of the current experiment in the Introduction section of the revised manuscript:

"Mobile smog chambers bridge the gap between laboratory studies and the real atmosphere. Past studies have designed mobile smog chambers to measure the aging of secondary pollutants (i.e., O$_3$, SOA) from certain emission source (Howard et al., 2008, 2010; Li et al., 2019; Platt et al., 2013; Presto et al., 2011). It is difficult to evaluate sensitivity of secondary pollutants formed from multiple sources using a single smog chamber. Recently, a mobile dual smog chamber system has been used to directly measure the SOA formation in ambient air (Jorga et al., 2020; Kaltsonoudis et al., 2019). Our smog chamber system consists of three chambers designed to simultaneously analyze the non-linear response of NO$_x$ and VOC to O$_3$ formation. The automated valve and sampling system also allows long-term remote field measurements to evaluate the seasonal trends in O$_3$ sensitivity."

2. The authors used artificial light to provide constant UV radiation in the chamber experiments, which is different from the real atmospheric conditions. Additionally, the settings of other parameters for the smog chambers, such as temperature, relative humidity, etc., are important in modifying the O$_3$ formation but they were not provided in the measurement section. More importantly, it is not reasonable to explore the seasonal changes of the O$_3$ sensitivity using chambers with constant UV radiation. Except for anthropogenic emissions changes, variations of solar radiation play a major role in the seasonal pattern of O$_3$ formation sensitivity. O$_3$ formation regime becomes more NO$_X$-sensitive in warm seasons, which is mainly caused by intensified solar radiation. Increasing solar radiation enhances BVOCs emissions that are light- and temperature-dependent, facilitates photochemical reactions, and promotes development of the planetary boundary layer to decrease near-surface NO$_2$ concentrations.

This comment is similar to several comments submitted by RC1. We have performed a comprehensive sensitivity analysis to investigate the effects temperature and constant UV radiation on the chamber measurements. We summarize the main points below, and refer the reviewer to the response to RC1 for an expanded discussion that includes plots from the Sensitivity Analysis.

The temperature in the reaction chambers was higher than the ambient temperature due to the heating effects of the UV lights. The difference between the chamber gas temperature and the ambient temperature increased by 5-10°C over the course of each experiment, with the exact temperature profile depending on the measurement month. Despite this temperature increase, all chambers experiences the same temperature profile, and so the comparison of $O_3$ formation between the chambers is not strongly biased by this issue. SAPRC11 chamber model simulations were used to quantify the effect of the chamber vs. ambient temperature difference. The difference between the chamber and ambient temperature has little effect on the $O_3$ sensitivity in each month. Temperature effects do not significantly modify the seasonal variation of the measured $O_3$ sensitivity in the current study. Please see plots in RC1 response.

The UV intensity in the chambers was intentionally maintained at a constant level through all seasons so that the effects of seasonal variation in the ambient concentrations would be more apparent without the added complication of varying UV intensity. A representative average UV intensity was selected for this purpose. As was the case with temperature, all chambers experience the same UV conditions and so this factor is not expected to overly bias the comparison between chambers that acts as the core of the current study. The actual seasonal cycle of UV radiation would generate higher photolysis rates in the summer and lower photolysis rates in the winter that would further amplify the seasonal signal already detected by the measurements with constant UV intensity.

SAPRC11 chamber model simulations were used to quantify the effect of seasonal variations in UV intensity. Simulations were carried out using the measured constant UV radiation in the chamber and using the clear sky UV intensity calculated with the routines in the UCD/CIT CTM based on the lat/lon of the measurement site and the day of year. The calculations show that the difference between the constant solar radiation and the seasonally adjusted solar radiation does not change the seasonal pattern of $O_3$ sensitivity to $NO_x$ and VOC perturbations. The seasonal changes to UV intensity slightly amplifies the magnitude of the seasonal trend in $O_3$ sensitivity (increase the absolute value of $\Delta O_3^{+NO_x}$), but the overall seasonal pattern is unchanged. Please see plots in RC1 response.

This information has been added to the new Sensitivity Analysis section in the revised manuscript.

3. $O_3$ formation sensitivity is investigated only by adding 8 ppb $NO_X$ in chamber #1 and 8 ppb surrogate VOCs in chamber #3. Lacking a series of linear experiments with different concentrations of precursor gases, the current conclusions are drawn from the effects of 8 ppb precursor perturbations on $O_3$ levels of the air masses sampled at one site, which is not sufficient to assess the $O_3$ formation sensitivity *in situ*, let alone the regional $O_3$ sensitivity.

This comment is similar to a comment about linearity submitted by RC1. We address this issue using a combination of measurements and chamber model calculations.

$O_3$ sensitivity measurements were conducted using $NO_x$ perturbations ranging from 1-10 ppb at the UC Davis campus from December 2021 to January 2022 to investigate the non-linear behavior of the chemistry. The results summarized

in Figure 1 below show the $O_3$ response expressed as $\Delta O_3$ (final $O_3$ concentration in base case chamber minus final $O_3$ concentration in $NO_x$ perturbed chamber). The $\Delta O_3$ is negative in all $NO_x$ perturbed tests due to the low VOC emission in winter in Davis, CA (similar to Sacramento). Increasing the magnitude of the $NO_x$ perturbation decreased the $\Delta O_3$ value but did not shift the chemistry into a different regime. It was not possible to make linearity measurements in the $NO_x$-limited regime during the cold winter season, and so these issues will be further explored using chamber model calculations as described below.

[Figure]

**Figure 1. Measured $\Delta O_3$ as a function of different $NO_x$ perturbations. Total number of data points is 24.**

The size of the $NO_x$ perturbation used in the chamber experiments is most important when ambient conditions are close to the ridgeline on the $O_3$ isopleth diagram. An 8 ppb $NO_2$ perturbation may jump over the ridgeline in this case, suggesting that the chemistry is $NO_x$-rich rather than $NO_x$-limited. SAPRC11 chamber model simulations were used to quantify the effect of the 8 ppb $NO_2$ perturbation vs. a smaller 2 ppb $NO_2$ perturbation. As shown in Figure 2 below, this issue does not affect the shape of the seasonal trend in $O_3$ sensitivity measurement, but it does affect the transition months when the atmospheric system changes to $NO_x$-limited behavior. The conclusions of the paper are not changed by this finding, but the revised figure and associated discussion in the new Sensitivity Analysis section of the revised manuscript help clarify this point for readers.

[Figure]

**Figure 2. Monthly variation of chamber $\Delta O_3^{+NO_x}$ at Sacramento using NO₂ perturbations of 2 ppb (solid box) and 8 ppb (open box) from April to December, 2020. Simulations are based on the actual chamber UV radiation and chamber temperature profile.**

4. O₃ production sensitivity is determined by the ratio of NO$_x$ to VOCs. Adding constant 8 ppb NO$_x$ or surrogate VOCs to experimental air masses sampled in different seasons with various precursor concentrations could lead to varying perturbations for the ratio, possibly contributing to the measured seasonal variations in O₃ production sensitivity.

The experimental design intentionally holds multiple factors constant so that the effects of changes in atmospheric composition on O₃ formation sensitivity are more apparent. The size of the perturbations for the NO$_x$ and VOC surrogates were one of these constant factors. The ambient air does go through a seasonal cycle of NO$_x$/VOC levels as summarized on the isopleth diagram in Figure 5 of the original paper. The constant perturbation displayed by the arrows in this figure that point towards the ridgeline of the isopleth.

The chosen size of the constant perturbation (+8 ppb) may mask the exact location of the ridgeline in the O₃ isopleth diagram. We evaluate this issue using SAPRC11 chamber model simulations. The response above indicates that this issue does not change the overall shape of the seasonal shift in O₃ sensitivity from NO$_x$-rich in the winter to NO$_x$-limited in the summer. To investigate whether the constant amount of perturbation would change our conclusion in this paper, we use the same chamber model and calculate the O₃ sensitivity under two conditions: (i) Add constant 2 ppb of NO$_x$; (ii) increase ambient NO$_x$ by 20%. The second case would investigate the O₃ sensitivity when NOx perturbations are very small in the summer season. Figure 3 shows the result of this analysis. Increasing NOx by 20% produces the same seasonal trend in O₃ sensitivity as adding a constant 2 ppb or 8 ppb of NOx, but the smaller size of

the perturbation reduces the $O_3$ response. This issue does not change the shape of the seasonal trend in $O_3$ sensitivity measurement, but it does affect the transition months when the atmospheric system changes to $NO_x$-limited behavior.

As a final note, we once again point out that the ground-based chamber measurements are in very good agreement with the TROPOMI satellite $HCHO/NO_2$ measurements. The similar trend of $HCHO/NO_2$ and $\Delta O_3^{+NO_x}$ indicates that the seasonal variation of $O_3$ sensitivity measured from chamber experiment exists in the real atmosphere.

[Figure]

**Figure 3. Monthly variation of predicted $\Delta O_3^{+NO_x}$ with $NO_x$ perturbation at 20% (solid box) and constant 2ppb (open box) from April to December, 2020 at the Sacramento measurement site.**

5. While smog chamber experiments have been used to simulate the photochemical reactions occurring in the atmosphere, the experiments can not accurately represent the complex real atmospheric conditions. This should be taken into consideration in discussing ozone sensitivity to the precursor gases and in drawing conclusions about emissions control policies

We acknowledge that the current experimental design does not capture all of the complexity in the real atmosphere. Mixing processes in the real atmosphere continue to change the composition at ground level as the planetary boundary layer grows throughout the afternoon. Fresh emissions will continue to impact the chemistry of $O_3$ formation. Only 3D chemical transport models can attempt to represent all of these competing effects, but measurements are needed to help evaluate those model calculations. The current experiment is focused on measuring the response of the chemical production term to changes in precursor $NO_x$ and VOC concentrations because this most closely approximates the local effects of potential emissions control programs. No technique will be perfect, but we believe

that the current measurements add information to the weight of science approach used to design effective emissions control programs.

These limitations to the current study have been clarified on Sensitivity Analysis section of the revised manuscript.

**Reference**

Howard, C. J., Yang, W., Green, P. G., Mitloehner, F., Malkina, I. L., Flocchini, R. G. and Kleeman, M. J.: Direct measurements of the ozone formation potential from dairy cattle emissions using a transportable smog chamber, Atmos. Environ., 42(21), 5267–5277, doi:10.1016/j.atmosenv.2008.02.064, 2008.

Howard, C. J., Kumar, A., Malkina, I., Mitloehner, F., Green, P. G., Flocchini, R. G. and Kleeman, M. J.: Reactive organic gas emissions from livestock feed contribute significantly to ozone production in central California, Environ. Sci. Technol., 44(7), 2309–2314, doi:10.1021/es902864u, 2010.

Jorga, S. D., Kaltsonoudis, C., Liangou, A. and Pandis, S. N.: Measurement of Formation Rates of Secondary Aerosol in the Ambient Urban Atmosphere Using a Dual Smog Chamber System, Environ. Sci. Technol., 54(3), 1336–1343, doi:10.1021/acs.est.9b03479, 2020.

Kaltsonoudis, C., Jorga, S. D., Louvaris, E., Florou, K. and Pandis, S. N.: A portable dual-smog-chamber system for atmospheric aerosol field studies, Atmos. Meas. Tech., 12(5), 2733–2743, doi:10.5194/amt-12-2733-2019, 2019.

Li, Y., Alaimo, C. P., Kim, M., Kado, N. Y., Peppers, J., Xue, J., Wan, C., Green, P. G., Zhang, R., Jenkins, B. M., Vogel, C. F. A., Wuertz, S., Young, T. M. and Kleeman, M. J.: Composition and Toxicity of Biogas Produced from Different Feedstocks in California, Environ. Sci. Technol., doi:10.1021/acs.est.9b03003, 2019.

Platt, S. M., Haddad, I. El, Zardini, A. A., Clairotte, M., Astorga, C., Wolf, R., Slowik, J. G. and Universit, A.: Secondary organic aerosol formation from gasoline vehicle emissions in a new mobile environmental reaction chamber, , 9141–9158, doi:10.5194/acp-13-9141-2013, 2013.

Presto, A. A., Nguyen, N. T., Ranjan, M., Reeder, A. J., Lipsky, E. M., Hennigan, C. J., Miracolo, M. A., Riemer, D. D. and Robinson, A. L.: Fine particle and organic vapor emissions from staged tests of an in-use aircraft engine, Atmos. Environ., 45(21), 3603–3612, doi:10.1016/J.ATMOSENV.2011.03.061, 2011.

---

## Author Response (AR1)

**Response to the reviewers — Article ACP-2021-708**

We thank reviewers for a set of very comprehensive comments. We have used a combination of measurements and model calculations to evaluate how these issues could impact the overall results of the paper. In summary, none of the issues raised in the reviewer comments changes the major findings of the manuscript. Our detailed responses for each comment are listed below, along with the changes made to the manuscript to make these findings clear to readers. Our responses to the comments are presented in blue. The comments are shown in black. All page and reference numbers in our response are based on the revised manuscript. The line and reference numbers mentioned in the reviewers' comments are kept intact and are based on the original manuscript.

**Reviewer 1**

**Major issues**

**RC 1.1** — The first line of the abstract states: "A new technique was used to directly measure O3 response to changes in precursor NOx and VOC concentrations in the atmosphere." However, neither this paper nor any of the references demonstrate the validity of this statement. In this regard, I see three major problems:

**RC 1.1.1** — First, the smog chamber system is inadequately described, and many questions remain in my mind regarding its performance. An adequate description could possibly be added to this manuscript, probably in the supplement, but the authors should consider publishing a stand-alone paper in a journal such as Atmospheric Measurement Techniques (AMT), before attempting to develop this paper describing the results.

We have addressed each of the issues raised by the reviewer in the current paper through a combination of additional measurements and calculations. A chamber model developed by Howard et al (2008, 2010a, 2010b) was employed as a part of this analysis to quantify the sensitivity of the $O_3$ response to $NO_x$ perturbations under different experimental configurations. The chemical reaction system used by the chamber model is based on the SAPRC11 chemical mechanism (Carter and Heo, 2013) with wall loss rates based on the measured value of 5% $hr^{-1}$. The time integration procedures used to solve the set of differential equations that predict concentrations as a function of time are taken from the full UCD/CIT chemical transport model (Venecek et al., 2018; Ying et al., 2007).

Day-specific values of NO, $NO_2$, and $O_3$ initial concentrations used in the chamber simulations are based on measurements near the study location. VOC initial concentrations used in the chamber simulations are based on UCD/CIT simulations over the study location. The seasonal profile of the simulated VOC concentrations matches the CO*biogenic trends illustrated in Figure 2 of the manuscript, but the amplitude of the simulated seasonal trend was damped. VOC initial concentrations used in the chamber simulations were therefore scaled to match the amplitude of the CO*biogenic factor. The seasonal pattern of $O_3$ response to $NO_x$ perturbations predicted by the SAPRC11 chamber model closely matches the measured trends shown in Figure 1. Chamber model calculations

will be used as part of each response to the reviewer comments below. The description about the model has been added in revised manuscript (Line 215-225).

[Figure]

**Figure 1. Monthly variation of the $\Delta O_3^{+NO_x}$ predicted by the chamber model (solid box) and directly measured in the chamber (open box) from April to December, 2020 at the Sacramento measurement site.**

Issues that should be included in that description are:

**RC 1.1.1.1** — Temperature control of the chambers – It has been widely reported (e.g., Coates et al., 2016) that temperature affects ozone formation. Are the chambers held at ambient temperature in spite of irradiation by the UV lamp panels? (I presume that the chambers are enclosed in a light-tight structure to exclude ambient sunlight – this should be fully described).

The temperature in the reaction chambers was higher than the ambient temperature due to the heating effects of the UV lights. Figure 2 below shows that the difference between the chamber gas temperature and the ambient temperature increased by 5-10°C over the course of each experiment, with the exact temperature profile depending on the measurement month. Despite this temperature increase, all 3 chambers experience the same temperature profile, and so the comparison of $O_3$ formation between the chambers is not strongly biased by this issue.

[Figure]

**Figure 2. Time series of chamber gas temperature (blue) and ambient temperature (red) for each month from April to December, 2020. The dots show the monthly averaged value, and the shaded area shows the standard deviation of the temperature in each month. (Figure S10 in revised manuscript)**

The SAPRC11 chamber model was used to quantify the effect of the chamber vs. ambient temperature difference illustrated in Figure 1 above. Figure 3 below shows the calculated $\Delta O_3^{+NO_x}$ during each month of the experiment under the chamber and ambient temperature profiles. The difference between the chamber and ambient temperature has little effect on the $O_3$ sensitivity in each month. Temperature effects do not significantly modify the seasonal variation of the measured $O_3$ sensitivity in the current study. This point has been clarified in the Sensitivity Analysis section (Section 4.1.1, line 491-500) added to the revised manuscript.

[Figure]

**Figure 3. Monthly variation of the predicted $\Delta O_3^{+NO_x}$ under the ambient temperature profile (solid box) and chamber gas temperature profile (open box) from April to December, 2020 at the Sacramento measurement site.**

**RC 1.1.1.2** — Light intensity control – It is generally recognized that ozone formation is a function of solar radiation intensity. In the ambient atmosphere, this intensity varies with solar zenith angle on diurnal and seasonal cycles, as well as ambient clouds and meteorological conditions (clouds, aerosol loading, stratospheric ozone column, etc.) Is there a mechanism to allow the chamber light intensity to mimic the ambient light intensity? Most important I suspect is the seasonal cycle of solar radiation intensity. Figure 3 of the paper shows the seasonal cycle of the smog-chamber results; does the experiment mimic the seasonal variation of the solar radiation intensity? If not, how can this "technique … directly measure O3 response to changes in precursor NOx and VOC concentrations in the atmosphere"?

The UV intensity in the chambers was intentionally maintained at a constant level through all seasons so that the effects of seasonal variation in the ambient concentrations would be more apparent without the added complication of varying UV intensity. A representative average UV intensity was selected for this purpose. As was the case with temperature, all chambers experience the same UV conditions and so this factor is not expected to overly bias the comparison between chambers that acts as the core of the current study. The actual seasonal cycle of UV radiation would generate higher photolysis rates in the summer and lower photolysis rates in the winter that would further amplify the seasonal signal already detected by the measurements with constant UV intensity.

SAPRC11 chamber model simulations were used to quantify the effect of seasonal variations in UV intensity. Simulations were carried out using the measured constant UV radiation in the chamber and using the clear sky UV intensity calculated with the routines in the UCD/CIT CTM based on the lat/lon of the measurement site and the day

of year. The calculations summarized in Figure 4 below show that the difference associated with the use of constant UV radiation does not change the seasonal pattern of $O_3$ sensitivity to $NO_x$ and VOC perturbations. As expected, the seasonal changes to UV intensity slightly amplifies the magnitude of the seasonal trend in $O_3$ sensitivity (increase the absolute value of $\Delta O_3^{+NO_x}$), but the overall seasonal pattern is unchanged. This information has been added to the new Sensitivity Analysis section (Section 4.1.2, line 502-515) in the revised manuscript.

[Figure]

**Figure 4. Monthly variation of predicted $\Delta O_3^{+NO_x}$ under constant chamber UV radiation (open box) and clear-sky solar radiation (solid box) from April to December, 2020 in the Sacramento measurement site.**

**RC 1.1.1.3** — Blank tests – To develop confidence in the reported results, the authors must show that results of "blank runs" (i.e., filling all three chambers with zero air, adding the standard perturbation amounts of NOx and VOC to the two perturbed chambers, and irradiating for the standard three hours) result in zero ozone formation in all three chambers.

As the reviewer surely knows, adding $NO_2$ to a chamber followed by UV irradiation will definitely form $O_3$. The check requested by the reviewer therefore cannot result in zero $O_3$ formation. Furthermore, "blank tests" with zero air and zero $O_3$ formation are far outside the relevant atmospheric conditions that are the focus of the current study. A much more relevant indicator of the uncertainty in the experimental results is the difference between $O_3$ formed in different chambers across a range of atmospherically-relevant $O_3$ concentrations, since these between-chamber comparisons form the basis of the reported data. The results already reported in the manuscript summarize that the

uncertainty between $O_3$ formation in different chambers operated under the same conditions is 1~2% for final $O_3$ concentrations between 40 – 125 ppb. This information has been highlighted on Figure S2 in the revised manuscript.

Even though the results are far outside the range of atmospherically-relevant concentrations, chambers were filled with zero air and irradiated for 180 min to address the request for a literal blank test. Figure 5 (Figure S2 in the revised SI) shows the results of this "blank" test alongside the original consistency test results measured at atmospherically-relevant $O_3$ concentrations. The final $O_3$ concentration in all 3 chambers during "blank" tests were less than 4 ppb and (more importantly) the difference between chambers that forms the basis of the reported $O_3$ sensitivity was less than 1 ppb (see points near the origin in Figure 5). These results are consistent with the sensitivity reported for atmospherically-relevant $O_3$ concentrations. This confirms that the $O_3$ measured in each chamber during normal operation is formed by the reaction of the ambient air plus perturbed gases. Any biases in the ozone formation have similar effects on all chambers and therefore very little effect on the comparison between chambers. This information has been added to the consistency test paragraph (line 142-151) in Section 2.1 in the manuscript.

[Figure]

**Figure 5. Consistency check of three 1 m³ FEP bags using equal NO$_x$-VOC mixture. Points near the origin were measured with zero air. The equation and $R^2$ shows the linear regression results of $O_3$ concentration in perturbed chamber to basecase chamber. The 95% confidence intervals (CI) of regression coefficient are (0.996, 1.017) for bag 1, and (1.002, 1.013) for bag 3.**

**RC 1.1.1.4** — Ambient condition tests – Again, to develop confidence that the reported results actually "directly measure $O_3$ response to changes in precursor NO$_x$ and VOC concentrations in the atmosphere" it would seem critical to remove any light-tight shroud around the chambers so that they are exposed to ambient solar radiation, to not operate the UV lamps, and then to compare the ozone evolution in the chamber with the evolution of ambient ozone. Only if

the chamber ozone actually tracks the ambient ozone, can it be accepted that a direct measurement of the ozone response is actually obtained.

The focus of the current study is to maintain UV intensity constant at an atmospherically-relevant level so that changes in $O_3$ sensitivity can be more directly associated with changes in atmospheric composition. We realize this is not a literal direct measurement of $O_3$ response, but rather it is a direct measurement of $O_3$ chemical production that is closer to a direct measurement than any other technique that has been previously demonstrated. If the reviewer (and Editor) feel that the claim of a direct measurement is too strong, then we would agree to soften the language slightly to claim direct measurements of the sensitivity of $O_3$ chemical production, or "semi-direct" measurements of $O_3$ sensitivity. The basecase $O_3$ chemical production rates are consistent with the ambient measurements as discussed below, and so we believe the measurements are atmospherically-relevant.

The first version of the manuscript used measurements from a preliminary experiment in Los Angeles to evaluate whether the chemical production rate of chamber $O_3$ was consistent with ambient measurements. The lower initial $O_3$ concentrations shown in that original figure were caused by the time-lag between the start time of ambient air injection and the start time of the chamber measurement. Figure S3 in the revised manuscript (shown as Figure 6 below) has been updated using a more comprehensive analysis over a longer time period for Sacramento to increase confidence in the analysis. Figure 4 shows the weekly-average ozone profile for each month of the year measured in the basecase chamber (dots) and the nearby ambient monitor (solid line). The initial $O_3$ concentrations in the base case chamber are similar to the ambient $O_3$ concentration at the start of each measurement period. The chemical production rate of $O_3$ measured in the basecase chamber is generally consistent with the rate of change in the $O_3$ concentrations measured at the ambient monitor between 10 am ~ 12 pm. The chemical production rate of $O_3$ in the chamber is higher than the increase in the ambient $O_3$ concentration because the ambient concentration is also affected by deposition and transport (Cazorla et al., 2012).

[Figure]

**Figure 6. Weekly averaged Ambient (solid line) vs. Chamber (solid circles) O₃ concentrations measured in Sacramento for each month from April to December, 2020. The shaded area indicates one standard deviation of the ambient O₃ concentration. Chambers were filled over a ~2hr period followed by a 30 min measurement period before UV lights were turned on. Hour is relative to the start of the experiment. (Appears as Figure S3 in revised manuscript)**

The following information will be added to Section 2.1 (line 153-164) in the revised manuscript to replace the original paragraph discussing Figure S3.

Weekly-averaged $O_3$ concentrations in the basecase chamber were compared to weekly-averaged ambient $O_3$ concentrations measured at the nearby monitoring station from April to December 2020 (Figure S3). The $O_3$ concentrations in the basecase chamber at the start of each experiment were similar to the ambient $O_3$ concentrations, indicating that the gas-phase chemical composition related to $O_3$ formation was not changed while injecting ambient air into the chamber. The $O_3$ formation in the chamber generally reflects the $O_3$ chemical production from the in-situ ambient air around 10 am ~ 12 pm in the morning, while the ambient $O_3$ is influenced by chemical production, mixing, and deposition (Cazorla et al., 2012). As expected, the initial rate of $O_3$ formation in the chamber is therefore higher than the initial rate of change in the ambient $O_3$ concentrations. The current experiment is focused on measuring the response of this chemical production rate to changes in precursor $NO_x$ and VOC concentrations because this most closely approximates the local effects of potential emissions control programs.

**RC 1.1.1.5** — Linearity tests – With regard to comment c below, the response of the smog-chamber system to different magnitude perturbation concentrations must be investigated. The figure at right shows example diurnal cycles of the $NO_X$ concentrations in four months, one from each season, measured at the monitoring site adjacent to the smog-chamber location (Fig. S5). In summer and spring (the seasons of most policy relevance) the added $NO_2$ perturbation (8 ppb) in the smog chamber more than doubles the $NO_X$ concentration. Thus, the physical significance of the derived ozone formation sensitivity is questionable.

$O_3$ sensitivity measurements were conducted using $NO_x$ perturbations ranging from 1-10 ppb at the UC Davis campus from December 2021 to January 2022 to investigate the non-linear behavior of the chemistry. The results summarized in Figure 7 below show the $O_3$ response expressed as $\Delta O_3$ (final $O_3$ concentration in base case chamber minus final $O_3$ concentration in $NO_x$ perturbed chamber). The $\Delta O_3$ is negative in all $NO_x$ perturbed tests due to the low VOC emission in winter in Davis, CA (similar to Sacramento). Increasing the magnitude of the $NO_x$ perturbation decreased the $\Delta O_3$ value but did not shift the chemistry into a different regime. It was not possible to make linearity measurements in the $NO_x$-limited regime during the cold winter season, and so these issues will be further explored using chamber model calculations as described below.

[Figure]

**Figure 7. Measured ΔO₃ as a function of different NOₓ perturbations. Total number of data points is 24.**

The size of the NOₓ perturbation used in the chamber experiments is most important when ambient conditions are close to the ridgeline on the O₃ isopleth diagram. An 8 ppb NO₂ perturbation may jump over the ridgeline in this case, suggesting that the chemistry is NOₓ-rich rather than NOₓ-limited. SAPRC11 chamber model simulations were used to quantify the effect of the 8 ppb NO₂ perturbation vs. a smaller 2 ppb NO₂ perturbation. As shown in Figure 8 below, this issue does not affect the shape of the seasonal trend in O₃ sensitivity measurement, but it does affect the transition months when the atmospheric system changes to NOₓ-limited behavior. The conclusions of the paper are not changed by this finding, but the revised figure and associated discussion in the new Sensitivity Analysis section (section 4.1.3, line 516-536) of the revised manuscript help clarify this point for readers.

[Figure]

**Figure 8. Monthly variation of chamber $\Delta O_3^{+NO_x}$ at Sacramento using $NO_2$ perturbations of 2 ppb (solid box) and 8 ppb (open box) from April to December, 2020. Simulations are based on the actual chamber UV radiation and chamber temperature profile.**

**RC 1.1.2** — Second, I do not believe that the system can actually directly measure ozone sensitivity in the sense that it accurately reflects how actual ambient ozone concentrations would respond to precursor emission changes. In the real atmosphere, during the photochemical active period of a day, ambient air parcels are transported through an air basin. During that transport, dilution and mixing processes occur, fresh emissions are injected into the air parcel and ozone is lost to surface deposition simultaneously with in situ photochemical ozone production. It seems to me that the smog chamber can only reproduce one (albeit very important) aspect of this extremely complex ambient ozone production process. A late morning, integrated air parcel is captured in the chamber, and then the in situ photochemical ozone production is mimicked in isolation from all other processes. This issue should be thoroughly discussed, and the authors should acknowledge that their approach can potentially determine the sensitivity of the in situ photochemical ozone production to precursor $NO_x$ and VOC, but likely that does not directly correspond to the sensitivity of the actual ozone concentrations in the ambient Sacramento boundary layer.

We believe that the measurement does reflect how ambient $O_3$ concentrations would respond to changes in emissions. The experiment measures the sensitivity of the $O_3$ chemical production term in response to the concentration of $NO_x$ and VOC. This is the most appropriate measurement of how local emission controls will affect the local $O_3$ concentrations. The experiment may not directly capture all of the atmospheric processes, but it represents the dominant processes. The agreement between the measured results at ground level and the satellite measurements build confidence that the results are capturing the most important features of the atmospheric system. An enhanced

discussion of the issues above has been added to the Sensitivity Analysis section (Section 4.1) of the revised manuscript.

**RC 1.1.3** — Third, when the sensitivity of ozone is discussed, it is generally understood that the sensitivity is referring to the response of ozone to decreases in precursor $NO_x$ and VOC. However, the smog chamber experiment operates by investigating increases in those precursors. If ozone production chemistry responded linearly to precursor changes, this distinction would be unimportant; however, it is widely acknowledged that ozone chemistry is highly non-linear. Thus, the smog chamber approach must give biased results. For example, if the ambient atmosphere were on the "ridgeline" of the corresponding ozone isopleth diagram, then the smog chamber data would indicate VOC sensitivity, since the ozone production would decrease with added $NO_x$ (i.e., $\Delta O_3^{+NO_x}$ would be negative). But if the experiment could be run with a NOx decrease, rather than an increase, the ozone produced would again decrease, indicating NOx sensitivity. The extent of the bias resulting from the non-linearity of the ozone response depends upon the relative magnitude of the precursor perturbations. A very small, potentially infinitesimal, perturbation would reduce, potentially eliminate, the bias; however, to obtain a precisely measurable response, I suspect that the precursor perturbations were rather large relative to the ambient concentrations. Since the $NO_x$ concentrations were actually measured in these experimental runs, a thorough discussion of this potential source of bias should be given in the context of the magnitude of the $NO_x$ perturbations relative to the initial ambient $NO_x$ concentrations in the chamber when the experimental run is initiated.

This question was addressed in the response to the comment about linearity in response to the $NO_x$ perturbation. To summarize, box model calculations confirm that the size of the positive $NO_x$ perturbation does not change the overall observation that $O_3$ sensitivity transitions from $NO_x$-rich during winter months to $NO_x$-limited in summer months. That is why the ground-based measurement trends match the independent TROPOMI satellite measurement trends. If anything, the 8 ppb $NO_2$ perturbation slightly changes the timing of the transition and damps the magnitude of the $O_3$ sensitivity during the transition months rather than artificially enhancing the trends. A thorough discussion about this issue is included in the Sensitivity Analysis section (section 4.1) of the revised manuscript.

**RC 1.2** — This paper emphasizes the policy relevance of the results. The last two sentences of the abstract state: "This challenging situation suggests that emissions control programs that focus on NOx reductions will immediately lower peak O3 concentrations, but slightly increase intermediate O3 concentrations until NOx levels fall far enough to re-enter the NOx-limited regime. The spatial pattern of increasing and decreasing O3 concentrations in response to a NOx emissions control strategy should be carefully mapped in order to fully understand the public health implications." However, the smog chamber work is analyzed from the perspective of the final ozone concentration in the chamber at the end of the experimental run. The policy relevance would be much more clearly evident in this work if the analysis perspective focused on the ambient MDA8 ozone concentration on the day of each run. In particular, Figure 6 would be more informative if the x-axis variable were the MDA8 ambient ozone concentration recorded at the monitoring

site adjacent to the smog chamber field location (see Figure S5). A great deal more support must be given before these policy-relevant statements can be accepted. In this regard, the findings must be directly related to the conditions that produce ambient MDA8 ozone concentrations that exceed the NAAQS, as discussed in the 2nd paragraph of the Introduction Section of the paper.

Figure 6 in the manuscript (shown as Figure 9 below) was updated to use MDA8 $O_3$ concentration from the nearby CARB monitoring station as requested. The text in Section 3.1.5 has been revised to describe the updated figure.

"The days with the highest measured $O_3$ concentrations are of particular interest in the current study since emissions control programs are traditionally tailored to reduce the $O_3$ design value, which is determined by daily maximum 8-hour average (MDA8) $O_3$ concentration. Figure 6 illustrates box-and-whisker plots of measured $\Delta O_3^{+NO_x}$, and $\Delta O_3^{+VOC}$ at Sacramento binned according to the MDA8 $O_3$ concentration measured at the monitoring station near the chamber measurement site. The right two bins, corresponding to the $O_3$-nonattainment days (MDA8 $O_3$ > 70 ppb), have $O_3$ sensitivity in the $NO_x$-limited regime where $NO_x$ addition increases $O_3$ concentrations and VOC addition has minor effects on $O_3$ concentrations. These measurements suggest that a $NO_x$ emissions control strategy would be most effective at reducing these peak $O_3$ concentrations. In contrast, a large portion of the days with MDA8 $O_3$ concentrations below 55 ppb were in the VOC-limited regime, suggesting that an emissions control strategy focusing on $NO_x$ reduction would increase $O_3$ concentrations. VOC controls on these intermediate days would be difficult, however, if biogenic VOCs account for the majority of the $O_3$ formation. This challenging situation suggests that emissions control programs that focus on $NO_x$ reductions will immediately lower peak $O_3$ concentrations, but slightly increase intermediate $O_3$ concentrations until $NO_x$ levels fall far enough to re-enter the $NO_x$-limited regime."

[Figure]

**Figure 9: Boxplot of O₃ sensitivity to NOₓ and VOC as a function of MDA8 O₃ concentration. (Appears as Figure 6 in revised manuscript)**

**Minor Issues**

**RC 1.3** — Lines 54-61: These sentences discuss references that propose causes of the increase of O3 design values in some air basins between the years 2015 – 2018. However, many of the cited references were published before that increase occurred, so they obviously do not directly address that increases. This discussion must be improved with the inclusion of appropriate references. To my knowledge tenable proposed causes include the influences of wildfire emissions and particularly pronounced heat waves; however the causes the authors discuss in these lines really are not tenable. For example, "growing importance of precursor VOC emissions not previously accounted for in the planning process" could possibly account for a slowing of the ozone decrease, but (unless those emissions increased markedly over that short 2015 – 2018 period) could not account for an increase. Similarly, climate has not changed markedly over that short 2015 – 2018 period, so this cause also is not tenable. If the authors wish to discuss this rather minor feature of Figure 1 (i.e., there are other wiggles in the trend of similar magnitude), then they should do so in a rigorous manner. Perhaps a cause could be sought that accounts for the increase in some (e.g. SoCAB and San Diego as discussed in the manuscript), but not in other Southern California air basins (e.g., South Central Coast Air Basin, which is adjacent to SoCAB).

The text in the Introduction section has been modified to include additional explanations and references as shown below. The modified part is mainly at Line 54-61 in the revised manuscript.

"$O_3$ levels are often described by the maximum 8-hr average concentration that occurs within each day. The annual fourth-highest daily maximum 8-hr average concentration averaged over three years has special regulatory significance. This "design value" determines whether the region containing the monitor complies with the $O_3$ NAAQS. $O_3$ design values in California decreased steadily between the years 1980 and 2019 (Figure 1) due to the success of emissions control programs that reduced concentrations of precursors broadly divided into two groups: oxides of nitrogen ($NO_x$) and volatile organic compounds (VOCs) (Parrish et al., 2016; Simon et al., 2015). Continued progress after the year 2010 has been slower, and $O_3$ design values even increased in some air basins between the years 2015 – 2018 (Figure 1). Multiple factors have been proposed to explain the lack of further reductions in $O_3$ concentrations in recent years. These potential factors include: (i) growing importance of precursor VOC emissions not previously accounted for in the planning process as major sources such as transportation have been controlled (McDonald et al., 2018; Shah et al., 2020), (ii) an imbalance in the historical degree of $NO_x$ and VOC reductions (Cox et al., 2013; Parrish et al., 2016; Pollack et al., 2013; Steiner et al., 2006), or (iii) more frequent heat waves (Jacob and Winner, 2009; Jing et al., 2017; Pusede et al., 2015; Rasmussen et al., 2013; Weaver et al., 2009) and wildfires (Jaffe et al., 2013; Lindaas et al., 2017; Lu et al., 2016; Singh et al., 2012) as a consequence of climate change. All these theories are supported to varying degrees by indirect measurements or model predictions, but there is an absence of strong direct evidence that identifies dominant factors contributing to the increased $O_3$ concentrations. The uncertainty that lingers over the recent $O_3$ trends suggests that fresh approaches are needed to directly verify the optimum emissions control path."

**RC 1.4** — Lines 143-153: This paragraph is not persuasive. The statement "The initial $O_3$ concentration in the basecase chamber was similar to the ambient $O_3$ concentration, indicating that the gas phase chemical composition related to $O_3$ formation is not modified during chamber injection" requires more discussion. Figure S3 clearly indicates that the initial $O_3$ concentration in the basecase chamber was always significantly (10-30 ppb) below the ambient concentrations at the initial time. This issue and its impacts on the entire analysis must be thoroughly discussed. I suggest that this discussion include an expanded time scale for some specific examples so comparison between the basecase chamber and ambient air is much more clearly illustrated. Further, the tests included in Fig. S3 were conducted in Los Angeles (an urban area with very different ozone levels and presumably photochemical environment) than Sacramento, where the primary field work was conducted (e.g., see Figure 1 of the paper).

This issue has been answered as a part of response for Ambient condition test in Major issue section. The text related to this issue is copied below:

The first version of the manuscript used measurements from a preliminary experiment in Los Angeles to evaluate whether the chemical production rate of chamber $O_3$ was consistent with ambient measurements. The lower initial $O_3$ concentrations shown in that original figure were caused by the time-lag between the start time of ambient air injection and the start time of the chamber measurement. Figure S3 in the revised manuscript (shown as Figure 6 below) has been updated using a more comprehensive analysis over a longer time period for Sacramento to increase confidence in the analysis. Figure 4 shows the weekly-average ozone profile for each month of the year measured in the basecase chamber (dots) and the nearby ambient monitor (solid line). The initial $O_3$ concentrations in the base case chamber are similar to the ambient $O_3$ concentration at the start of each measurement period. The chemical production rate of $O_3$ measured in the basecase chamber is generally consistent with the rate of change in the $O_3$ concentrations measured at the ambient monitor between 10 am ~ 12 pm. The chemical production rate of $O_3$ in the chamber is higher than the increase in the ambient $O_3$ concentration because the ambient concentration is also affected by deposition and transport (Cazorla et al., 2012).

**RC 1.5** — Lines 143-153: This same paragraph discusses the comparison of the $O_3$ increase in the basecase chamber and in the ambient air; that discussion is greatly oversimplified. In the ambient atmosphere, the early morning $O_3$ increase is largely driven by mixing down of ozone rich air from aloft as the boundary layer rapidly grows during that period. In the SoCAB, the land-sea breeze circulation affects the diurnal ozone cycle during the day. The statement "The $O_3$ formation in the chamber, therefore, captures a realistic "worst-case scenario" for surface-level $O_3$ formation under conditions where atmospheric mixing cannot dilute the $NO_x$ and VOC concentrations that build up in the nocturnal ground-level stagnation layer." is simply not justified – the conditions inside the chambers are very different from ambient conditions. These differences must be thoroughly discussed – not simply "hand waved" away. It should be realized that the predominant growth of the convective boundary layer generally approaches its

maximum extent by noon, which is the time that the experimental run begins (e.g.. see Figures 4-7 of Bianco et al., 2011).

The description of Figure S3 has been revised in the main manuscript in Line 153-164. It's also shown in the Major issue about Ambient condition test section. The revised paragraph is copied below:

Weekly-averaged $O_3$ concentrations in the base case chamber were compared to weekly-averaged ambient $O_3$ concentrations measured at the nearby monitoring station from April to December 2020 (Figure S3). The $O_3$ concentrations in the base case chamber at the start of each experiment were similar to the ambient $O_3$ concentrations, indicating that the gas-phase chemical composition related to $O_3$ formation was not changed while injecting ambient air into the chamber. The $O_3$ formation in the chamber generally reflects the $O_3$ chemical production from the in-situ ambient air around 10 am ~ 12 pm in the morning, while the ambient $O_3$ is influenced by chemical production, mixing, and deposition (Cazorla et al., 2012). As expected, the initial rate of $O_3$ formation in the chamber is therefore higher than the initial rate of change in the ambient $O_3$ concentrations. The current experiment is focused on measuring the response of this chemical production term to changes in precursor $NO_x$ and VOC concentrations because this most closely approximates the local effects of potential emissions control programs.

**RC 1.6** — Many of the figures show linear regression fits. However, It appears that there may be shortcomings and errors in some of them. These issues should be checked and corrected if necessary; specifically:

**RC 1.6.1** — Figure S2. Confidence limits (preferably 2 sigma or 95%) for the slopes should be included to indicate that the slopes are indeed consistent with unity.

The Figure S2 has been updated as mentioned in Major issue. The confidence interval has been added in the caption of Figure S2.

**RC 1.6.2** — Figure S4. Given the large scatter in the data points and the small correlation coefficients, the exceedingly small p values, and the relatively small shaded areas appear to me to not be realistic (and the meaning of the shaded areas should be defined.) Please check all such fits in all figures to be sure the fitting is properly calculated.

The calculation has been checked. The shaded area indicates the 95% confidence interval of the mean response of the predicted value. The confidence interval of the mean response is tighter than the scatter in the individual data points. The meaning of the shaded area has been described in the caption of all such plots in both the manuscript and SI)

**RC 1.7** — The discussion of the VOC and CO relationships (lines 155-173) requires improvement.

**RC 1.7.1** — The statement "Biogenic sources do not emit CO and so any correlation between biogenic VOCs and CO purely reflects the utility of CO as an indicator of atmospheric mixing that equally affects all sources" is incorrect and misleading. An important source of CO is partial oxidation of biogenic VOCs, so their correlation is more complex than indicated here. Further, atmospheric mixing does not equally affect all sources, since the result of mixing is dependent on the background concentrations in the diluting air.

The statement has been revised as "Biogenic sources do not emit CO but biogenic VOCs can react in the atmosphere to produce CO (Hudman et al., 2008). CO also acts as an indicator of atmospheric mixing that equally affects all primary sources." In Line 173-175 in revised manuscript.

**RC 1.7.2** — The quantity CO*Biogenic is not clearly defined. Where were the sites of those VOC measurements? More details of the "temperature and relative humidity-induced enhancement factor for isoprene emissions" must be given. The cited reference is now 30 years old; in the intervening 3 decades a great deal has been learned about biogenic VOC emissions. Is this "enhancement factor" consistent with current understanding?

The Model of Emissions of Gases and Aerosols from Nature (MEGAN) model is widely used to estimate the BVOC emissions. The temperature response used in our manuscript is based on the published MEGAN temperature response (Guenther et al., 2012). We believe this factor is still appropriate for use in the current study.

**RC 1.7.3** — Given the quoted R value in Fig. S4, it should be mentioned that use of CO*Biogenic as an approximate surrogate for VOCR only captures ~36% of the variance of VOCR at the site where the CO and VOCR measurements were made.

The following text has been added to discuss the details of Figure S4 in the revised SI.

"Figure S 4 shows the correlation between the sum of species measured in the PAMS network multiplied by their $O_3$ formation potential (=VOCR) vs. candidate surrogate measures of VOC reactivity (=CO and CO*biogenic). The p-value in each panel quantifies the probability that the surrogate has zero correlation with VOCR. The R-value in each panel quantifies the amount variation about the mean value of VOCR that is explained by the surrogate. CO*biogenic explains 36% of the VOCR variability about the mean VOCR value, while CO alone explains 15% of the VOCR variability about the mean VOCR value. CO*biogenic is therefore selected as the preferred (but not perfect) surrogate for VOC concentrations in the current study."

**RC 1.7.4** — It should be explicitly stated whether the CO and VOC measurements were made at the same monitoring site, and the location of this site relative to the location of the chamber measurements should be discussed.

CO and VOC are from the same monitoring site in Sacramento. A detailed description of the data sources has been added in caption of Figure S4 in revised Supplementary. The location of CO and VOC data source is the closest monitoring site to the chamber measurement site that have both CO and VOC data available.

**RC 1.8** — Section 2.3. The brief experimental description is not adequate. Questions that occur to me include: How can air be sampled from the chambers without disturbing the environment? Do the sides of the chamber gradually collapse? If the sides collapse, what fraction of the air is exhausted through the sampling process over the 210-minute experimental run? Why is a linear extrapolation required? Section 2.1 reports that ozone loss rates were 5%/hour in the chambers; was correction made for this loss rate? Figure S2 indicates that the perturbed chambers gave 1 to 2% greater ozone production than the base chamber; was correction made for this difference? How many experimental runs were made over the 11 month period of Figure 2, and included in the box and whisker plots? In addition to an expanded experimental discussion that answers these questions, I suggest including sample chamber measurement data from a typical experimental run as a section in the Supplement. That section should clearly describe all steps included in the process of deriving the DO3 values of Figure 3 from the 3-hr time series of concentration measurements. It would also be useful to indicate the number of experimental runs included in each box and whisker plot in Figure 3 (and in subsequent figures).

All monitors exhaust through tubing that is released several meters away from the trailer at the ground level. The nearby CARB monitoring station is several hundred meters away from the trailer and so it is not influenced by trailer operations. One experiment was conducted each day, releasing a total of 3 m$^3$ to the ambient air over a 3–4 hour period. This low level will not influence nearby measurements.

The chamber sides collapse as air is withdrawn from the chamber. The total sample flow rate for all monitors is approximately 3 L/min. Seven measurements with a duration of 10 min are made from each chamber resulting in a total sample volume of 210 L air, or approximately 21% of the chamber volume (leaving 79% of the total air in the chamber). The shape of the chambers is not greatly distorted at any point during the experiment. These points are clarified in the Methods section of the revised manuscript in Line 134-137.

The sequential sampling strategy means that measurements from different chambers are always made at different times. There will always be a difference of at least 10 min between O$_3$ measurements in each chamber. We fit a linear regression to O$_3$ concentrations as a function of time to enable a comparison of O$_3$ concentrations at the same time at the end of each experiment. These points are clarified in the Methods section of the revised manuscript in Line 207-209.

The ozone loss rate of 5% per hour was used to correct the O$_3$ concentration in each chamber before we apply the linear regression. This point is clarified in the Section 8 in revised Supplementary in Line 115-117.

Figure S2 shows the reproducibility of $O_3$ formation in the three chambers. From the coefficient of the linear regression, the $O_3$ concentration in the perturbed chamber may have ~ 1% difference with the concentration in the basecase chamber. This is the uncertainty of the chamber comparison, not the bias, and so it was not necessary to correct the comparison between chamber measurements. The ~1% uncertainty of $O_3$ formation from three chambers is acceptable for the $O_3$ sensitivity analysis. This point is clarified in the Methods section of the revised manuscript in Line 147-148.

There are 222 experiment runs from Apr 14 to Dec 20, 2020 (out of a total of 251 days). We will add this information in the main text (Line 197). The example of a typical day of results is shown and added to the revised SI Section 8:

[Figure]

**Figure S7. $O_3$ concentration in 3 chambers under the UV exposure during a typical chamber experiment on August 16, 2020 in Sacramento. Lines shows the linear regression result of $O_3$ concentration under UV exposure in each chamber.**

Figure S7 shows an example of the time series of chamber $O_3$ concentration under the UV exposure. The x-axis reflects the UV exposure duration time in the chamber. Each dot is 10-min averaged $O_3$ concentration corrected by $O_3$ wall loss rate. Dots with different colors correspond to different chambers. Linear regression was applied to $O_3$ concentrations in each chamber and the results are summarized as solid lines. The projected $O_3$ concentration at the end of the 180-min UV exposure time was calculated based on the regression results (hereafter referred to as $3hr\ O_3^{Bag\ 1}$, $3hr\ O_3^{Bag\ 2}$, and $3hr\ O_3^{Bag\ 3}$). The measured sensitivities $\Delta O_3^{+NO_x}$, and $\Delta O_3^{+VOC}$ were calculated using the equation below:

$$\Delta O_3^{+NO_x} = 3hr\ O_3^{Bag\ 2} - 3hr\ O_3^{Bag\ 1}$$

$$\Delta O_3^{+VOC} = 3hr\ O_3^{Bag\ 3} - 3hr\ O_3^{Bag\ 1}$$

**RC 1.9** — In this regard, Figure S1 seems to indicate that four lamps were mounted on the floor of the middle chamber, and eight were mounted on the floor of each end chamber. Does this difference in the figure reflect the reality of the chambers? If so, please explain why this arrangement was used, and give more discussion regarding why this arrangement does not bias the results.

Figure S1 exactly represents the light configuration used in the experiment. The three smog chambers are contained in a single rectangular enclosure with reflective wall panels. The length of the enclosure is approximately 0.5 m longer than the combined length of the chambers. This geometry uses space efficiently but the distance between the reflective end walls and the outside chambers is different than the distance between the reflective end walls and the center chamber. The distribution of lights was chosen to achieve equal UV intensity for each chamber in this geometric configuration. Multiple light configurations were tested with UV measurements at each chamber. The configuration summarized in Figure S1 achieved the most uniform distribution of UV among the chambers. The consistency of $O_3$ formation in all chambers initialized with the same composition confirms that the light distribution produces the same photolysis rates in each chamber. Moreover, the chamber named bag1,2,3 in the consistency test only represent the position of chamber in the system. The actual chambers were rotated during the consistency checks to verify that the equivalent $O_3$ formation across chambers was not caused by compensating errors. These points have been clarified in the text associated with Figure S1 in the revised manuscript (Line 15-21).

**RC 1.10** — The discussion of Figure 2 is not adequate. Why are there no TROPOMI measurements in November and December? Reading the figure caption seems to indicate that CO measurements were made in the ground-based chambers; however, Section 2.1 seems to indicate that only $NO_x$, $NO_y$, $O_3$, temperature and relative humidity were measured in the chambers. Please explain clearly how the CO*biogenic values were determined. Evidently the isoprene concentrations in Figure 2 were measured at an EPA PAMS site; Figure S5 indicates two monitoring sites. It should be indicated which (if either) of those sites reported the isoprene measurements discussed here. The meaning of the lines in the box and whisker plots should be explicitly indicated, here and in later figures.

TROPOMI data for November and December were not available during the first round of data analysis for this paper. We have updated the 'Figure 2' and 'Figure 3' (shown below as Figure 10 and Figure 11) in the revised manuscript with TROPOMI data through December 2020. TROPOMI data in November and December matches well with the chamber measurement. The CO data was collected from a nearby CARB monitoring site that have CO concentration available in Sacramento. The Figure S5 indicates the monitoring site for ambient CO, $NO_x$, and $O_3$ concentration. The caption has been revised to correspond the site to each pollutant species. The EPA PAMS site information has been added in the revised SI in Section 5.

[Figure]

**Figure 10. Monthly concentrations of NO₂ (panels a) and CO\*Biogenic/HCHO/Isoprene (panels b) from February to December 2020. Ground-based chamber measurements use the left axis with results shown as box and whisker plots. TROPOMI measurements use the right axis and are shown as diamonds. Isoprene from ground monitoring station shown as blue triangles. The open box and points show the results after removing wildfire days.**

[Figure]

**Figure 11. Monthly variance of TROPOMI HCHO/NO₂ (diamond) and ΔO₃ (box) due to NOₓ addition ($\Delta O_3^{+NO_x}$) and VOC addition ($\Delta O_3^{+VOC}$) from April to December including wildfire days (top) and without wildfire days (bottom).**

**RC 1.11**—The final paragraph of Section 3.1.1 discusses VCPs, but requires improvement. VCP emissions are not related to either CO emissions or isoprene emissions (except if isoprene is one of the VCPs). Thus, there is no reason to expect seasonal pattern similarity between VCPs and CO*biogenic values, or between VCPs and isoprene. Nevertheless, there is nothing here to indicate that VCPs are not important (or even dominant) in driving ozone production in Sacramento (although I agree that this is very unlikely). This paragraph should be modified or eliminated.

The statements suggest that possible increased VCP emission from sanitizing products due to COVID-19 likely did not change the seasonal trend of VOC. This paragraph did not say that VCP is not an important precursor of $O_3$ formation, but it discusses the potential influence of COVID-19 on VOC emission in Sacramento. This paragraph correctly notes that the seasonal trend of BVOC is more consistent with the measured trends in Sacramento. We believe the paragraph was clear as originally written.

**Reviewer 2**

**RC 2.1** — Clearly this research is very relevant to policymakers for the development of emission control strategies to improve air quality. This study is not the first to develop and apply mobile smog chambers to air quality measurements at specific sites. For example, Mobile Smog Chamber, https://www.psi.ch/en/lac/mobile-smog-chamber; Kaltsonoudis et al., A portable dual-smog-chamber system for atmospheric aerosol field studies, Atmos. Meas. Tech., 12, 2733–2743, 2019. I believe that there was even a commercial chamber, about the size of a soccer ball, that was marketed to directly measure $NO_x$ and VOC limitation by an Australian company (I apologize for not being able to find a reference). My point is that the authors should include a couple of paragraphs summarizing previous mobile chambers and discuss how their new system is an improvement.

The original manuscript only has 2 sentences talking about previous studies that employed transportable smog chambers. As requested, we have searched for additional studies that use transportable smog chambers. A more thorough discussion has been added to the Introduction section in the revised manuscript (Line 91-100).

"Mobile smog chambers bridge the gap between laboratory studies and the real atmosphere. Past studies have designed mobile smog chambers to measure the aging of secondary pollutants (i.e., $O_3$, SOA) from certain emission source (Howard et al., 2008, 2010b; Li et al., 2019; Platt et al., 2013; Presto et al., 2011). It is difficult to evaluate sensitivity of secondary pollutants formed from multiple sources using a single smog chamber. Recently, a mobile dual smog chamber system has been used to directly measure the SOA formation in ambient air (Jorga et al., 2020; Kaltsonoudis et al., 2019). Our smog chamber system consists of three chambers designed to simultaneously analyze the non-linear response of $O_3$ formation to $NO_x$ and VOC perturbations. The automated valve and sampling system also allows long-term remote field measurements to evaluate the seasonal trends in $O_3$ sensitivity."

**RC 2.2** — The TROPOMI measurements are especially interesting in that they indicate how important biogenic emissions may be in California. I commend the authors for including both Figure 9 and Figure 10. Examination of Figure 9 seems to suggest that the $HCHO/NO_2$ ratio in the most populated regions, the San Francisco Bay Area and Los Angeles South Coast Air Basin (SoCAB), do not have a strong seasonal dependence while Figure 10 makes it clear that they have some. It might be good if the authors expanded their discussion of the relative and rather strong seasonal differences in the $HCHO/NO_2$ ratio between different sites in California.

Figure 9 is the TROPOMI $HCHO/NO_2$ map for California. Figure 10 is the monthly averaged TROPOMI $HCHO/NO_2$ averaged for each air basin. We expand Figure 10 below to show the monthly variation of $HCHO/NO_2$ for different cities in SOCAB. The paragraph discussing this revised Figure has been updated in Section 3.3 of the revised manuscript (Line 458-470).

"The seasonal variation of $O_3$ sensitivity can be observed over the entire state of California using the TROPOMI $HCHO/NO_2$ (Table S1). Figure 10a shows how the $O_3$ sensitivity seasonal pattern differs among different air basins. The air basins with the highest populations have suppressed seasonal variation of $O_3$ sensitivity because of the higher

anthropogenic $NO_x$ emissions. The difference in the seasonal variation of $O_3$ sensitivity can also be observed within air basins. Figure 10b illustrates the TROPOMI $HCHO/NO_2$ monthly variation for different cities in SoCAB between February to October, 2020. The cities inside/around the LA urban core have $HCHO/NO_2 < 4.6$ throughout the entire year with a weak seasonal variation. This might be caused by reduced BVOC emissions in the urban center. The remote areas (darker colors in Figure 10b) have greater seasonal variation and higher peak $HCHO/NO_2$. The sharp increase of $HCHO/NO_2$ in summer leads to a shift in $O_3$ sensitivity from the $NO_x$-saturated regime to the $NO_x$-limited regime in the cities further away from the urban core. Due to the different seasonal variation of $HCHO/NO_2$ at different sites, the $NO_x$-saturated region around the urban core will shrink in the summer and expand in the winter. Figure S8 shows this seasonal pattern of $O_3$ sensitivity regime distribution in Los Angeles as an example."

[Figure]

**Figure 10. Monthly variation of TROPOMI $HCHO/NO_2$ in different air basins (a) and in different cities in South Coast Air Basin (SoCAB) (b). The darker colors in the right panel indicate increasing distance from the urban center of Los Angeles.**

[Figure]

**Figure S8.** Spatial distribution of O₃ sensitivity regime based on TROPOMI satellite (HCHO/NO₂) ratios in Los Angeles for April – October 2020. Light area is in NOₓ-limited regime (HCHO/NO₂ > 4.6), dark area is in NOₓ-saturated regime (HCHO/NO₂ <= 4.6)

**Reviewer 3**

**RC 3.1** — The authors stated that "A new technique was used to directly measure $O_3$ response to changes in precursor $NO_x$ and VOC concentrations...…"; it will be helpful for the readers to have more detailed description of the measurements and the improvements compared to other recent smog chambers studies.

This comment is similar to the comment in RC 2.1. We have clarified the novel features of the current experiment in the Introduction section of the revised manuscript (Line 92-100):

"Mobile smog chambers bridge the gap between laboratory studies and the real atmosphere. Past studies have designed mobile smog chambers to measure the aging of secondary pollutants (i.e., $O_3$, SOA) from certain emission source (Howard et al., 2008, 2010b; Li et al., 2019; Platt et al., 2013; Presto et al., 2011). It is difficult to evaluate sensitivity of secondary pollutants formed from multiple sources using a single smog chamber. Recently, a mobile dual smog chamber system has been used to directly measure the SOA formation in ambient air (Jorga et al., 2020; Kaltsonoudis et al., 2019). Our smog chamber system consists of three chambers designed to simultaneously analyze the non-linear response of $NO_x$ and VOC to $O_3$ formation. The automated valve and sampling system also allows long-term remote field measurements to evaluate the seasonal trends in $O_3$ sensitivity."

**RC 3.2** — The authors used artificial light to provide constant UV radiation in the chamber experiments, which is different from the real atmospheric conditions. Additionally, the settings of other parameters for the smog chambers, such as temperature, relative humidity, etc., are important in modifying the $O_3$ formation but they were not provided in the measurement section. More importantly, it is not reasonable to explore the seasonal changes of the $O_3$ sensitivity using chambers with constant UV radiation. Except for anthropogenic emissions changes, variations of solar radiation play a major role in the seasonal pattern of $O_3$ formation sensitivity. $O_3$ formation regime becomes more $NO_x$-sensitive in warm seasons, which is mainly caused by intensified solar radiation. Increasing solar radiation enhances BVOCs emissions that are light- and temperature-dependent, facilitates photochemical reactions, and promotes development of the planetary boundary layer to decrease near-surface $NO_2$ concentrations.

This comment is similar to several comments submitted by RC1 (mainly in RC1.1). We have performed a comprehensive sensitivity analysis to investigate the effects temperature and constant UV radiation on the chamber measurements. We summarize the main points below, and refer the reviewer to the response to RC1 for an expanded discussion that includes plots from the Sensitivity Analysis.

The temperature in the reaction chambers was higher than the ambient temperature due to the heating effects of the UV lights. The difference between the chamber gas temperature and the ambient temperature increased by 5-10°C over the course of each experiment, with the exact temperature profile depending on the measurement month. Despite this temperature increase, all chambers experience the same temperature profile, and so the comparison of $O_3$

formation between the chambers is not strongly biased by this issue. SAPRC11 chamber model simulations were used to quantify the effect of the chamber vs. ambient temperature difference. The difference between the chamber and ambient temperature has little effect on the $O_3$ sensitivity in each month. Temperature effects do not significantly modify the seasonal variation of the measured $O_3$ sensitivity in the current study. Please see plots in RC1 response.

The UV intensity in the chambers was intentionally maintained at a constant level through all seasons so that the effects of seasonal variation in the ambient concentrations would be more apparent without the added complication of varying UV intensity. A representative average UV intensity was selected for this purpose. As was the case with temperature, all chambers experience the same UV conditions and so this factor is not expected to overly bias the comparison between chambers that acts as the core of the current study. The actual seasonal cycle of UV radiation would generate higher photolysis rates in the summer and lower photolysis rates in the winter that would further amplify the seasonal signal already detected by the measurements with constant UV intensity.

SAPRC11 chamber model simulations were used to quantify the effect of seasonal variations in UV intensity. Simulations were carried out using the measured constant UV radiation in the chamber and using the clear sky UV intensity calculated with the routines in the UCD/CIT CTM based on the lat/lon of the measurement site and the day of year. The calculations show that the difference between the constant solar radiation and the seasonally adjusted solar radiation does not change the seasonal pattern of $O_3$ sensitivity to $NO_x$ and VOC perturbations. The seasonal changes to UV intensity slightly amplifies the magnitude of the seasonal trend in $O_3$ sensitivity (increase the absolute value of $\Delta O_3^{+NO_x}$), but the overall seasonal pattern is unchanged. Please see plots in RC1 response.

This information has been added to the new Sensitivity Analysis section (Section 4.1) in the revised manuscript.

**RC 3.3** — $O_3$ formation sensitivity is investigated only by adding 8 ppb $NO_x$ in chamber #1 and 8 ppb surrogate VOCs in chamber #3. Lacking a series of linear experiments with different concentrations of precursor gases, the current conclusions are drawn from the effects of 8 ppb precursor perturbations on $O_3$ levels of the air masses sampled at one site, which is not sufficient to assess the $O_3$ formation sensitivity *in situ*, let alone the regional $O_3$ sensitivity.

This comment is similar to a comment about linearity submitted by RC1 (**RC 1.1.1.5**). We address this issue using a combination of measurements and chamber model calculations. This information has been added in the revised manuscript (Line 517-538).

$O_3$ sensitivity measurements were conducted using $NO_x$ perturbations ranging from 1-10 ppb at the UC Davis campus from December 2021 to January 2022 to investigate the non-linear behavior of the chemistry. The results summarized in Figure 12 below show the $O_3$ response expressed as $\Delta O_3$ (final $O_3$ concentration in base case chamber minus final $O_3$ concentration in $NO_x$ perturbed chamber). The $\Delta O_3$ is negative in all $NO_x$ perturbed tests due to the low VOC

emission in winter in Davis, CA (similar to Sacramento). Increasing the magnitude of the $NO_x$ perturbation decreased the $\Delta O_3$ value but did not shift the chemistry into a different regime. It was not possible to make linearity measurements in the $NO_x$-limited regime during the cold winter season, and so these issues will be further explored using chamber model calculations as described below.

[Figure]

**Figure 12. Measured $\Delta O_3$ as a function of different $NO_x$ perturbations. Total number of data points is 24.**

The size of the $NO_x$ perturbation used in the chamber experiments is most important when ambient conditions are close to the ridgeline on the $O_3$ isopleth diagram. An 8 ppb $NO_2$ perturbation may jump over the ridgeline in this case, suggesting that the chemistry is $NO_x$-rich rather than $NO_x$-limited. SAPRC11 chamber model simulations were used to quantify the effect of the 8 ppb $NO_2$ perturbation vs. a smaller 2 ppb $NO_2$ perturbation. As shown in Figure 13 below, this issue does not affect the shape of the seasonal trend in $O_3$ sensitivity measurement, but it does affect the transition months when the atmospheric system changes to $NO_x$-limited behavior. The conclusions of the paper are not changed by this finding, but the revised figure and associated discussion in the new Sensitivity Analysis section of the revised manuscript help clarify this point for readers.

[Figure]

**Figure 13. Monthly variation of chamber $\Delta O_3^{+NO_x}$ at Sacramento using NO₂ perturbations of 2 ppb (solid box) and 8 ppb (open box) from April to December, 2020. Simulations are based on the actual chamber UV radiation and chamber temperature profile.**

**RC 3.4** — O₃ production sensitivity is determined by the ratio of $NO_x$ to VOCs. Adding constant 8 ppb $NO_x$ or surrogate VOCs to experimental air masses sampled in different seasons with various precursor concentrations could lead to varying perturbations for the ratio, possibly contributing to the measured seasonal variations in O₃ production sensitivity.

The experimental design intentionally holds multiple factors constant so that the effects of changes in atmospheric composition on O₃ formation sensitivity are more apparent. The size of the perturbations for the $NO_x$ and VOC surrogates were one of these constant factors. The ambient air does go through a seasonal cycle of $NO_x$/VOC levels as summarized on the isopleth diagram in Figure 5 of the original paper. The constant perturbation displayed by the arrows in this figure that point towards the ridgeline of the isopleth.

The chosen size of the constant perturbation (+8 ppb) may mask the exact location of the ridgeline in the O₃ isopleth diagram. We evaluate this issue using SAPRC11 chamber model simulations. The response above indicates that this issue does not change the overall shape of the seasonal shift in O₃ sensitivity from $NO_x$-rich in the winter to $NO_x$-limited in the summer. To investigate whether the constant amount of perturbation would change our conclusion in this paper, we use the same chamber model and calculate the O₃ sensitivity under two conditions: (i) Add constant 2 ppb of $NO_x$; (ii) increase ambient $NO_x$ by 20%. The second case would investigate the O₃ sensitivity when NOx perturbations are very small in the summer season. Figure 14 shows the result of this analysis. Increasing NOx by 20% produces the same seasonal trend in O₃ sensitivity as adding a constant 2 ppb or 8 ppb of NOx, but the smaller size of

the perturbation reduces the $O_3$ response. This issue does not change the shape of the seasonal trend in $O_3$ sensitivity measurement, but it does affect the transition months when the atmospheric system changes to $NO_x$-limited behavior.

As a final note, we once again point out that the ground-based chamber measurements are in very good agreement with the TROPOMI satellite $HCHO/NO_2$ measurements. The similar trend of $HCHO/NO_2$ and $\Delta O_3{}^{+NOx}$ indicates that the seasonal variation of $O_3$ sensitivity measured from chamber experiment exists in the real atmosphere.

[Figure]

**Figure 14. Monthly variation of predicted $\Delta O_3^{+NO_x}$ with $NO_x$ perturbation at 20% (solid box) and constant 2ppb (open box) from April to December, 2020 at the Sacramento measurement site.**

**RC 3.5** — While smog chamber experiments have been used to simulate the photochemical reactions occurring in the atmosphere, the experiments cannot accurately represent the complex real atmospheric conditions. This should be taken into consideration in discussing ozone sensitivity to the precursor gases and in drawing conclusions about emissions control policies

We acknowledge that the current experimental design does not capture all of the complexity in the real atmosphere. Mixing processes in the real atmosphere continue to change the composition at ground level as the planetary boundary layer grows throughout the afternoon. Fresh emissions will continue to impact the chemistry of $O_3$ formation. Only 3D chemical transport models can attempt to represent all of these competing effects, but measurements are needed to help evaluate those model calculations. The current experiment is focused on measuring the response of the chemical production term to changes in precursor $NO_x$ and VOC concentrations because this most closely approximates the local effects of potential emissions control programs. No technique will be perfect, but we believe that the current measurements add information to the weight of science approach used to design effective emissions control programs.

These limitations to the current study have been clarified on Sensitivity Analysis section of the revised manuscript (Section 4.1).

**Community comments**

**CC 1 —** This is an interesting study presenting direct measurements of ozone response to emissions perturbations in California. I am not a referee of this paper. I post my comments to anticipate a better study.

Thank you for your comments.

**CC 1.1 —** Both NO2 and O3 in main text and supplement should be expressed using subscript.

Changes will be made throughout the manuscript as suggested.

**CC 1.2 —** "Trend" is usually used for the variability of long term scale, at least, for year-scale. I don't think we can call the diurnal, seasonal, or even 1-3 years of change rates (variability) as "trend".

Different fields may use the term "trend" for different purposes. The most general definition is "a general direction in which something is developing or changing". We believe that "trend" is the most appropriate term to describe the pattern of changing concentrations / sensitivities as a function of day-of-the-year given that our response variable is measured daily.

**CC 1.3 —** line 67-69, "that lower NOx concentrations are associated with higher O3 concentrations on weekends", I would say "....higher O3 concentrations on weekends are associated with lower NOx concentrations ..... ". Same revision for the subsequent sentence.

Change will be made as suggested.

**CC 1.4 —** The HCHO/NO2 is a time and region dependent indicator. Especially the change regime threshold is trick stuff. Please double check the comparison between the TROPOMI-based and ground-based values.

All TROPOMI data has been checked, and two additional months of TROPOMI data have been added to the analysis.

[revised manuscript text omitted]

---

## Referee Report (RR1)

2nd Review of "Direct Measurements of Ozone Response to Emissions Perturbations in California" by Shenglun Wu et al.

MS Number: acp-2021-708

**Summary:**

This paper aims to characterize the overall ozone response to NOx and VOC emission reductions in California by synthesizing satellite data with the results of experiments that perturbed the ambient photochemistry through in-the-field smog chamber runs. This is an interesting study that provides a useful addition to our understanding of ozone formation in California. The authors have done a commendable job of addressing the two major shortcomings in the description and interpretation of the smog chamber experiments that concerned me in my first review. I am impressed by the sensitivity analysis presented in Section 4.1; that analysis presents a balanced discussion of the usefulness of that approach, as well as its limitations. I recommend that the paper be accepted after the authors address the minor issues discussed below.

It should be noted that my expertise is in laboratory and field measurements, and that I have little expertise in the interpretation of satellite data. The discussion of the TROPOMI measurements are compelling to me, and it seems to be supported by references, which I have not reviewed. I would be more comfortable if the satellite data discussion were reviewed by someone with the necessary experience to thoroughly review the interpretation.

**Minor Issues:**

1. Line 26 of the text has the phrase "… with baseline chamber $O_3$ concentrations above 90 ppb …." I suggest that this sentence and the following sentence be reworded to reflect the discussion of Figure 6 based on ambient MDA8 ozone, rather than on the chamber ozone. I think this would make the information in the abstract more policy relevant, and better reflect the discussion in the paper.

2. Line 88: Should "source" be plural?

3. I suggest adding a sentence to the end of Section 2.4 that mentions the extensive sensitivity tests that were performed to ensure the relevance of the results, and will be discussed in Section 4.1

4. The discussion of Figure 2 in Section 3.1.1 requires improvement. It is noted that good agreement is observed between the time trends of the chamber and TROPOMI satellite remote sensing measurements. It is also suggested that the upward trend in $NO_2$ concentrations in October–December, 2020 is likely associated with decreased boundary layer heights and increased fuel consumption for heating during the colder fall – winter season. However, the satellite measurement is a column measurement; the in situ chamber $NO_2$ concentrations would depend upon boundary layer height, but a true column measurement would be independent of the boundary layer height. A more accurate discussion is required here, and that discussion should fully consider the averaging kernel of the satellite column measurement.

5. The discussion in the preceding comment also applies to the comparison between the in situ and satellite column measurements of HCHO and CO*Biogenic; this discussion also should

be improved.

6. The caption of Figure 3 mentions "variance", a term that has a specific statistical meaning. I suggest replacing it with "variation".

7. The figures are mis-numbered: two numbered 6 and figs. 8 and 9 incorrectly labeled Figs 7 and 8. The text does refer to the correct numbering.

8. In Figure 10, the cities in the right graph are not all in SoCAB; some are in the Salton Sea AB; I also think that it would be useful to include the Salton Sea AB in the left graph,

9. Line 460: please correct figure number and tense "Figure 10 shows that the chamber model can accurately predicted the measured …."

10. Line 464: should refer to Fig. S9.

11. The sentence beginning on Line 492 is easy to misinterpret; I suggest rewording: "Increasing the magnitude of the $NO_X$ perturbation increased the absolute magnitude of the $\Delta O3$ value but did not shift the chemistry into a different regime."

12. The last sentence in Section 4.1.4 should be amplified slightly; I suggest rewording: "It should be noted that operation of the mobile smog chamber system in cities with higher ambient NOx concentrations is expected to give $O_3$ sensitivity results that are even less dependent on the $NO_2$ perturbation size.

13. Line 513: I suggest that the sentence begin with "Current California …", since California has a history of addressing a great many precursor emission sectors.

14. I would strongly argue that the sentence beginning on line 518 is inaccurate. Ambient measurements indicate that over decades VOCs and NOx have decreased at average annual rates of about 7.5% (Warneke et al., 2012) and 2.6%, respectively. Over 30-years (1980 to 2010) those rates correspond to decreases of factors of 10 and 2.2, respectively. These are based on measurements in the SoCAB, but are relevant for the entire state. The Cox et al. and Rasmussen et al. references rely on emission inventories for their estimates, which are far inferior to actual ambient measurements. In my view it is important that the tremendous success of emission control efforts is highlighted at every opportunity. At the very least, this paper should discuss both the inventory and ambient measurement estimates of emission decreases.

**Reference:**

Pollack, I. B., T. B. Ryerson, M. Trainer, J. A. Neuman, J. M. Roberts, and D. D. Parrish (2013), Trends in ozone, its precursors, and related secondary oxidation products in Los Angeles, California: A synthesis of measurements from 1960 to 2010, J. Geophys. Res. Atmos., 118, 5893–5911, doi:10.1002/jgrd.50472.

Warneke, C., J. A. de Gouw, J. S. Holloway, J. Peischl, T. B. Ryerson, E. Atlas, D. Blake, M. Trainer, and D. D. Parrish (2012), Multiyear trends in volatile organic compounds in Los Angeles, California: Five decades of decreasing emissions, J. Geophys. Res., 117, D00V17, doi:10.1029/2012JD017899.

---

## Author Response (AR2)

**Response to the reviewers — Article ACP-2021-708**

We thank reviewers for their thoughtful comments. Our detailed responses for each comment are listed below, along with the changes made to the manuscript to make these findings clear to readers. Our responses to the comments are presented in blue. The comments are shown in black. All page and reference numbers in our response are based on the revised manuscript. The line and reference numbers mentioned in the reviewers' comments are kept intact and are based on the original manuscript.

**Reviewer 1**

**Minor Issues:**

**RC1.1.** Line 26 of the text has the phrase "… with baseline chamber O3 concentrations above 90 ppb …." I suggest that this sentence and the following sentence be reworded to reflect the discussion of Figure 6 based on ambient MDA8 ozone, rather than on the chamber ozone. I think this would make the information in the abstract more policy relevant, and better reflect the discussion in the paper.

We have revised the sentence mentioned by reviewer in Line 27:

"The $O_3$-nonattainment days (MDA8 $O_3$ > 70 ppb) have $O_3$ sensitivity in the $NO_x$-limited regime, suggesting that a $NO_x$ emissions control strategy would be most effective at reducing these peak $O_3$ concentrations. In contrast, a large portion of the days with MDA8 $O_3$ concentrations below 55 ppb were in the VOC-limited regime, suggesting that an emissions control strategy focusing on $NO_x$ reduction would increase $O_3$ concentrations."

**RC1.2.** Line 88: Should "source" be plural?

'source' has been changed to 'sources'.

**RC1.3.** I suggest adding a sentence to the end of Section 2.4 that mentions the extensive sensitivity tests that were performed to ensure the relevance of the results, and will be discussed in Section 4.1

A sentence has been added in the end of Section 2.4:

'Section 4.1 presents a sensitivity study on the chamber measurement result using the chamber model described here.'

**RC1.4.** The discussion of Figure 2 in Section 3.1.1 requires improvement. It is noted that good agreement is observed between the time trends of the chamber and TROPOMI satellite remote sensing measurements. It is also suggested that the upward trend in NO2 concentrations in October–December, 2020 is likely associated with decreased boundary layer heights and increased fuel consumption for heating during the colder fall – winter season. However, the satellite measurement is a column measurement; the in situ chamber NO2 concentrations would depend upon boundary layer height, but a true column measurement would be independent of the boundary layer height. A more accurate discussion is required here, and that discussion should fully consider the averaging kernel of the satellite column measurement.

Increased boundary layer heights are often associated with increased wind speeds in the boundary layer, leading to downwind advection and dispersion of pollutants. This increased dispersion reduces the column concentrations of $NO_x$ or HCHO measured by the TROPOMI satellite. This point has been clarified on line 247 of the revised manuscript.

**RC1.5.** The discussion in the preceding comment also applies to the comparison between the in situ and satellite column measurements of HCHO and CO*Biogenic; this discussion also should be improved.
Same response to RC1.4.

**RC1.6.** The caption of Figure 3 mentions "variance", a term that has a specific statistical meaning. I suggest replacing it with "variation".
'variance' has been changed to 'variation'.

**RC1.7.** The figures are mis-numbered: two numbered 6 and figs. 8 and 9 incorrectly labeled Figs 7 and 8. The text does refer to the correct numbering.
Figure numbers have been corrected.

**RC1.8.** In Figure 10, the cities in the right graph are not all in SoCAB; some are in the Salton Sea AB; I also think that it would be useful to include the Salton Sea AB in the left graph,
Results for the Salton Sea Air Basin have been added to Figure 10(a) and the caption for Figure 10(b) now refers to the region as "Southern California", instead of "SoCAB".

[Figure]

**Figure 10. Monthly variation of TROPOMI HCHO/NO₂ in different air basins (left panel) and in different cities in Southern California (right panel). The darker colors in the right panel indicate increasing distance from the urban center of Los Angeles.**

**RC1.9.** Line 460: please correct figure number and tense "Figure 10 shows that the chamber model can accurately predicted the measured …."
The figure number has been changed to 'Figure 11', and the word 'predicted' has been changed to 'predict'.

**RC1.10.** Line 464: should refer to Fig. S9.

This change has been made in revised manuscript.

**RC1.11.** The sentence beginning on Line 492 is easy to misinterpret; I suggest rewording: "Increasing the magnitude of the NOx perturbation increased the absolute magnitude of the $\Delta O_3$ value but did not shift the chemistry into a different regime."

This change has been made in revised manuscript in Line 529.

**RC1.12.** The last sentence in Section 4.1.4 should be amplified slightly; I suggest rewording: "It should be noted that operation of the mobile smog chamber system in cities with higher ambient $NO_x$ concentrations is expected to give $O_3$ sensitivity results that are even less dependent on the $NO_2$ perturbation size.

This change has been made in revised manuscript in Line 547.

**RC1.13.** Line 513: I suggest that the sentence begin with "Current California …", since California has a history of addressing a great many precursor emission sectors.

This change has been made in revised manuscript in Line 550.

**RC1.14.** I would strongly argue that the sentence beginning on line 518 is inaccurate. Ambient measurements indicate that over decades VOCs and NOx have decreased at average annual rates of about 7.5% (Warneke et al., 2012) and 2.6%, respectively. Over 30-years (1980 to 2010) those rates correspond to decreases of factors of 10 and 2.2, respectively. These are based on measurements in the SoCAB, but are relevant for the entire state. The Cox et al. and Rasmussen et al. references rely on emission inventories for their estimates, which are far inferior to actual ambient measurements. In my view it is important that the tremendous success of emission control efforts is highlighted at every opportunity. At the very least, this paper should discuss both the inventory and ambient measurement estimates of emission decreases.

We have added a sentence on line 556 of the revised manuscript to describe the trends in ambient measurement that support the decreasing emissions.

Revised text:

The estimated VOC emissions decreased by a factor of 3 while $NO_x$ emission decreased by a factor of 1.5 between 1980 to 2010 according to the California inventory (Cox et al., 2013; Rasmussen et al., 2013). Long-term ambient measurements in the SoCAB confirm that ambient VOC concentrations decreased at an average rate of 7.5% $yr^{-1}$ , while ambient $NO_x$ concentrations decreased at an average rate of 2.6% $yr^{-1}$ between the years 1980 to 2010 (Pollack et al., 2013; Warneke et al., 2012).

**Reference:**

Cox, P., Delao, A. and Komorniczak, A.: The California Almanac of Emissions and Air Quality - 2013 Edition.

[online] Available from: https://www.arb.ca.gov/aqd/almanac/almanac13/almanac13.htm, 2013.

Pollack, I. B., Ryerson, T. B., Trainer, M., Neuman, J. A., Roberts, J. M., Parrish, D. D., Pollack, C. :, Ryerson, T. B., Trainer, M., Roberts, J. M. and Parrish, D. D.: Trends in ozone, its precursors, and related secondary oxidation products in Los Angeles, California: A synthesis of measurements from 1960 to 2010, J. Geophys. Res. Atmos., 118(11), 5893–5911, doi:10.1002/JGRD.50472, 2013.

Rasmussen, D. J., Hu, J., Mahmud, A. and Kleeman, M. J.: The ozone-climate penalty: Past, present, and future, Environ. Sci. Technol., 47(24), 14258–14266, doi:10.1021/es403446m, 2013.

Warneke, C., De Gouw, J. A., Holloway, J. S., Peischl, J., Ryerson, T. B., Atlas, E., Blake, D., Trainer, M., Parrish, D. D., Warneke, C. :, De Gouw, J. A., Holloway, J. S., Peischl, J., Ryerson, T. B., Atlas, E., Blake, D., Trainer, M. and Parrish, D. D.: Multiyear trends in volatile organic compounds in Los Angeles, California: Five decades of decreasing emissions, J. Geophys. Res. Atmos., 117(D21), 0–17, doi:10.1029/2012JD017899, 2012.

**Reviewer 4**

This paper provides results about the response of ozone concentration to changes in precursor $NO_x$ and VOC concentrations. The bulk of the experimental and analysis setup relies on smog chambers analysis. They authors offer quantitative results illustrating the complexities of near surface ozone chemistry and atmospheric conditions and its seasonal variation.

The paper also makes use of TROPOMI observations. To better understand the validity of the results obtained from TROPOMI some questions rise:

**RC4.1.** Discuss and provide references evaluating the accuracy of TROPOMI HCHO and $NO_2$ products. Furthermore, are the satellite retrievals corrected to account for the validation results reported in the literature? A good starting point for $NO_2$ will be https://amt.copernicus.org/articles/14/481/2021/, and for HCHO https://amt.copernicus.org/articles/13/3751/2020/amt-13-3751-2020.html. If there is no correction, how do these uncertainties translate into results derived using TROPOMI $NO_2$ and HCHO?

We have added discussions and references evaluating the accuracy of TROPOMI HCHO and $NO_2$ products (Verhoelst et al. 2021, Vigouroux et al. 2020) on Line 189-200 in the revised manuscript as the reviewer recommended.

"Correction factors were not applied to TROPOMI data in the current study. Verhoelst et al. (2021)and Vigouroux et al. (2020a) analyzed the accuracy of the TROPOMI data using ground-based measurement sites across the globe. Measurements were not made in California, but several of the evaluation sites had attributes similar to locations in California.  Bias in daily TROPOMI $NO_2$ retrievals varied between -15% to -56% in moderately polluted areas with $NO_2$ column measurements between $3 \times 10^{15}$ - $14 \times 10^{15}$ molec cm$^{-2}$ (typical for moderate-sized cities in California). The bias in TROPOMI HCHO measurements ranged between +26%±5% at low HCHO levels to -30.8%±1.4% at

high HCHO levels. HCHO levels measured in Sacramento ($\sim0.6 \times 10^{15}$ molec cm$^{-2}$) had a bias of approximately zero. These results suggest that TROPOMI measurements over California almost certainly contain some amount of bias that could only be removed through a comparison to measurements from a ground-based network. Application of global-average bias correction factors would not change the trends in HCHO and $NO_2$ in time and space even if they would change the absolute magnitude of those values. The current analysis will therefore focus on trends in the TROPOMI measurements."

**RC4.2.** What is the reason behind using QA bigger than 0.5 for both products? QA for $NO_2$ and HCHO have slightly different definition. From the TROPOMI HCHO user document: "In order to avoid misinterpretation of the data quality, it is recommended to only use those TROPOMI pixels associated with a qa_value above 0.5 (no error flag, cloud radiance fraction at 340 nm<0.5, Solar Zenith Angle (SZA)<=70°, surface albedo<=0.2, no snow/ice warning, air mass factor>0.1).". The NO2 user guide has a different definition: "qa_value > 0.75. For most users this is the recommended pixel filter. This removes cloud-covered scenes (cloud radiance fraction > 0.5), part of the scenes covered by snow/ice, errors and problematic retrievals.

As the reviewer mentioned, the ESA recommends QA values > 0.75 for most users retrieving values for individual days. In our use case, we averaged TROPOMI data to monthly concentrations, which mitigates the effects of uncertainty inherent in any individual data point. According to the ESA (ATBD document of TROPOMI $NO_2$), QA values > 0.50 add "the good quality retrievals over clouds and over scenes covered by snow/ice. Errors and problematic retrievals are still filtered out." Data with QA values > 0.50 therefore seems reasonable when constructing monthly averages. Incorporating TROPOMI $NO_2$ data with QA values > 0.5 increased the number of available data points and produced more robust statistics than calculations that only used individual data points that passed the highest level of QA. These points have been included in the discussion on Line 186 of the revised manuscript.

Revised text:
Quality assurance (QA) values were obtained alongside the HCHO and $NO_2$ data, and only measurements with QA values $\geq$ 0.50 were retained to ensure good data quality and sufficient data points when computing monthly averages (Van Geffen et al., 2021).

**RC4.3.** How was the satellite data averaged in the spatial domain for the state wide results?
For the statewide analysis, we re-gridded all the satellite data into 5 km grids and calculated the monthly averages of each data product in each grid. The monthly averages of HCHO and $NO_2$ were used to calculate the ratios. We have added the following in the caption of Figure 9: "TROPOMI $NO_2$ and HCHO data are re-gridded to 5 km resolution when calculating monthly-average ratios."

**RC4.4**. While references to TROPOMI $NO_2$ and HCHO retrieval algorithm papers are provided the paper could benefit of a short description of each one of them.

As the reviewer recommended, we have added a short description of each retrieval algorithm in 1st paragraph of Section 2.2 as follows: "The retrieval algorithms for TROPOMI $NO_2$ data use the measurements of the earth's radiance in the visible absorption wavelengths (405 – 465 nm) made by the hyperspectral imaging spectrometer. The algorithms first derive the total slant column density of $NO_2$ using a Differential Optical Absorption Spectroscopy (DOAS) method. The total slant column $NO_2$ is then separated into stratospheric and tropospheric slant column densities of $NO_2$ while utilizing information from a data assimilation system. Finally, the tropospheric vertical column density of $NO_2$ is obtained by applying conversion factors, called air mass factors (AMFs), to the tropospheric slant column density of $NO_2$. The retrievals of TROPOMI HCHO data apply a similar DOAS method to the ultraviolet (UV) wavelengths (328.5 – 359 nm) of the solar spectrum."

**Other minor comments:**

**RC4.5.** Lines 86-90 provide information about previous studies using satellite observations to derive HCHO/$NO_2$ ratios. It will be good to include some more recent papers using instruments recently launched such as TROPOMI. Motivated by the COVID-19 lockdowns there is significant amount of literature looking at it, for example https://www.science.org/doi/10.1126/sciadv.abe1178. The reference to OMI nadir pixel resolution (13 k x 24 km) in line 89 is ambiguous since it seems to refer to the resolution at NADIR of the instrument not the HCHO/$NO_2$ ratio studies.

We have removed 'with 13 km x 24 km resolution' to avoid confusing readers. We have added recent publications (Chossiere et al. 2021) in the revised manuscript in Line 75.

Revised text:

Satellite retrievals of HCHO/$NO_2$ from Global Ozone Monitoring Experiment (GOME), SCanning Imaging Absorption spectroMeter for Atmospheric CartograpHY (SCIAMACHY), Ozone Monitoring Instrument (OMI) and TROPOspheric Monitoring Instrument (TROPOMI) have extended these $O_3$ sensitivity calculations over broad geographical regions (Chossière et al., 2021; Duncan et al., 2010; Jin et al., 2017; Martin et al., 2004; Schroeder et al., 2017a).

**RC4.6.** While discussing figure 3 and the seasonal trends on $O_3$ sensitivity elevated fire plumes are mentioned repeatedly to explain discrepancies. It would be a more convincing argument if those satellite observations were clearly associated to fires by quantitatively explaining the range of $NO_2$ and HCHO TROPOMI columns in fire and non-fire plumes as well as if considered necessary, utilizing other sources of information such as back trajectories or satellite aerosol retrievals.

The wildfire detection method used in the current manuscript uses ground-based measurements to detect rapid changes in concentrations that are indicative of wildfire impacts. Many of the wildfire plumes detected using this approach were also detected by TROPOMI. Figure 1 compares TROPOMI HCHO and $NO_2$ measurements based on a 5 km radius buffer during days classified as "no-wildfire" and "wildfire" based on ground-based measurements between August to October 2020. The median TROPOMI $NO_2$ and HCHO measurements on "wildfire" days are approximately 14% and 44% higher than measurements on "no-wildfire" days, respectively.

The transport of fire plumes is strongly affected by smoke injection height, which is a function of fire intensity. Plumes with large amounts of thermal energy can be injected above the daytime mixing depth and can be transported aloft without reaching the ground. These plumes would not trigger the ground-based wildfire detection method, but they would still be visible to TROPOMI. In September 2020, many wildfires occurred in high-elevation areas such as the Sierra Nevada Mountains to the east of the ground-based measurement site. We have visually checked satellite images provided by NASA WorldView and confirmed the presences of fire plumes transported from those mountainous areas (example below).

[Figure]

Image: September 8, 2020 (center: Sacramento; orange dots: fire locations detected by MODIS).

CALIPSO satellite products report the vertical profile of aerosols, but coverage over the study period is limited. More widely available aerosol optical depth (AOD) from MODIS MAIAC (1 km resolution; (Lyapustin et al., 2018)) confirms the presence of wildfire plumes during fall 2020, but does not differentiate between elevated plumes and plumes that reach the ground. We describe the potential influence of elevated plumes as a plausible explanation for the discrepancies between ground-based measurements and TROPOMI measurements, but further research would be needed to test this hypothesis.

Figure 1 has been added in the revised SI as Figure S7. A sentence has been added in revised manuscript to describe Figure 1 in Line 312:

'Figure S7 compares TROPOMI HCHO and NO$_2$ on wildfire days and non-wildfire days. Median TROPOMI HCHO measurements increased by 44% and TROPOMI NO$_2$ measurements increased by 14% on wildfire days.'

[Figure]

**Figure 1. Monthly box and whisker plot of TROPOMI HCHO and NO$_2$ in wildfire days (solid box) and non-wildfire days (open box) from August to October, 2020. TROPOMI HCHO and NO$_2$ is in the 5km radii buffer of the chamber measurement site in Sacramento.**

**RC4.7.** Figure 3 may be easier to interpret if the TRPOMI HCHO/NO$_2$ scale for panels a) and b) is similar. Why is it inverted in panel b)?

Figure 3a shows the monthly variation of O$_3$ response to NO$_x$ perturbation, while Figure 3b shows the monthly variation of O$_3$ response to VOC perturbation. Those two parameters have the exact opposite seasonal trend due to the NO$_x$-O$_3$-VOC chemistry. To show the consistency in the seasonal trend of chamber-measured and satellite-based O$_3$ sensitivity, we inverted the right Y axis (TROPOMI HCHO/NO$_2$) in Figure 3b. The result shows that the seasonal trend of chamber $\Delta O_3^{+VOC}$ is quite similar to the inverted TROPOMI HCHO/NO$_2$ trend. This helps to build confidence in the ground-based chamber measurements for $\Delta O_3^{+VOC}$.

**RC4.8.** Regarding the TROPOMI HCHO/NO$_2$ ratio, regime transition value of 4.6 it would be very interesting if the authors could provide some context of how it compares to previously published studies. As far as I can tell 4.6 is in the higher end of the values reported in the literature for urban areas.

As the reviewer recommended, we have added some context about previous studies of HCHO/NO$_2$ regime transition value in the past paragraph in Section 3.2 in revised manuscript:

Revised text:

'The HCHO/NO$_2$ transition point directly measured in the current study is consistent with previous estimates constructed from the combination of satellite measurements and routine ground-based O$_3$ monitoring data (Jin et al., 2020). Other previous efforts to estimate HCHO/NO$_2$ value at the transition point between NO$_x$-limited and VOC-limited regimes typically couple satellite HCHO/NO$_2$ measurements with O$_3$ sensitivity or O$_3$ sensitivity indicators (i.e., LNO$_x$/LRO$_x$) predicted using reactive chemical transport models. These hybrid studies predict HCHO/NO$_2$ transition points lower than the value of 4.6 derived in the current study. Martin (2004) used HCHO/NO$_2$ from GOME to calculate the regime transition value HCHO/NO$_2$=1.0 for polluted areas across the globe. Duncan (Duncan et al., 2010) used OMI to estimate the regime transition value HCHO/NO$_2$=1~2 across the continental U.S.. Schroeder (2017) found the transition range could between HCHO/NO$_2$=1.3~5.0 during DISCOVER-AQ in Houston. These estimated HCHO/NO$_2$ transition values vary due to the different satellite resolution, retrieval algorithms, and inherent air pollution patterns over the different study areas. The finer resolution satellite data used in the current study combined with direct ground-based measurements of O$_3$ sensitivity should provide accurate information for the HCHO/NO$_2$ transition point between chemical regimes over California.'

**RC4.9.** Do the correlation plots and equations shown in figure 7 and S6 consider the variance and uncertainty of both parameters?

We used ordinary lease square (OLS) linear regression in the original versions of Figure 7 and S6. This assumes that O$_3$ sensitivity measured in chambers has little to no error. This assumption is supported by the consistency tests described in the 3$^{rd}$ paragraph of Section 2.1 that show good agreement among 3 chambers, with only 1% uncertainty between measurements. In the revised manuscript, we repeated the test for the transition HCHO/NO$_2$ threshold between chemical regimes using reduced major axis (RMA) regression. RMA regression assumes both x and y variables include errors. Figure 2 (has been added in revised SI) compares the results of the two regression models. The RMA regression estimates a transition HCHO/NO$_2$=4.4 between chemical regimes, which is in good agreement with the original OLS result of 4.6.

We have described the method of linear regression (OLS) in the captions for Figures 7 and S6. We have also added the text below to Section 3.2 of the revised manuscript:

'Ordinary lease square (OLS) regression was used to estimate the transition point HCHO/NO$_2$=4.6 between chemical regimes. This approach does not account for uncertainty in chamber $\Delta O_3^{+NO_x}$. Repeating the analysis using reduced major axis (RMA) regression that accounts for errors in both x and y yields an estimated transition point HCHO/NO$_2$=4.4 between chemical regimes.'

[Figure]

**Figure 2.** Correlation between weekly averaged TROPOMI HCHO/NO$_2$ at 5 km circular buffers and the weekly averaged $\Delta O_3^{+NO_x}$ from ground-based measurement during non-wildfire days. The shaded area shows the 95% confidence interval of the mean response of the predicted value. Red regression line generated using ordinary least squares regression. Green regression line generated using reduced major axis regression.

**Reference:**

Chossière, G. P., Xu, H., Dixit, Y., Isaacs, S., Eastham, S. D., Allroggen, F., Speth, R. L. and Barrett, S. R. H.: Air pollution impacts of COVID-19–related containment measures, Sci. Adv., 7(21), doi:10.1126/SCIADV.ABE1178/ASSET/C9139A3B-89F3-4EC2-B4A0-1E6157B0E236/ASSETS/GRAPHIC/ABE1178-FX2.JPEG, 2021.

Duncan, B. N., Yoshida, Y., Olson, J. R., Sillman, S., Martin, R. V., Lamsal, L., Hu, Y., Pickering, K. E., Retscher, C., Allen, D. J. and Crawford, J. H.: Application of OMI observations to a space-based indicator of NOx and VOC controls on surface ozone formation, Atmos. Environ., 44(18), 2213–2223, doi:10.1016/j.atmosenv.2010.03.010, 2010.

Van Geffen, J. H. G. M., Eskes, H. J., Boersma, K. F. and Veefkind, J. P.: TROPOMI ATBD of the total and tropospheric NO 2 data products document number : S5P-KNMI-L2-0005-RP., 2021.

Ialongo, I., Virta, H., Eskes, H., Hovila, J. and Douros, J.: Comparison of TROPOMI/Sentinel-5 Precursor NO2 observations with ground-based measurements in Helsinki, Atmos. Meas. Tech., 13(1), 205–218, doi:10.5194/amt-13-205-2020, 2020.

Jin, X., Jin, X., Fiore, A., Fiore, A., Boersma, K. F., Boersma, K. F., Smedt, I. De and Valin, L.: Inferring Changes in Summertime Surface Ozone-NOx-VOC Chemistry over U.S. Urban Areas from Two Decades of Satellite and Ground-Based Observations, Environ. Sci. Technol., 54(11), 6518–6529, doi:10.1021/acs.est.9b07785, 2020.

Lyapustin, A., Wang, Y., Korkin, S. and Huang, D.: MODIS Collection 6 MAIAC algorithm, Atmos. Meas. Tech., 11(10), 5741–5765, doi:10.5194/AMT-11-5741-2018, 2018.

Martin, R. V., Fiore, A. M. and Van Donkelaar, A.: Space-based diagnosis of surface ozone sensitivity to

anthropogenic emissions, Geophys. Res. Lett., 31(6), 2–5, doi:10.1029/2004gl019416, 2004.

Schroeder, J. R., Crawford, J. H., Fried, A., Walega, J., Weinheimer, A., Wisthaler, A., Müller, M., Mikoviny, T., Chen, G., Shook, M., Blake, D. R. and Tonnesen, G. S.: New insights into the column CH2O/NO2 ratio as an indicator of near-surface ozone sensitivity, J. Geophys. Res. Atmos., 122(16), 8885–8907, doi:10.1002/2017JD026781, 2017.

Veefkind, J. P., Aben, I., McMullan, K., Förster, H., de Vries, J., Otter, G., Claas, J., Eskes, H. J., de Haan, J. F., Kleipool, Q., van Weele, M., Hasekamp, O., Hoogeveen, R., Landgraf, J., Snel, R., Tol, P., Ingmann, P., Voors, R., Kruizinga, B., Vink, R., Visser, H. and Levelt, P. F.: TROPOMI on the ESA Sentinel-5 Precursor: A GMES mission for global observations of the atmospheric composition for climate, air quality and ozone layer applications, Remote Sens. Environ., 120(2012), 70–83, doi:10.1016/j.rse.2011.09.027, 2012.

Verhoelst, T., Compernolle, S., Pinardi, G., Lambert, J. C., Eskes, H. J., Eichmann, K. U., Fjæraa, A. M., Granville, J., Niemeijer, S., Cede, A., Tiefengraber, M., Hendrick, F., Pazmiño, A., Bais, A., Bazureau, A., Folkert Boersma, K., Bognar, K., Dehn, A., Donner, S., Elokhov, A., Gebetsberger, M., Goutail, F., Grutter De La Mora, M., Gruzdev, A., Gratsea, M., Hansen, G. H., Irie, H., Jepsen, N., Kanaya, Y., Karagkiozidis, D., Kivi, R., Kreher, K., Levelt, P. F., Liu, C., Müller, M., Navarro Comas, M., Piters, A. J. M., Pommereau, J. P., Portafaix, T., Prados-Roman, C., Puentedura, O., Querel, R., Remmers, J., Richter, A., Rimmer, J., Cárdenas, C. R., De Miguel, L. S., Sinyakov, V. P., Stremme, W., Strong, K., Van Roozendael, M., Pepijn Veefkind, J., Wagner, T., Wittrock, F., Yela González, M. and Zehner, C.: Ground-based validation of the Copernicus Sentinel-5P TROPOMI NO2 measurements with the NDACC ZSL-DOAS, MAX-DOAS and Pandonia global networks, Atmos. Meas. Tech., 14(1), 481–510, doi:10.5194/AMT-14-481-2021, 2021.

Vigouroux, C., Langerock, B., Augusto Bauer Aquino, C., Blumenstock, T., Cheng, Z., De Mazière, M., De Smedt, I., Grutter, M., Hannigan, J. W., Jones, N., Kivi, R., Loyola, Di., Lutsch, E., Mahieu, E., Makarova, M., Metzger, J. M., Morino, I., Murata, I., Nagahama, T., Notholt, J., Ortega, I., Palm, M., Pinardi, G., Röhling, A., Smale, D., Stremme, W., Strong, K., Sussmann, R., Té, Y., Van Roozendael, M., Wang, P. and Winkler, H.: TROPOMI-Sentinel-5 Precursor formaldehyde validation using an extensive network of ground-based Fourier-transform infrared stations, Atmos. Meas. Tech., 13(7), 3751–3767, doi:10.5194/AMT-13-3751-2020, 2020a.

Vigouroux, C., Langerock, B., Augusto, C., Aquino, B., Blumenstock, T. and Cheng, Z.: TROPOMI – Sentinel-5 Precursor formaldehyde validation using an extensive network of ground-based Fourier-transform infrared stations, , 3751–3767, 2020b.

---

## Author Response (AR3)

**Response to the Editor — Article ACP-2021-708**

**Editor Comment 1**

For clarity in line 556, I suggest adopting the reviewer's statement that the annual decrease rates for VOCs and NOx obtained from ambient measurements correspond to a reduction by a factor of 10 for VOCs and a factor of 2.2 for NOx. Explicitly stating these factors makes it easier for the reader to compare them to the factors estimated from the emissions inventory.

**Response:**

This change has been made as requested on line 559 of the revised manuscript.